# S²MAM: Semi-supervised Meta Additive Model for Robust Estimation and Variable Selection

## Abstract

Semi-supervised learning with manifold regularization is a classical family for learning from both labeled and unlabeled data jointly, where the key requirement is that the support of the unknown marginal distribution possesses the geometric structure of a Riemannian manifold. Typically, the Laplace-Beltrami operator-based manifold regularization can be approximated empirically by the Laplacian regularization associated with the entire training data and its corresponding graph Laplacian matrix. However, the graph Laplacian matrix depends heavily on the pre-specifying similarity metric and may result in inappropriate penalties when facing redundant and noisy input variables. To address the above issues, this paper proposes a new *Semi-Supervised Meta Additive Model* (S²MAM) based on a bilevel optimization scheme, which automatically identifies informative variables, updates the similarity matrix, and achieves interpretable predictions simultaneously. Theoretical guarantees are provided for S²MAM, including the computing convergence and the statistical generalization bound. Experimental assessments on synthetic and real-world datasets validate the robustness and interpretability of the proposed approach.

## 1 Introduction

Manifold regularization provides an elegant and practical framework for developing semi-supervised learning (SSL) models by utilizing a large amount of unlabeled data in conjunction with limited labeled data (Belkin & Niyogi, 2004; Belkin et al., 2005; 2006; Geng et al., 2012; Van Engelen & Hoos, 2020; Yao & Xia, 2025). The key assumption of manifold regularization is that the support of the intrinsic marginal distribution has the geometric structure of a Riemannian manifold (Belkin & Niyogi, 2004; Belkin et al., 2006; Johnson & Zhang, 2007; 2008)). Usually, the Laplace-Beltrami operator-based manifold regularization can be approximated empirically by the Laplacian regularization associated with the whole training data and the corresponding similarity (adjacent) matrix (Belkin & Niyogi, 2004; Belkin et al., 2006; Roweis & Saul, 2000), where the similarity matrix is constructed by the principles of Gaussian fields and harmonic functions (Zhu et al., 2003b) or the local and global consistency (Zhou et al., 2003). Typical manifold regularization schemes include Laplacian regularized least squares (LapRLS) and Laplacian regularized support vector machine (LapSVM) (Belkin et al., 2006). Moreover, Nie et al. (2010) considered a flexible manifold embedding for semi-supervised dimension reduction, and Qiu et al. (2018) further developed an accelerated version (called fast flexible manifold embedding (f-FME)) by reconstructing a smaller adjacency matrix with low-rank and sparse constraints.

Despite rapid progress, it is still scarce to validate the intrinsic manifold assumption (Belkin & Niyogi, 2004; Belkin et al., 2006; Johnson & Zhang, 2007; 2008; Li et al., 2024) for different types of data, e.g., data with redundant or even noisy variables. Moreover, the investigation for the robustness and interpretability of manifold regularization is far below its empirical applications only concerning the prediction accuracy. The existing manifold regularization models require the similarity matrices to be pre-specified before the semi-supervised training procedures, where the adaptivity and robustness of manifold learning are largely unexplored. For real applications, they unavoidably involve some abundant irrelevant and even noisy variables, and the pre-specified similarity metric associated with all the variables can not reflect the true adjacent relations properly. The uninformative and noisy variables often result in a significant deviation in estimating the manifold structure, which seriously

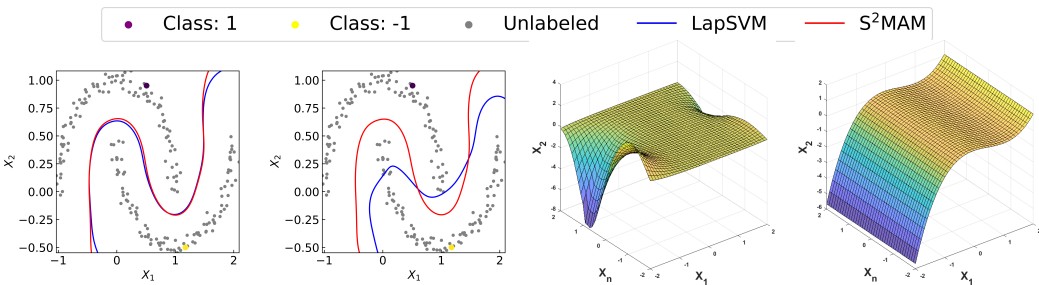

(a) Training on clean data  (b) Training on noisy data  (c) LapSVM on noisy data  (d) S²MAM on noisy data

Figure 1: Toy examples on the impact of noisy variables in the moon dataset for LapSVM and our S$^2$MAM. (a) and (b) show the 2D prediction curves w.r.t the original input $X_1$ and $X_2$, where LapSVM is sensitive to feature corruptions $X_n$. (c) and (d) present the 3D decision surfaces on corrupted data, where S$^2$MAM is robust against the varying noisy variable $X_n$. Clean moon dataset contains inputs, $X_1$ and $X_2$. The corrupted data involves another noisy variable $X_n \in \mathcal{N}(100, 100)$. The used moon dataset contains 99 unlabeled points and only one labeled point for each class. Please refer to *Appendix C.5* for detailed descriptions.

degrades the prediction capability of manifold regularization methods. As illustrated in Figure 1, the clean unlabeled data are beneficial to better fit the decision curve, while the randomly added noisy variables obviously hurt the performance of LapSVM (See *Appendix C.5* for detailed illustrations). The primary reason for the degraded performance is the computational bias in the similarity matrix, which directly affects the entire set of input variables (Nie et al., 2019; 2021). This motivates the following open questions:

"***How to alleviate the impact of redundant and even noisy variables on SSL models with manifold regularization? How to design a new manifold regularization scheme enjoying the robustness, interpretability, and prediction effectiveness simultaneously?***"

Intuitively, we can address the above questions using a two-stage framework, i.e., selecting the informative variables first (e.g., via Lasso (Tibshirani, 1994), SpAM (Ravikumar et al., 2009)) and then implementing manifold regularization approaches with the refined input variables. However, this variable selection strategy is independent of the intrinsic manifold structure, and its accuracy cannot be guaranteed due to the scarcity of labeled data. Inspired by meta-learning for coreset selection (Borsos et al., 2020; Zhou et al., 2022), this paper considers assigning masks to all input variables for both labeled and unlabeled data, retaining only those truly informative variables for modeling and constructing the similarity matrix.

Nevertheless, there are several challenges along this way: 1) It is NP-hard to learn the discrete mask variables taking values in $\{0, 1\}$ directly. 2) The bilevel optimization usually needs the computation on Hessian and Jacobian matrices, which leads to a heavy computation burden. 3) Most kernel-based manifold regularization models construct the Gram matrix based on sample distance, which lacks the result's interpretability, e.g., screening the key variables associated with the response.

### 1.1 CONTRIBUTION

To address the aforementioned challenges, we incorporate the meta-learning strategy and sparse additive models into a manifold-regularized SSL framework, and formulate a new *Semi-Supervised Meta Additive Model* (S$^2$MAM) to enable automatic variable masking and sparse approximation for high-dimensional inputs, even in the presence of noisy variables.

The core technique involves updating the decision function and similarity matrix simultaneously, using proper masks on input variables. The masks of S$^2$MAM are learned through a probabilistic meta-strategy. Moreover, an efficient implementation is employed here to solve the bilevel optimization problem, which avoids the heavy computing burden on the implicit hypergradient calculation (Pedregosa, 2016), Neumann series, and some variants with Hessian-vector or Jacobian-vector products (Ghadimi & Wang, 2018; Lorraine et al., 2020; Liu et al., 2022a).

The main contributions of this paper are summarized below:

Table 1: Properties of our S$^2$MAM and related models where "SSL" stands for semi-supervised learning. ($\checkmark$ = enjoying the given information, and $\times$ = not available for the information).

| | SpAM | LapRLS | f-FME | AWSSL | RER | SemiReward | PBCS | S$^2$MAM (ours) |
|---|---|---|---|---|---|---|---|---|
| Learning Task | Supervised | SSL | SSL | SSL | SSL | Supervised | Supervised | SSL |
| Optimization Framework | 1-level | 1-level | 1-level | 1-level | 1-level | 1-level | Bilevel | Bilevel |
| Interpretability | $\checkmark$ | $\times$ | $\times$ | $\times$ | $\times$ | $\times$ | $\times$ | $\checkmark$ |
| Variable Selection | $\checkmark$ | $\times$ | $\checkmark$ | $\checkmark$ | $\checkmark$ | $\times$ | $\times$ | $\checkmark$ |
| Noisy Variable Robustness | $\times$ | $\times$ | $\times$ | $\checkmark$ | $\checkmark$ | $\checkmark$ | $\times$ | $\checkmark$ |
| Convergence Analysis | $\times$ | $\times$ | $\times$ | $\times$ | $\times$ | $\times$ | $\checkmark$ | $\checkmark$ |
| Generalization Analysis | $\checkmark$ | $\times$ | $\times$ | $\times$ | $\times$ | $\times$ | $\times$ | $\checkmark$ |
| Computation Complexity Analysis | $\times$ | $\times$ | $\times$ | $\times$ | $\times$ | $\times$ | $\times$ | $\checkmark$ |

- *New statistical modeling.* To the best of our knowledge, our S$^2$MAM is the first meta-learning method for manifold-regularized additive models, where a novel bilevel optimization scheme is formulated to achieve robust estimation and data-driven automatic variable selection simultaneously. By assigning flexible masks to individual variables, the proposed S$^2$MAM is capable of reducing the impact of noisy variables on SSL tasks.

- *Computing and Theoretical Supports.* An efficient probabilistic bilevel optimization is developed to additionally learn the discrete masks, utilizing both policy gradient estimation and the projection operation. This computational algorithm alleviates the computational burden of the discrete bilevel optimization framework and provides theoretical guarantees of convergence in optimization. Additionally, we establish the upper bounds of excess risk for the baseline model of S$^2$MAM, which implies that the proposed approach can achieve polynomial decay in generalization error.

- *Empirical competitiveness.* Empirical results on several synthetic and real-world benchmarks demonstrate that the proposed S$^2$MAM can identify truly informative variables and achieve robust prediction even in the presence of redundant and noisy input variables.

## 1.2 COMPARISONS WITH THE RELATED WORKS

*Semi-supervised dimensionality reduction.* Recently, some efforts were made towards constructing a flexible similarity matrix against feature corruptions for SSL with manifold regularization (Chen et al., 2018; Nie et al., 2019; Bao et al., 2024). By rescaling the regression coefficients as variable weights, Chen et al. (2018) developed an efficient SSL method to identify important variables, known as rescaled linear square regression. Another weighting approach in (Nie et al., 2019) is called auto-weighting semi-supervised learning (AWSSL), which adaptively assigns continuous weights on variables to update the similarity matrix. After the dimension reduction process, a specific classifier is employed for downstream tasks. A robust graph learning (RGL) method (Kang et al., 2020) combined label ranking regression and label propagation into a unified framework for weight graph construction and semi-supervised learning. Semi-supervised adaptive local embedding learning (SALE) (Nie et al., 2021) adaptively constructs two affinity graphs (based on labeled data and all embedding samples) separately to explore the local and global structures. Bao et al. (2024) proposes an efficient model, robust embedding regression (RER), integrating the low-rank representation and Laplacian regularization. Unlike these works, this paper considers the automatic assignment of discrete masks (0/1) to input features (variables) for screening the truly active variables.

*Sparse additive models.* Additive models (Stone, 1985; Hastie & Tibshirani, 1990), as natural nonparametric extensions of linear models, have been burgeoning in high-dimensional data analysis due to their attractive properties, i.e., overcoming the curse of dimensionality, the flexibility of function approximation, and the ability of variable selection (Meier et al., 2009; Christmann & Hable, 2012; Yuan & Zhou, 2016; Chen et al., 2020). In recent years, many sparse additive models have been proposed from various theoretical or empirical motivations, see e.g., (Lv et al., 2018; Haris et al., 2022; Bouchiat et al., 2024; Duong et al., 2024). Naturally, the paradigm of additive models can be applied to semi-supervised learning settings. As far as we know, there are only three papers that touched on this topic (Culp & Michailidis, 2008; Culp et al., 2009; Culp, 2011). However, not all of them consider the robustness of manifold learning against noisy variables and ignore data-driven variable structure discovery. These stringent restrictions on the predefined similarity matrix and variable structure may result in a severe degradation of existing models under complex noise conditions.

*Meta learning for sample/variable selection.* The meta-based masking policy was developed in (Borsos et al., 2020), where a bilevel neural network is designed for automatic supervised coreset selection. Furthermore, its improved version with probabilistic bilevel optimization is proposed for supervised classification (Zhou et al., 2022), especially for corrupted and imbalanced data. Indeed, Zhou et al. (2022) also provides an example of variable selection, while it is limited to the supervised learning case and doesn't concern the impact of noisy variables. To the best of our knowledge, there has been no any endeavor before to explore the meta-based masking policy for semi-supervised additive models.

To better highlight the novelty of our $S^2$MAM, we summarize its properties in Table 1 compared with several related state-of-the-art models, including sparse additive models (SpAM) (Ravikumar et al., 2009)), LapRLS (Belkin et al., 2006)), fast flexible manifold embedding (f-FME) (Qiu et al., 2018), auto-weighting semi-supervised learning (AWSSL) (Nie et al., 2019), RER (Bao et al., 2024) SemiReward (Li et al., 2024) and the probabilistic bilevel coreset selection (PBCS) (Zhou et al., 2022). Table 1 shows that the proposed $S^2$MAM enjoys desirable properties, including variable selection, robust estimation, and computational guarantees.

## 2 SEMI-SUPERVISED ADDITIVE MODELS

This section first introduces a manifold-regularized semi-supervised additive model (Culp, 2011) as the basic model and then formulates the $S^2$MAM under the discrete bilevel optimization framework. Furthermore, a probabilistic bilevel scheme solves the NP-hard discrete optimization problem.

### 2.1 REVISITING MANIFOLD REGULARIZED SPARSE ADDITIVE MODEL

Let $\mathcal{X} = \{\mathcal{X}^{(1)}, \cdots, \mathcal{X}^{(p)}\} \in \mathbb{R}^p$ be a compact input space and the output space $\mathcal{Y} \in \mathbb{R}$. Denote $\rho$ as the joint distribution on $\mathcal{X} \times \mathcal{Y}$, and $\rho_{\mathcal{X}}$ as the marginal distribution with respect to $\mathcal{X}$ induced by $\rho$. The training set $\mathbf{z} = \{\mathbf{z}_l, \mathbf{z}_u\}$ involves the labeled set $\mathbf{z}_l = \{(x_i, y_i)\}_{i=1}^l$ and the unlabeled set $\mathbf{z}_u = \{x_i\}_{i=l+1}^{l+u}$, where each input $x_i = (x_i^{(1)}, \cdots, x_i^{(p)})^T \in \mathbb{R}^p$ with $x_i^{(j)} \in \mathcal{X}^{(j)}$ and output $y_i \in \mathbb{R}$. The hypothesis space of additive models can be formulated as $\mathcal{F} = \{f : f(x) = \sum_{j=1}^p f^{(j)}(x^{(j)}), f^{(j)} \in \mathcal{F}^{(j)}\}$, where $x^{(j)} \in \mathcal{X}^{(j)}$ and $\mathcal{F}^{(j)}$ is the component function space on $\mathcal{X}^{(j)}$ (Ravikumar et al., 2009). Typical candidates of additive hypothesis space include the basis expansion space (Meier et al., 2009; Ravikumar et al., 2009), the reproducing kernel Hilbert space (RKHS) (Raskutti et al., 2012), and the network-based space (Agarwal et al., 2021; Yang et al., 2020).

This paper chooses $\mathcal{H}_{K^{(j)}}$ to form the additive hypothesis space, where $\mathcal{H}_{K^{(j)}}$ is the RKHS associated with Mercer kernel $K^{(j)}$ defined on $\mathcal{X}^{(j)} \times \mathcal{X}^{(j)}, j \in \{1, \ldots, p\}$. Equipped by component function $f^{(j)} : \mathcal{X}^{(j)} \to \mathbb{R}, j \in \{1, \ldots, p\}$, the additive hypothesis space can be further defined as $\mathcal{H} = \{f = \sum_{j=1}^p f^{(j)} : f^{(j)} \in \mathcal{H}_{K^{(j)}}, 1 \le j \le p\}$ with $\|f\|_K^2 = \inf\{\sum_{j=1}^p \|f^{(j)}\|_{K^{(j)}}^2 : f = \sum_{j=1}^p f^{(j)}\}$. Indeed, $\mathcal{H}$ is an RKHS associated with kernel $K = \sum_{j=1}^p K^{(j)}$ (Christmann & Zhou, 2016). Due to the Representer Theorem of RKHS (Smola & Schölkopf, 1998), the prediction function of supervised additive models in RKHS often enjoys a parameterized representation (Yuan & Zhou, 2016)

$$f(\cdot) = \sum_{j=1}^p \sum_{i=1}^l \alpha_i^{(j)} K_i^{(j)}(x_i^{(j)}, \cdot). \tag{1}$$

Given a predictor $f : \mathcal{X} \to \mathbb{R}$, denote $\mathbf{f} = (f(x_1), \ldots, f(x_{l+u}))^T$ as the prediction vector associated with the labeled data $\mathbf{z}_l$ and the unlabeled data $\mathbf{z}_u$. Let $\lambda_1, \lambda_2 > 0$ be the regularization coefficients and let $\tau_j$ be the positive weight to different input variables for $j = 1, \cdots, p$. Then, the additive model for regularized Laplacian regression can be formulated as

$$f_{\mathbf{z}} = \arg\min_{f \in \mathcal{H}} \left\{ \mathcal{E}_{\mathbf{z}}(f) + \lambda_1 \Omega_{\mathbf{z}}(f) + \frac{\lambda_2}{(l+u)^2} \mathbf{f}^T \mathbf{L} \mathbf{f} \right\}, \tag{2}$$

where empirical risk $\mathcal{E}_{\mathbf{z}}(f) = \frac{1}{l} \sum_{i=1}^l (y_i - f(x_i))^2$, the sparse regularization $\Omega_{\mathbf{z}}(f)$ is formulated by $\inf_{\alpha^{(j)}} \{ \sum_{j=1}^p \tau_j \|\boldsymbol{\alpha}^{(j)}\|_2 : f = \sum_{j=1}^p \sum_{i=1}^l \alpha_i^{(j)} K_i^{(j)}(x_i^{(j)}, \cdot) \}$, and the term $\mathbf{f}^T \mathbf{L} \mathbf{f}$ is the manifold

regularization (Belkin & Niyogi, 2004; Culp, 2011). Here, $\boldsymbol{L} = \boldsymbol{D} - \boldsymbol{W}$ is the graph Laplacian, and diagonal matrix $\boldsymbol{D}$ satisfies $D_{ii} = \sum_{j=1}^{l+u} W_{ij}$ and $W_{ij}$ is the adjacent weight for inputs $x_i$ and $x_j$, e.g., $W_{ij} = \exp\{-\|x_i - x_j\|_2^2/\mu^2\}$ with bandwidth $\mu$.

**Remark 1** *If the $j$-th variable is not truly informative,* $\boldsymbol{\alpha_z^{(j)}} = (\alpha_{\mathbf{z},1}^{(j)}, \ldots, \alpha_{\mathbf{z},l+u}^{(j)})^T \in \mathbb{R}^{l+u}$ *is expected to satisfy* $\|\boldsymbol{\alpha_z^{(j)}}\|_2 = \sqrt{\sum_{i=1}^{l+u} \left|\alpha_{\mathbf{z},i}^{(j)}\right|^2} = 0$. *Thus, $\ell_{2,1}$-regularizer is employed as the penalty. Obviously, noisy input variables may bring an inappropriate similarity matrix $\boldsymbol{W}$. Naturally, it is necessary to improve the robustness of (2) against noisy variables by replacing the pre-specified similarity measure (i.e., $\boldsymbol{W}, \boldsymbol{L}$) in manifold regularization with an adaptive masking strategy.*

## 2.2 DISCRETE BILEVEL FRAMEWORK FOR S²MAM

To mitigate the negative impact of noisy variables on Laplacian regularization in (2), we introduce a bilevel optimization framework for automatically learning variable masks. In particular, both the decision function $f$ and Laplacian matrix $\boldsymbol{L}$ are updated by the learned masks.

Denote $\ell(\cdot)$ as the loss function, $f(x; \boldsymbol{\alpha})$ as a decision function in RKHS $\mathcal{H}$ with spanning parameter $\boldsymbol{\alpha}$ and the mask $\boldsymbol{m} \in \{0,1\}^p$ as a binary vector, where $m_i = 1$ implies $i$-th variable is selected as the informative one and otherwise ignored. $\alpha$ denotes the coefficient parameter of the additive model. The bilevel framework for directly learning the discrete masks is formulated as follows.

**Upper Level:** Given the meta dataset $D_{meta} = \{(x_i, y_i)\}_{i=1}^l$, we formulate the discrete optimization

$$\min_{\boldsymbol{m} \in \tilde{\mathcal{C}}} \mathcal{L}(\hat{\boldsymbol{\alpha}}(\boldsymbol{m})) = \frac{1}{l} \sum_{i=1}^l \ell(f(x_i; \hat{\boldsymbol{\alpha}}(\boldsymbol{m})), y_i), \tag{3}$$

where the mask $\boldsymbol{m}$ is the learnable parameter in the upper level, $\boldsymbol{\alpha}$ is the parameter of the decision function in the lower level depending on $\boldsymbol{m}$, and $\tilde{\mathcal{C}} = \{\boldsymbol{m} : m_i \in \{0,1\}, \|\boldsymbol{m}\|_0 \le C, i = 1, 2, \cdots, p\}$ is the feasible region of $\boldsymbol{m}$ with the size of selected variables $C$.

**Lower Level:** Based on the whole training set $D_{total}$ involving $D_{meta}$ and unlabeled samples $\{x_i\}_{i=l+1}^{l+u}$, the predictor of lower level optimization problem is

$$\hat{f}(x) = \sum_{j=1}^p \hat{f}^{(j)}(m_j x^{(j)}) = \sum_{j=1}^p \sum_{i=1}^l \alpha_i^{(j)} K_i^{(j)}(m_j x_i^{(j)}, m_j x^{(j)}), \tag{4}$$

where $\hat{\boldsymbol{\alpha}} = \arg\min_{\boldsymbol{\alpha} \in \mathbb{R}^{(l+u) \times p}} \mathcal{R}(\boldsymbol{\alpha}; \boldsymbol{m}; \boldsymbol{L})$, with risk $\mathcal{R}(\boldsymbol{\alpha}; \boldsymbol{m}; \boldsymbol{L}) = \frac{1}{l} \sum_{i=1}^l \ell(f(x_i \odot \boldsymbol{m}; \boldsymbol{\alpha}), y_i) + \lambda_1 \sum_{j=1}^p \tau_j \|\boldsymbol{\alpha}^{(j)}\|_2 + \frac{\lambda_2}{(l+u)^2} \mathbf{f}^T \boldsymbol{L} \mathbf{f}$.

Different from (2), the Laplacian matrix $\boldsymbol{L}$ is computed based on the masked similarity matrix $\boldsymbol{W}$ with measure function $\mathcal{W}(\cdot, \cdot)$ and element $W_{ij} = \mathcal{W}(x_i \odot \boldsymbol{m}, x_j \odot \boldsymbol{m}), i, j \in \{1, 2, \cdots, l+u\}$.

Usually, it is intractable to solve the above discrete bilevel problem directly. Fortunately, we can formulate its continuous probabilistic form with the help of policy gradient estimation (Zhou et al., 2022), and develop an efficient gradient-based optimization algorithm in the following section.

## 2.3 PROBABILISTIC BILEVEL FRAMEWORK FOR S²MAM

It is popular to transform the discrete tuning parameter space into the continuous probability space for bilevel optimization (Zhao et al., 2023; Zhou et al., 2022). For simplicity, $m_i$ can be considered as a Bernoulli random variable $m_i \sim \text{Bern}(s_i)$, where $s_i \in [0,1]$ represents the probability of $m_i = 1$. Denote the domain on probability variable $\boldsymbol{s} = (s_1, ..., s_p) \in \mathbb{R}^p$ as

$$\mathcal{C} = \{\boldsymbol{s} : 0 \preceq s_i \preceq 1, \|\boldsymbol{s}\|_1 \le C, i = 1, 2, \cdots, p\}. \tag{5}$$

The discrete bilevel optimization in Section 2.2 can be relaxed into the following expected form

$$\min_{\boldsymbol{s} \in \mathcal{C}} \Phi(\boldsymbol{s}) = \mathbb{E}_{p(\boldsymbol{m}|\boldsymbol{s})} \mathcal{L}(\boldsymbol{\alpha}^*(\boldsymbol{m})), \text{ s.t. } \boldsymbol{\alpha}^*(\boldsymbol{m}) \in \arg\min_{\boldsymbol{\alpha} \in \mathbb{R}^{(l+u) \times p}} \mathcal{R}(\boldsymbol{\alpha}; \boldsymbol{m}; \boldsymbol{L}). \tag{6}$$

**Remark 2** *Under the independent assumption on variable $m_i$, we can derive its distribution $p(\boldsymbol{m} \mid \boldsymbol{s}) = \Pi_{i=1}^{p}(s_i)^{m_i}(1-s_i)^{(1-m_i)}$. Since $\mathbb{E}_{\boldsymbol{m} \sim p(\boldsymbol{m}|\boldsymbol{s})}\|\boldsymbol{m}\|_0 = \sum_{i=1}^{p} s_i$, the original domain $\tilde{\mathcal{C}} = \{\boldsymbol{m} : m_i \in \{0,1\}, \|\boldsymbol{m}\|_0 \leq C, i = 1, 2, \cdots, p\}$ is transformed into $\mathcal{C}$ on probability $\boldsymbol{s}$.* *Relaxing the independence condition on $m_i$ is also meaningful in realistic scenarios in further research.*

## 2.4 Computing Algorithm of S$^2$MAM

Initialize the decision parameter $\boldsymbol{\alpha}^0 = \boldsymbol{0}$, mask $\boldsymbol{m}^0 = \boldsymbol{1}$, probability $\boldsymbol{s}^0 = \frac{C}{p} \cdot \boldsymbol{1}$ and select Laplacian matrix associated with original $(x_1, \cdots, x_{l+u})$ as $\boldsymbol{L}^0$. Before each iteration, a sample batch $\mathcal{B}$ is selected from the whole training set. The computing steps of probabilistic S$^2$MAM are summarized in Algorithm 1. The procedures for solving (6) at the $t$-th iteration contain:

**Step 1: Computing $\boldsymbol{\alpha}^t$ with $\boldsymbol{m}^{t-1}$ and $\boldsymbol{L}^{t-1}$,** where $\boldsymbol{\alpha}^t = \underset{\boldsymbol{\alpha} \in \mathbb{R}^{(l+u) \times p}}{\arg \min} \mathcal{R}(\boldsymbol{\alpha}^{t-1}; \boldsymbol{m}^{t-1}; \boldsymbol{L}^{t-1})$, with $\mathcal{R}(\boldsymbol{\alpha}^{t-1}; \boldsymbol{m}^{t-1}; \boldsymbol{L}^{t-1})$. The computing algorithm for Step 1, based on the alternating direction method of multipliers, is presented in *Appendix H.4*.

**Step 2: Computing $\boldsymbol{s}^t$ and $\boldsymbol{m}^t$ with $\boldsymbol{\alpha}^t$.** From the probabilistic S$^2$MAM in (6), the learning target changes from the discrete masks $\boldsymbol{m}$ into the continuous probability $\boldsymbol{s}$, which is updated by the policy gradient estimator (Zhou et al., 2022) as $\nabla_{\boldsymbol{s}}\Phi(\boldsymbol{s}) = \mathbb{E}_{p(\boldsymbol{m}|\boldsymbol{s})}\mathcal{L}(\boldsymbol{\alpha}^*(\boldsymbol{m}))\nabla_{\boldsymbol{s}} \ln p(\boldsymbol{m} \mid \boldsymbol{s})$. This computing procedure provides unbiased gradient estimation without a heavy computational burden on the inverse of the Hessian matrix or implicit differentiation.

Denote $\eta^t$ as the step size for updating the upper level parameter $\boldsymbol{s}$ at the $t$-th step. Given $\boldsymbol{\alpha}^t$, $\boldsymbol{s}$ can be updated by the projected stochastic gradient descent below

$$\boldsymbol{s}^t \leftarrow \mathcal{P}_{\mathcal{C}}\left(\boldsymbol{s}^{t-1} - \eta^t \mathcal{L}\left(\boldsymbol{\alpha}^t\right) \nabla_{\boldsymbol{s}} \ln p(\boldsymbol{m}^{t-1} \mid \boldsymbol{s}^{t-1})\right), \tag{7}$$

where the projection $\mathcal{P}_{\mathcal{C}}(\boldsymbol{s})$ from $\boldsymbol{s}$ to the domain $\mathcal{C}$ is summarized in Algorithm 2 in Appendix H.2. Then, $\boldsymbol{m}^t = (m_1^t, \cdots, m_p^t) \in \mathbb{R}^p$ follows from Bernoulli distribution, where $m_i^t \sim \text{Bern}\left(s_i^t\right)$. *Appendix H.1* proves the closed-form solution in the projection computation.

**Step 3: Updating Laplacian matrix $\boldsymbol{L}^t$ with $\boldsymbol{m}^t$**

$$\boldsymbol{L}^t = \boldsymbol{D}^t - \boldsymbol{W}^t, \tag{8}$$

where the diagonal matrix $\boldsymbol{D}^t \in \mathbb{R}^{(l+u) \times (l+u)}$ satisfies $D_{ii}^t = \sum_{j=1}^{l+u} W_{ij}$, and $W_{ij} = \exp\{-\|x_i \odot \boldsymbol{m}^t - x_j \odot \boldsymbol{m}^t\|_2^2/\mu^2\}$ with the bandwidth parameter $\mu > 0$. The metric $W_{ij}$ evaluates the similarity between samples $x_i$ and $x_j$ that share the same mask $\boldsymbol{m}^t$. Finally, we obtain the decision function in (4) with coefficient $\boldsymbol{\alpha}$ and mask $\boldsymbol{m}$.

To mitigate the quadratic cost of the graph Laplacian and probabilistic mask optimization on high-dimensional or large-size datasets, we've adopted two efficient strategies for acceleration, including preprocessing high-dimensional inputs (e.g., large images) via a pretrained CNN to extract a low-dimensional embedding, and replacing exact kernel evaluations with Random Fourier Features (RFF), which reduces complexity from $\mathcal{O}((l+u)^2)$ to $\mathcal{O}((l+u)D)$ where $D \ll l+u$.

# 3 Theoretical Assessments

For the proposed S$^2$MAM, this section presents its computational convergence and generalization analysis for the basic model (2) in Section 2.1. All proofs are left in *Appendices F&G*.

## 3.1 Computing Convergence Analysis

We now establish the theoretical guarantee of optimization convergence for the policy gradient estimation in Step 2. The following assumption has been widely used to characterize the convergence behavior of projection operation algorithms (Pedregosa, 2016; Zhou et al., 2022) and bilevel optimization with sample batches (Shu et al., 2023).

**Assumption 1** *Denote $\mathcal{L}_{\mathcal{B}}$ as the loss on selected batch $\mathcal{B}$. Assume that $\Phi(\boldsymbol{s})$ is L-smooth, constant $\sigma > 0$, there hold $\mathbb{E}[\mathcal{L}_{\mathcal{B}}(\boldsymbol{\alpha}^*(\boldsymbol{m}))\nabla_{\boldsymbol{s}} \ln p(\boldsymbol{m} \mid \boldsymbol{s}^t) - \nabla_{\boldsymbol{s}}\Phi(\boldsymbol{s}^t)] = 0$, and $\mathbb{E}\|\mathcal{L}_{\mathcal{B}}(\boldsymbol{\alpha}^*(\boldsymbol{m}))\nabla_{\boldsymbol{s}} \ln p(\boldsymbol{m} \mid \boldsymbol{s}^t) - \nabla_{\boldsymbol{s}}\Phi(\boldsymbol{s}^t)\|^2 \leq \sigma^2$.*

**Theorem 1** *At the $t$-th iteration, let the step size $\eta^t = \frac{c}{\sqrt{t}} \leq \frac{1}{L}$ for some constant $c > 0$, and denote the gradient mapping $\mathcal{G}^t = \frac{1}{\eta^t}\left(s^t - \mathcal{P}_\mathcal{C}\left(s^t - \eta^t \nabla_s \Phi\left(s^t\right)\right)\right)$. Under Assumption 1, there holds*

$$\min_{1 \leq t \leq T} \mathbb{E}\left\|\mathcal{G}^t\right\|^2 \lesssim \mathcal{O}\left(T^{-\frac{1}{2}}\right).$$

**Remark 3** *Indeed, Zhou et al. (2022) demonstrates that the average gradient $\frac{1}{T}\sum_{t=1}^{T}\mathbb{E}\left\|\mathcal{G}^t\right\|^2$ of the policy gradient estimation converges to a small constant as $T \to \infty$. With the help of refined step size $\eta^t = \frac{c}{\sqrt{t}}$, our results in Theorem 1 shows better convergence property w.r.t. $T$. The empirical and theoretical analysis of algorithmic computation complexity is left in Appendix E & H.5.*

### 3.2 GENERALIZATION ERROR ANALYSIS

The expected risk of $f : \mathcal{X} \to \mathcal{Y}$, w.r.t. $\mathcal{E}_\mathbf{z}(f)$ in (2), is measured by $\mathcal{E}(f) = \int_{\mathcal{X} \times \mathcal{Y}}(f(x) - y)^2 d\rho(x, y)$. It is well known that $f_\rho = \int_\mathcal{Y} y d\rho(y|\cdot)$ is the minimizer of $\mathcal{E}(f)$ over all measurable functions, where $\rho(y|x)$ denotes the conditional distribution of $y$ for given $x$. This work describes how fast $f_\mathbf{z}$ defined in (2) approximates $f_\rho$ as the sample size increases. To the best of our knowledge, this is the first theoretical endeavor to analyze the generalization behavior of semi-supervised additive models.

Before presenting our results, we recall some necessary assumptions and definitions involved here, which have been widely used in bounding the excess risk for supervised learning algorithms (Shi et al., 2011; Shi, 2013; Christmann & Zhou, 2016; Wang et al., 2023; Deng et al., 2023) and SSL models (Belkin et al., 2006; Liu & Chen, 2018; Chen et al., 2018).

**Assumption 2** *(Christmann & Zhou (2016)) For any $x \in \mathcal{X}$, there exists some $M \geq 0$ such that $\rho(\cdot \mid x)$ is almost everywhere supported on $[-M, M]$. Assume $f_\rho = \sum_{j=1}^{p} f_\rho^{(j)}$ with $0 < r \leq \frac{1}{2}$ and $f_\rho^{(j)} = L_{K^{(j)}}^r\left(g_j^*\right)$ with some $g_j^* \in L_2(\rho(\mathcal{X}^{(j)}))$ for any $j \in \{1, \ldots, p\}$, where $L_2(\rho(\mathcal{X}^{(j)}))$ is the square-integrable space on $\mathcal{X}^{(j)}$ and $L_{K^{(j)}}^r$ is $r$-power of integral operator $L_{K^{(j)}} : L_2(\rho(\mathcal{X}^{(j)})) \to L_2(\rho(\mathcal{X}^{(j)}))$ associated with kernel $K^{(j)}$.*

**Assumption 3** *Each entry of similarity matrix $\boldsymbol{W}$ satisfies $0 \leq W_{ij} \leq w$ for a positive constant $w$.*

**Assumption 4** *Let $C^v$ be a $\nu$-times continuously differentiable function set. Assume that $K^{(j)} \in C^\nu\left(\mathcal{X}^{(j)} \times \mathcal{X}^{(j)}\right), j \in \{1, \ldots, p\}$.*

Define $\pi(f)(x) = \max\{\min\{f(x), M\}, -M\}, \forall f \in \mathcal{H}$, as truncated output under Assumption 2. This truncated operator has been used extensively for error analysis of learning algorithms, see e.g., (Steinwart et al., 2009; Shi et al., 2019). Since $\mathcal{E}(\pi(f)) \leq \mathcal{E}(f)$ for any $f \in \mathcal{H}$, here we state the upper bound of $\mathcal{E}\left(\pi\left(f_\mathbf{z}\right)\right) - \mathcal{E}\left(f_\rho\right)$ to get a tighter generalization characterization for the manifold regularized additive model in (2).

**Theorem 2** *Let $\lambda_1 = (l + u)^{-\Delta}$, $\lambda_2 = \lambda_1^{1-r}$ for some $\Delta > 0$ and $0 < r \leq 1/2$. Under Assumptions 2-4, for any $0 < \delta < 1/2$, there holds*

$$\mathcal{E}\left(\pi\left(f_\mathbf{z}\right)\right) - \mathcal{E}\left(f_\rho\right) \lesssim \log(\frac{8}{\delta})\left(\mathcal{O}\left((l + u)^{-\Theta}\right) + \mathcal{O}\left(l^{-1/2}\right)\right),$$

*with confidence at least $1 - 2\delta$, where $\Theta = \min\{\Delta r, 1 + \Delta(r - 1), \Delta(5r/2 - 3/2) + 1/2, 2/(2 + \zeta), 3/2 - \Delta r, 1/2\}$ with $\zeta = \begin{cases} \frac{2}{1+2v}, & v \in (0, 1] \\ \frac{2}{1+v}, & v \in (1, 3/2] \\ \frac{1}{v}, & v \in (3/2, \infty) \end{cases}$.*

**Remark 4** *Theorem 2 guarantees the learning rate $\mathcal{O}(1/\sqrt{l})$ as setting $\Delta = 1$, $r = 1/2$, $v \to \infty$ and $u \geq l^2$, which interprets the role of unlabeled sample size $u$. Besides the additional advantage of the interpretability of input variables, the basic model (2) of $S^2MAM$ also achieves the polynomial decay rate of excess risk, which is comparable with supervised (Christmann & Zhou, 2016; Wang et al., 2023) and SSL models (Cao & Chen, 2012; Liu & Chen, 2018).*

Table 2: Average Accuracy $\pm$ standard deviation (%) on synthetic additive data for classification with fixed label percentages in each class ($r = 5\%$), uninformative variable ($p_u$) and noisy variable numbers ($p_n$).

| Model | r = 5%, $p_u = p_n = 0$ | | r = 5%, $p_u = 10, p_n = 0$ | | r = 5%, $p_u = 0, p_n = 10$ | | r = 5%, $p_u = p_n = 10$ | |
|---|---|---|---|---|---|---|---|---|
| | Unlabeled | Test | Unlabeled | Test | Unlabeled | Test | Unlabeled | Test |
| $\ell_1$-SVM | - | $83.914 \pm 6.410$ | - | $62.713 \pm 6.098$ | - | $62.261 \pm 6.550$ | - | $54.791 \pm 6.951$ |
| SpAM | - | $84.150 \pm 6.104$ | - | $65.091 \pm 5.917$ | - | $64.814 \pm 6.039$ | - | $54.413 \pm 6.295$ |
| CSAM | - | $86.597 \pm 5.424$ | - | $69.717 \pm 5.101$ | - | $65.178 \pm 5.255$ | - | $61.980 \pm 5.701$ |
| TSpAM | - | $86.993 \pm 5.340$ | - | $71.044 \pm 5.079$ | - | $67.340 \pm 4.959$ | - | $63.145 \pm 5.130$ |
| LapSVM | $88.814 \pm 5.398$ | $88.850 \pm 5.269$ | $59.992 \pm 5.259$ | $60.325 \pm 5.184$ | $55.630 \pm 8.213$ | $55.957 \pm 8.292$ | $55.137 \pm 8.414$ | $55.203 \pm 8.496$ |
| f-FME | $89.141 \pm 3.172$ | $89.305 \pm 3.359$ | $64.495 \pm 4.033$ | $64.611 \pm 4.208$ | $59.671 \pm 6.473$ | $59.801 \pm 6.655$ | $59.311 \pm 6.602$ | $59.407 \pm 6.659$ |
| AWSSL | $\mathbf{91.259 \pm 2.871}$ | $90.211 \pm 3.077$ | $83.691 \pm 3.423$ | $83.950 \pm 3.519$ | $73.701 \pm 4.105$ | $73.859 \pm 4.322$ | $72.255 \pm 4.211$ | $72.370 \pm 4.428$ |
| RGL | $90.422 \pm 2.909$ | $90.026 \pm 3.477$ | $84.065 \pm 4.501$ | $84.879 \pm 4.711$ | $77.726 \pm 4.591$ | $78.041 \pm 4.510$ | $75.155 \pm 4.965$ | $75.413 \pm 4.708$ |
| SALE | $89.717 \pm 2.811$ | $90.149 \pm 2.665$ | $85.742 \pm 4.132$ | $85.971 \pm 4.018$ | $79.071 \pm 4.709$ | $79.844 \pm 4.277$ | $77.201 \pm 4.697$ | $77.891 \pm 4.431$ |
| SSNP | $90.492 \pm 3.059$ | $89.871 \pm 3.218$ | $\mathbf{86.130 \pm 3.922}$ | $85.908 \pm 4.105$ | $78.250 \pm 4.294$ | $78.062 \pm 4.133$ | $77.462 \pm 4.412$ | $77.601 \pm 5.513$ |
| RER | $89.416 \pm 3.407$ | $89.930 \pm 3.622$ | $85.195 \pm 3.642$ | $85.870 \pm 3.703$ | $80.933 \pm 4.016$ | $81.049 \pm 4.055$ | $78.981 \pm 4.302$ | $79.112 \pm 4.517$ |
| S$^2$MAM (ours) | $89.979 \pm 3.255$ | $\mathbf{90.309 \pm 3.409}$ | $85.517 \pm 3.481$ | $\mathbf{86.015 \pm 3.575}$ | $\mathbf{81.702 \pm 3.897}$ | $\mathbf{81.855 \pm 4.055}$ | $\mathbf{80.012 \pm 4.177}$ | $\mathbf{80.112 \pm 4.370}$ |

## 4 EXPERIMENTAL EVALUATIONS

This section validates the effectiveness of S$^2$MAM on simulated and real-world data. All experiments are implemented in Python on RTX 3060 GPU and Intel Core i7 with 32 GB of memory. Due to space limitations, experiments on more synthetic, UCI and image datasets are left in *Appendices C-E*.

### 4.1 BASELINES AND PARAMETER SELECTION

**Baselines and Criterion:** For classification, the competitors include $\ell_1$-SVM (Zhu et al., 2003a), SpAM (with logistic loss) (Ravikumar et al., 2009), LapSVM (Belkin et al., 2006), f-FME (Qiu et al., 2018), AWSSL (Nie et al., 2019), RGL (Kang et al., 2020), SALE (Nie et al., 2021), RER (Bao et al., 2024), SemiReward (Li et al., 2024), Correntropy-based Sparse Additive Machine (CSAM) (Yuan et al., 2023), Tilted Sparse Additive Model (TSpAM) (Wang et al., 2023) and semi-supervised neural processes (SSNP) (Wang et al., 2022a). S$^2$MAM is equipped with the logistic loss. Similarity measure $W_{ij} = \exp\{-\|x_i - x_j\|_2^2/\mu^2\}$ and accuracy criterion are exploited. For the regression tasks, we compare the proposed S$^2$MAM with sparse supervised models (Lasso (Tibshirani, 1994) and SpAM (Ravikumar et al., 2009)), Deep Analytic Networks (DAN) (Dinh & Ho, 2020), LapRLS (Belkin et al., 2006), co-training regressor (COREG) (Lu et al., 2023) and deep SSL methods, including the variational autoencoder (VAE) (Cemgil et al., 2020) and the semi-supervised deep kernel learning (SSDKL (Jean et al., 2018) and pseudo-label filtering (PLF (Jo et al., 2024). For simplicity, the squared loss is selected as the loss function for SpAM and S$^2$MAM. The supervised methods are trained with merely labeled data. The mean squared error (MSE) is used as the criterion. Partial results are included in the Appendix, as SemiReward and PLF are primarily designed for image tasks.

**Hyperparameters:** For fairness, the penalty coefficients are tuned across $[10^{-4}, 10^{-3}, 10^{-2}, 10^{-1}]$ via leave-one-out cross-validation, which are shared for all regularized approaches. Let $\tau_j = 1$ for all $j \in [1, 2, \cdots, p]$ for additive baselines (Wang et al., 2023). The bandwidth $\mu$ for similarity measure is selected within $[10^{-4}, 10^{-3}, 10^{-2}, 10^{-1}, 1]$. We repeat each experiment 100 times and report the average accuracy as well as the standard deviation under different data settings. The selection of informative feature size $C$ is stated in Remark 6. The parameters for other methods were set according to the corresponding references.

**Benchmarks:** As stated in *Appendix B.1*, 4 synthetic, 8 UCI and 4 real-world datasets are utilized in the experiments, including the high-dimensional Alzheimer's Disease Neuroimaging Initiative (ADNI) clinical records, COIL-20 image, CelebA-HQ images (Lee et al., 2020), and AgeDB images (Moschoglou et al., 2017). To evaluate the robustness of S$^2$MAM, $p_u$ uninformative variables in $\mathcal{N}(0, 1)$ and $p_n$ noisy variables in $\mathcal{N}(100, 100)$ are designed as corruptions (Bao et al., 2024). Due to space limitations, empirical results on more datasets with interpretable visualizations are left in *Appendices C-E*. Notably, Table 12 verifies the efficiency when employing the random Fourier transformation (Rahimi & Recht, 2007; Wang et al., 2023) for accelerating the training process.

### 4.2 EXPERIMENTS ON SYNTHETIC DATA

Following the experimental design in (Chen et al., 2020; Wang et al., 2023), we consider the following additive discriminant function $f^*(x_i) = (x_i^{(1)} - 0.5)^2 + (x_i^{(2)} - 0.5)^2 - 0.08$, where

Table 3: Average MSE $\pm$ standard deviation of 10 repeated experiments on ADNI datasets with different label percentages ($r$) and noisy variable numbers ($p_n$). Notably, noisy features are drawn from $\mathcal{N}(100, 100)$. The upper and lower tables refer to prediction results on "Fluency" and "ADAS" cognitive scores of ADNI.

| Model | r = 20%, $p_n$ = 0 | | r = 20%, $p_n$ = 10 | | r = 50%, $p_n$ = 0 | | r = 50%, $p_n$= 10 | |
|---|---|---|---|---|---|---|---|---|
| | Unlabeled | Test | Unlabeled | Test | Unlabeled | Test | Unlabeled | Test |
| Lasso | - | $0.941 \pm 0.281$ | - | $1.359 \pm 0.733$ | - | $0.668 \pm 0.124$ | - | $0.833 \pm 0.474$ |
| SpAM | - | $0.831 \pm 0.228$ | - | $1.266 \pm 0.646$ | - | $0.589 \pm 0.110$ | - | $0.732 \pm 0.417$ |
| DAN | - | $0.794 \pm 0.197$ | - | $1.210 \pm 0.611$ | - | $0.637 \pm 0.105$ | - | $0.793 \pm 0.373$ |
| LapRLS | $0.915 \pm 0.301$ | $0.932 \pm 0.313$ | $1.478 \pm 0.812$ | $1.617 \pm 0.834$ | $0.823 \pm 0.215$ | $0.838 \pm 0.224$ | $1.142 \pm 0.511$ | $1.167 \pm 0.525$ |
| VAE | $0.743 \pm 0.324$ | $0.754 \pm 0.330$ | $0.812 \pm 0.397$ | $0.825 \pm 0.411$ | $0.474 \pm 0.115$ | $0.493 \pm 0.123$ | $0.526 \pm 0.226$ | $0.541 \pm 0.241$ |
| COREG | $0.748 \pm 0.308$ | $0.761 \pm 0.316$ | $0.984 \pm 0.423$ | $1.020 \pm 0.434$ | $0.527 \pm 0.276$ | $0.546 \pm 0.283$ | $0.513 \pm 0.384$ | $0.531 \pm 0.393$ |
| SSDKL | $\mathbf{0.721 \pm 0.321}$ | $\mathbf{0.739 \pm 0.337}$ | $0.848 \pm 0.446$ | $0.867 \pm 0.462$ | $0.442 \pm 0.271$ | $0.454 \pm 0.279$ | $0.524 \pm 0.391$ | $0.547 \pm 0.403$ |
| RER | $0.780 \pm 0.184$ | $0.794 \pm 0.201$ | $0.807 \pm 0.249$ | $0.821 \pm 0.266$ | $0.437 \pm 0.142$ | $0.448 \pm 0.157$ | $0.477 \pm 0.225$ | $0.496 \pm 0.249$ |
| $S^2$MAM (ours) | $0.730 \pm 0.133$ | $0.747 \pm 0.147$ | $\mathbf{0.786 \pm 0.214}$ | $\mathbf{0.804 \pm 0.228}$ | $0.423 \pm 0.119$ | $\mathbf{0.430 \pm 0.130}$ | $\mathbf{0.464 \pm 0.196}$ | $\mathbf{0.483 \pm 0.205}$ |
| Lasso | - | $1.179 \pm 0.376$ | - | $1.469 \pm 0.817$ | - | $0.824 \pm 0.255$ | - | $0.961 \pm 0.511$ |
| SpAM | - | $1.250 \pm 0.335$ | - | $1.545 \pm 0.748$ | - | $0.831 \pm 0.217$ | - | $1.017 \pm 0.470$ |
| DAN | - | $1.470 \pm 0.346$ | - | $1.844 \pm 0.773$ | - | $0.962 \pm 0.230$ | - | $1.672 \pm 0.515$ |
| LapRLS | $1.075 \pm 0.416$ | $0.973 \pm 0.423$ | $1.813 \pm 0.934$ | $1.706 \pm 0.945$ | $0.944 \pm 0.290$ | $0.898 \pm 0.296$ | $1.379 \pm 0.532$ | $1.409 \pm 0.544$ |
| VAE | $0.816 \pm 0.399$ | $0.808 \pm 0.418$ | $1.089 \pm 0.553$ | $0.924 \pm 0.571$ | $0.642 \pm 0.253$ | $0.633 \pm 0.261$ | $0.794 \pm 0.509$ | $0.760 \pm 0.521$ |
| COREG | $\mathbf{0.766 \pm 0.374}$ | $\mathbf{0.748 \pm 0.386}$ | $0.968 \pm 0.515$ | $0.735 \pm 0.528$ | $0.619 \pm 0.277$ | $0.625 \pm 0.285$ | $0.762 \pm 0.452$ | $0.736 \pm 0.467$ |
| SSDKL | $0.818 \pm 0.383$ | $0.794 \pm 0.396$ | $0.941 \pm 0.532$ | $0.920 \pm 0.541$ | $0.617 \pm 0.282$ | $\mathbf{0.605 \pm 0.269}$ | $0.772 \pm 0.473$ | $0.730 \pm 0.481$ |
| RER | $0.782 \pm 0.265$ | $0.801 \pm 0.273$ | $0.828 \pm 0.351$ | $0.817 \pm 0.358$ | $0.624 \pm 0.228$ | $0.618 \pm 0.208$ | $0.680 \pm 0.272$ | $0.698 \pm 0.287$ |
| $S^2$MAM (ours) | $0.771 \pm 0.241$ | $0.783 \pm 0.255$ | $\mathbf{0.816 \pm 0.321}$ | $\mathbf{0.801 \pm 0.330}$ | $0.614 \pm 0.204$ | $0.609 \pm 0.192$ | $\mathbf{0.663 \pm 0.251}$ | $\mathbf{0.681 \pm 0.266}$ |

Table 4: Extended experiments with average accuracy, standard deviation (SD), and training time cost (minutes) on COIL-20 image data. Merely 30% samples in the training set are labeled.

| Models | $\ell_1$-SVM | SpAM | CSAM | TSpAM | LapSVM | f-FME | AWSSL | RGL | SALE | SSNP | RER | SemiReward | $S^2$MAM |
|---|---|---|---|---|---|---|---|---|---|---|---|---|---|
| Accuracy | 67.329 | 69.917 | 73.577 | 72.230 | 81.092 | 85.518 | 86.821 | 83.416 | 87.235 | 83.370 | 85.219 | 87.518 | **88.211** |
| SD | 0.583 | 0.709 | 0.622 | 0.616 | 0.417 | **0.408** | 0.430 | 0.527 | 0.616 | 0.429 | 0.452 | 0.397 | 0.427 |
| Time Cost | **0.2** | 0.9 | 2.3 | 2.5 | 0.6 | 1.5 | 2.7 | 3.1 | 2.2 | 4.1 | 1.8 | 7.4 | 2.4 |

$x_i^{(j)} = (W_{ij} + U_i)/2$. $W_{ij}$ and $U_i$ are independently from $U(0, 1)$ for $i = 1, \cdots, 200$, $j = 1, \cdots, 100$. The category label satisfies $y_i = 0$ when $f(x_i) \leq 0$ and 1 otherwise. After equally dividing the entire dataset into training and testing sets, 5% samples for each class from the labeled set are randomly selected as the labeled set. As present in Table 2, both irrelevant and noisy features are harmful. Fortunately, even with irrelevant and noisy information, $S^2$MAM still exhibits superior prediction accuracy and stronger stability with the smallest variance compared to its supervised or semi-supervised competitors. Moreover, the extended visualization results in Figure 9 help to demonstrate the interpretability of $S^2$MAM more effectively.

### 4.3 EXPERIMENTS ON ADNI AND COIL DATASETS

As for the ADNI data, the records of "Fluency" and "ADAS" cognitive scores involving 326 features are selected as the identification targets. Table 3 demonstrates that $S^2$MAM enjoys competitive performance and even stronger robustness against variable corruptions compared to the other baselines, e.g., average 0.119 lower MSE on "ADAS" score with 20% labeled samples and 10 noisy features.

The following experiments are conducted for classifying the 12th and 13th objects in the COIL-20 image data. Inspired by some supervised (Su et al., 2023) and semi-supervised works (Qiu et al., 2018; Kang et al., 2020; Bao et al., 2024; Nie et al., 2019; 2021), a practical approach for dealing with high-dimensional data like COIL images is to extract the variable vectors first. As stated in *Appendix E*, a CNN is utilized to learn the vectors for each image, which realizes a rough dimensional reduction. However, this may not remove those irrelevant or even noisy variables (Nie et al., 2019; 2021). From the results in Table 4 above, $S^2$MAM provides competitive and robust prediction performance. See Tables 12-18 for results on noisy COIL, CelebA-HQ and AgeDB images with pixel-level corruptions.

### 5 CONCLUSION

This paper proposes a semi-supervised meta additive model, called $S^2$MAM, to enhance the robustness and interpretability of manifold regularization (Belkin et al., 2006) in settings with redundant and noisy input variables. Compared with existing SSL models with manifold regularization (Nie et al., 2019; Bao et al., 2024) and deep SSL models (Li et al., 2024; Jo et al., 2024), the proposed approach is capable of achieving variable selection, interpretability, and robust estimation simultaneously. Theoretical and empirical evaluations verify its superiority over some state-of-the-art learning models.

ETHICS STATEMENTS

This research does not raise any ethical concerns. The study exclusively involved the analysis of publicly available data sets and published literature, which did not contain any personally identifiable information. No human participants, animals, or sensitive data were involved in this research. All sources are properly cited in accordance with academic standards. The authors confirm that this work was conducted in accordance with the principles of academic integrity and research ethics.

REPRODUCIBILITY STATEMENT

We ensure full reproducibility by publicly releasing relevant materials, code, and data resources. All results were generated using fixed computational resources detailed in Section 4 and Appendices B-E. This enables independent verification of the findings of this research.

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

CONTENT OF THIS PAPER

# Appendix

## A  Notations

Some used notations are summarized in Table 5.

Table 5: Notations

| Notations | Descriptions |
|---|---|
| $p$ | the dimension of the input |
| $\mathcal{X}, \mathcal{Y}$ | the input space $\mathcal{X} = \{\mathcal{X}^{(1)}, \cdots, \mathcal{X}^{(p)}\} \in \mathbb{R}^p$ and the output space $\mathcal{Y} \subset \mathbb{R}$, respectively |
| $\rho$ | the jointed distribution on $\mathcal{X} \times \mathcal{Y}$ |
| $\rho_{\mathcal{X}}$ | the marginal distribution with respect to $\mathcal{X}$ induced by $\rho$ |
| $l/u$ | the number of labeled / unlabeled samples |
| $x_i; y_i$ | input $x_i = (x_i^{(1)}, \cdots, x_i^{(p)})^T \in \mathbb{R}^p$ with $x_i^{(j)} \in \mathcal{X}^{(j)}$; output $y_i \in \mathcal{Y}$ |
| $\mathbf{z}_l; \mathbf{z}_u$ | the labeled dataset $\mathbf{z}_l = \{(x_i, y_i)\}_{i=1}^l$; the unlabeled dataset $\mathbf{z}_u = \{x_i\}_{i=l+1}^{l+u}$ |
| $\mathcal{H}$ | the hypothesis space $\mathcal{H} = \left\{ f = \sum_{j=1}^p f^{(j)} : f^{(j)} \in \mathcal{H}_{K^{(j)}}, 1 \le j \le p \right\}$ |
| $\mathcal{H}_{K^{(j)}}$ | the RKHS associated with Mercer kernel $K^{(j)}$ defined on $\mathcal{X}^{(j)} \times \mathcal{X}^{(j)}, j \in \{1, \ldots, p\}$ |
| $L_{K^{(j)}}$ | integral operator $L_{K^{(j)}} : L_2(\rho(\mathcal{X}^{(j)})) \to L_2(\rho(\mathcal{X}^{(j)}))$ based on the square-integrable space $L_2$ |
| $L_{K^{(j)}}^r$ | the $r$-power of $L_{K^{(j)}}$ associated with feature $\mathcal{X}^{(j)}$ and kernel $K^{(j)}$ |
| $f(\cdot)$ | the prediction function of supervised additive models in RKHS where $f(\cdot) = \sum_{j=1}^p \sum_{i=1}^l \alpha_i^{(j)} K_i^{(j)}(x_i^{(j)}, \cdot)$ |
| $f^*$ | the ground truth function |
| $\mathbf{f}$ | the prediction vector $\mathbf{f} = (f(x_1), \ldots, f(x_{l+u}))^T$, associated with $\mathbf{z}_l$ and $\mathbf{z}_u$ |
| $f_{\mathbf{z}}$ | the empirical decision function of manifold regularized additive model |
| $\tau_j$ | the weight of $j$-th variable |
| $\alpha$ | the coefficient of the lower level additive model |
| $\boldsymbol{W}$ | the similarity matrix for SSL tasks |
| $\boldsymbol{D} ; \boldsymbol{L}$ | the diagonal matrix $D_{ii} = \sum_{j=1}^{l+u} W_{ij}$; the graph Laplacian $\boldsymbol{L} = \boldsymbol{D} - \boldsymbol{W}$ |
| $\boldsymbol{m}$ | the variable mask vector $\boldsymbol{m} \in \{0, 1\}^p$ |
| $\boldsymbol{s}$ | the vector $\boldsymbol{s} = (s_1, \cdots, s_p)$ where $s_i$ stands for the probability of $m_i = 1$ |

## B  Descriptions for Benchmarks and Baselines

In this paper, we select 4 synthetic datasets and 12 real-world datasets for our experiments. Indeed, these datasets have been widely used for validating additive models (Ravikumar et al., 2009; Lahiri et al., 2016; Chen et al., 2020; Wang et al., 2023) or semi-supervised learning models (Jean et al., 2018; Qiu et al., 2018; Nie et al., 2019; 2021; Bao et al., 2024). We briefly summarize the datasets used and some learning methods for baselines as follows.

### B.1  Data Description

Denote $N$ and $p$ ($p = p^* + p_u + p_n$) as the total number of samples and the dimensions in each dataset, where the training set involves $l$ labeled data and $u$ unlabeled data, and the remaining samples are left for testing. We generate $p_u$ uninformative variables and $p_n$ noisy variables, which are added into the truly informative variables $p^*$ from all samples within the dataset (including the training and testing sets).

The 16 datasets used in this paper include:

- (1) Friedman data for regression. The corresponding generation function is provided in the experiment section, which involves 200 samples, $p^* = 5$ true informative features, and $p_u = 95$ uninformative features following $\mathcal{N}(0, 1)$. And $p_n = 10$ noisy features in

$\mathcal{N}(100, 100)$ are also considered to highlight the robustness better. Denote $\epsilon$ as the Gaussian noise $\mathcal{N}(0, 1)$, the output $y$ is generated by

$$f(X) = 10\sin\left(\pi X^{(1)} X^{(2)}\right) + 20\left(X^{(3)} - 0.5\right)^2 + 10X^{(4)} + 5X^{(5)} + \epsilon.$$

- (2) Synthetic additive data for regression. It involves $N = 200$ samples, $p^* = 8$ true informative features, and $p_u = 92$ uninformative features. We also consider adding $p_n = 10$ noisy features following $\mathcal{N}(100, 100)$ into the whole dataset,

$$Y = f^*(X) + \epsilon = \sum_{j=1}^{8} f^{(j)}(X^{(j)}) + \epsilon, \tag{9}$$

where $f^{(1)}(u) = -2\sin(2u)$, $f^{(2)}(u) = 8u^2$, $f^{(3)}(u) = \frac{7\sin u}{2 - \sin u}$, $f^{(4)}(u) = 6e^{-u}$, $f^{(5)}(u) = u^3 + \frac{3}{2}(u - 1)^2$, $f^{(6)}(u) = 5u$, $f^{(7)}(u) = 10\sin(e^{-u/2})$, $f^{(8)}(u) = -10\widetilde{\phi}(u, \frac{1}{2}, \frac{4}{5}^2)$. Notably, to validate the additive models on testing sets, the Gram matrices or new splined features for the testing sets must be generated.

- (3) Synthetic additive data for classification. It involves $N = 200$ samples, $p^* = 2$ informative features, $p_u = 98$ uninformative redundant features following $\mathcal{N}(0, 1)$ and $p_n = 10$ noisy features following $\mathcal{N}(100, 100)$, and the output

$$f^*(x_i) = (x_i^{(1)} - 0.5)^2 + (x_i^{(2)} - 0.5)^2 - 0.08,$$

where $x_i^{(j)} = (W_{ij} + U_i)/2$. $W_{ij}$ and $U_i$ are independently from $U(0, 1)$ for $i = 1, \cdots, 200$, $j = 1, \cdots, 100$. The label satisfies $y_i = 0$ when $f(x_i) \leq 0$ and 1 otherwise. This synthetic data for classification has been widely used in some existing research for evaluating the performance of additive models (Chen et al., 2020; Wang et al., 2023)

- (4) Synthetic Moon data for classification. It involves two classes with a total of 200 samples, $p^* = 2$ informative features, $p_u =$ uninformative, redundant features, and $p_n =$ additional, noisy features. This data has been widely used for estimating the model's capability for correctly identifying different categories (Qiu et al., 2018; Nie et al., 2019; 2021).

- (5) Alzheimer's Disease Neuroimaging Initiative (ADNI) dataset for regression. To better highlight the robustness in real-world applications, the ADNI (https://adni.loni.usc.edu/) dataset (795 instances, p = 326) is also considered.

- Four datasets from the UCI repository for regression.

  (6) Buzz prediction on the Twitter dataset for regression. It involves a total of $38,393$ samples, $p^* = 77$ original features, and additional $p_n = 10$ noisy features. This dataset helps to predict the mean number of active discussions.

  (7) Boston Housing Price dataset for regression. It involves merely 506 samples, $p^* = 13$ original features, and additional $p_n = 10$ noisy features. This dataset has been widely used for estimating the performance of regression models.

  (8) Ozone Level Detection dataset for regression. It includes $N = 2536$ instances with $p^* = 73$ attributes, aiming to forecast ground ozone pollution using the given features. We also add $p_n = 10$ noisy features into the original dataset.

  (9) SkillCraft Master dataset for regression. The dataset is made of $N = 3395$ observations and $p^* = 19$ input variables. And $p_n = 10$ noisy features are further added to the original dataset.

- Four datasets from the UCI repository for classification.

  (10) Predicting Buzz Magnitude in the Social Media dataset for classification. It involves $N = 38393$ instances with $p^* = 77$ original features. We further add $p_n = 10$ noisy features into the original datasets for comparing the robustness of these baselines.

  (11) Breast Cancer Wisconsin dataset for classification. There are 569 instances and $p^* = 29$ original input features. $p_n = 10$ noisy features following $\mathcal{N}(100, 100)$ are further added into the original dataset.

  (12) Phishing Websites dataset for classification. It contains 31 columns, with 30 features and one target. The dataset has 2456 observations.

(13) Statlog (Heart) dataset for classification. It involves $N = 270$ instances with $p^* = 13$ input features. Noisy features are further added for comparison.

- Three image datasets for classification or regression.

(14) The image data from the COIL20 image library, which initially contains 20 objects, is used for classification. For simplicity, the 12th and 13th digits are selected, where there are $N = 72$ instances for each digit and $p^* = 16384$ original features (gray images with a size of $128 \times 128$). This dataset has been used for evaluating the prediction performance of semi-supervised learning models on feature reduction (Nie et al., 2019; 2021).

(15) CelebA-HQ images, which were initially derived from the original CelebA, are used for classification. For simplicity, the 12th and 13th digits are selected, where there are $N = 30,000$ instances for each digit and $p^* = 262,144$ original features (with a size of $512 \times 512$).

(16) AgeDB is a specialized facial image dataset that comprises over $N = 16,000$ high-quality facial images of 568 distinct subjects, with each subject represented across a significant age span (averaging 13.0 years between the youngest and oldest images per identity). All photos are standardized to a uniform resolution of $224 \times 224$ pixels ($p^* = 50,176$), ensuring consistency for model training and evaluation.

The above real-world datasets have undergone preliminary data cleaning, where those entries with empty values are filled with mean values, or even removed when significant features are missing (ratio of missing features $\geq 20\%$).

### B.2    BASELINES & PARAMETER SETTINGS

#### B.2.1    REGRESSION TASKS

The baselines for regression tasks include:

- (1) Lasso (Tibshirani, 1994), is a type of supervised linear regression model that is used for variable selection with sparsity-induced regularization. The regularization parameter $\lambda$ is tuned across $[10^{-4}, 10^{-3}, 10^{-2}, 10^{-1}, 1]$.

- (2) SpAM (Ravikumar et al., 2009), is an additive supervised nonparametric model for high-dimensional nonparametric regression and classification tasks. The regularization parameter $\lambda$ is tuned across $[10^{-4}, 10^{-3}, 10^{-2}, 10^{-1}, 1]$.

- (3) DAN (Dinh & Ho, 2020) is designed to identify a subset of relevant features in deep learning models. The core technology employs the adaptive group Lasso selection procedure, with group Lasso serving as the base estimator, which has been demonstrated to be selection-consistent for a broad class of networks.

- (4) LapRLS (Belkin et al., 2006), learns a semi-supervised linear model using the labeled data by minimizing a regularized least squares objective function. The regularization term incorporates the graph Laplacian matrix, which captures the assumption of smoothness, where similar points are expected to have similar labels. The regularization parameters $\lambda_1$ and $\lambda_2$ are both tuned across $[10^{-4}, 10^{-3}, 10^{-2}, 10^{-1}, 1]$.

- (5) Variational autoencoder (VAE) (Goodfellow et al., 2014), is designed as a semi-supervised generative model by first learning an unsupervised embedding of the data and then using the embeddings as input to a supervised multilayer perceptron.

- (6) Co-training regressor (COREG) (Lu et al., 2023), is a co-training algorithm for regression tasks that uses two $k$-NN regressors with different distance metrics. During the training process, each regressor generates labels for the other.

- (7) Semi-supervised deep kernel learning (SSDKL) (Jean et al., 2018), is a semi-supervised regression model based on minimizing predictive variance in the posterior regularization framework. It combines the hierarchical learning of networks with the probabilistic modeling capabilities of Gaussian processes.

- (8) Pseudo-label filtering (PLF) (Jo et al., 2024) is a novel semi-supervised regression framework for extending SSL methodologies beyond classification tasks. It first filters unreliable pseudo-labels through uncertainty estimation and then refines the remaining

pseudo-labels through similarity-based information propagation from labeled to unlabeled examples.

- (9) SemiReward (Li et al., 2024) is a general and pluggable reward framework designed for semi-supervised learning that evaluates and selects high-quality pseudo-labels to enhance both performance and convergence speeds of self-training techniques. SemiReward implements an efficient two-stage training pipeline assisted by a generator network and a lightweight rewarder network.

For fairness, a network with a $[d - 100 - 50 - 50 - 2]$ structure is employed here for the downstream regression task. Following (Jean et al., 2018), the same base network is shared for all deep semi-supervised models, including VAE and SSDKL. The learning rates for the neural network and the Gaussian process are $10^{-3}$ and $10^{-1}$, respectively. The training process of VAE, COREG, and SSDKL follows the settings in (Jean et al., 2018). Besides, the bandwidth $\mu$ for the Gaussian similarity function ($W_{ij} = \exp\{-\|x_i - x_j\|_2^2/\mu^2\}$) is also tuned across $[10^{-4}, 10^{-3}, 10^{-2}, 10^{-1}, 1]$ for all SSL methods for computing the similarity and Laplacian matrices. Notice that the similarity matrix for S$^2$MAM is calculated by $W_{ij} = \exp\{-\|x_i \odot \boldsymbol{m} - x_j \odot \boldsymbol{m}\|_2^2/\mu^2\}$ with learned mask $\boldsymbol{m}$, $i, j \in \{1, 2, \cdots, l + u\}$. In practice, the proportion of labeled points in a single batch is consistent with the settings in the whole training set to avoid empty labeled sets or inconsistency among each batch.

Notably, both PLF and SemiReward are designed with specific modules, such as generative networks, for processing images. Thus, they are adopted in the experiments on COIL-20, CelebA-HQ, and AgeDB images in this paper, rather than the synthetic tubular data or the UCI datasets.

### B.2.2 CLASSIFICATION TASKS

The baselines for classification tasks include:

- (10) $\ell_1$-SVM (Zhu et al., 2003a), is a supervised classification model with $\ell_1$ sparse regularization based on the classical SVM. The regularization parameter $\lambda$ is tuned across $[10^{-4}, 10^{-3}, 10^{-2}, 10^{-1}, 1]$.

- (11) SpAM (induced by logistic loss) (Ravikumar et al., 2009), is equipped with logistic loss for classification, which has been introduced above. Its regularization parameter $\lambda$ is tuned across $[10^{-4}, 10^{-3}, 10^{-2}, 10^{-1}, 1]$.

- (12) LapSVM (Belkin et al., 2006), utilizes the concept of the graph Laplacian, which captures the underlying manifold structure of the data. The objective of LapSVM is to find a decision boundary that not only separates the labeled data accurately but also respects the smoothness assumption captured by the graph Laplacian. The regularization parameters $\lambda_1$ and $\lambda_2$ are both tuned across $[10^{-4}, 10^{-3}, 10^{-2}, 10^{-1}, 1]$

- (13) f-FME (Qiu et al., 2018) is an improved version of classical flexible manifold embedding (FME) that employs additional anchor graphs to reduce the time cost and computational burden of FME.

- (14) AWSSL (Nie et al., 2019), is a semi-supervised learning model that constructs an adaptive graph for propagating label information and using special strategies for ranking the importance of variables. An auto-weighting matrix is learned to select informative variables from both labeled and unlabeled data.

- (15) RGL (Kang et al., 2020) constructs a graph from the pristine data derived from restored technology, subsequently utilizing this resilient graph to improve the performance of semi-supervised classification tasks.

- (16) SALE (Nie et al., 2021) merges the processes of adaptive graph formation and label dissemination into a singular optimization framework, simultaneously developing an automatic weighting matrix that discerns and emphasizes significant variables across the entire dataset.

- (17) CSAM (Yuan et al., 2023) utilizes a robust error metric based on the statistical correntropy measure, which yields a robust additive model for classification with noisy labels.

- (18) TSpAM (Wang et al., 2023) constructs a robust additive model based on the tilted empirical risk. It's capable of robust estimation and imbalanced classification. Notably, an efficient random Fourier features approach is used to accelerate the kernel-based computation.

---

**Algorithm 1:** Computing Procedure for S$^2$MAM

---

**Input**: Labeled data $\mathbf{z}_l = \{(x_i, y_i)\}_{i=1}^{l}$, unlabeled data $\mathbf{z}_u = \{x_i\}_{i=l+1}^{l+u}$, step size $\eta^t$, core size
   $C, \mathbf{1} = (1, ..., 1) \in \mathbb{R}^p$.
**Initialization**: $\boldsymbol{\alpha}^0, \boldsymbol{s}^0 = \frac{C}{p} \cdot \mathbf{1}, \boldsymbol{m}^0, \boldsymbol{L}^0$.
  **for** $t = 1$ to $T$ **do**
    1) Update $\boldsymbol{\alpha}^t$ based on Step 1 with $\mathbf{z}_l$ & $\mathbf{z}_u$
    2) Update $\boldsymbol{s}^t$ based on Step 2 with $\mathbf{z}_l$
    3) Update $\boldsymbol{m}^t$ sampled from $p(\boldsymbol{m}|\boldsymbol{s}^t)$
    4) Update $\boldsymbol{L}^t$ based on Step 3 with $\mathbf{z}_l$ & $\mathbf{z}_u$
  **end for**
**Output**: Decision function $\hat{f}$.

---

- (19) SSNP (Wang et al., 2022a) integrates neural processes with semi-supervised learning for image classification tasks. The innovation lies in adapting NPs, a probabilistic model that approximates Gaussian Processes, to the SSL framework. The CNN structure is slightly modified to satisfy 1D value-based inputs.

- (20) Robust Embedding Regression (RER) (Bao et al., 2024) is a novel semi-supervised learning approach that addresses the performance degradation of existing methods when confronted with noisy and redundant data. RER adaptively constructs weighted graphs, incorporating low-rank representation to reduce noise and redundancy, and applies appropriate norm constraints for feature selection and improved model stability.

- (21 / 9) SemiReward (Li et al., 2024) is also capable of regression estimation on image data (e.g., AgeDB images). Please refer to the regression baseline (9) for a detailed description.

### B.2.3 ALGORITHM AND PARAMETER SETTINGS

Before introducing the detailed parameter settings, we first present Algorithm 1, which summarizes the computational process of our S$^2$MAM. For simplicity, the parameter $\tau_j = 1$ for all $j \in \{1, 2, \cdots, p\}$. The regularization parameters for regularized models are all tuned across $[10^{-4}, 10^{-3}, 10^{-2}, 10^{-1}, 1]$. As introduced in (Qiu et al., 2018; Nie et al., 2021; Bao et al., 2024), the 1-nearest neighbor (1NN) classifier with Euclidean distance is recommended for evaluating classification accuracy after dimension reduction. The number of selected variables, $C$, is shared for S$^2$MAM and those baselines used for dimension reduction.

To avoid singular solutions or unfair comparisons, each experiment has been repeated 20 times, and the similarity (weight) graph is constructed following (Nie et al., 2019; 2021; Bao et al., 2024) for those baselines with the Laplacian matrix. Each dataset is divided into training and testing sets with a ratio of $1 : 1$. Then we select $l$ samples from each class as the labeled set, and the remaining training samples are considered the unlabeled set. Every semi-supervised method that employs two regularization coefficients is evaluated on the grid $(\lambda_1, \lambda_2) \in \{10^{-4}, 10^{-3}, 10^{-2}, 10^{-1}, 1\}$. Supervised baselines with a single penalty (Lasso, $\ell_1$-SVM, SpAM, TSpAM) search the coefficient in $\{10^{-4}, 10^{-3}, 10^{-2}, 10^{-1}, 1\}$, which also aligns with the settings in their publications (SpAM, TSpAM). The 1-nearest neighbor classifier with Euclidean distance is employed in f-FME and AWSSL. Furthermore, $\tau$ within regularization was utilized to provide flexibility in assigning different weights to variables based on prior knowledge or importance.

The leave-one-out cross-validation strategy is utilized for parameter tuning, given the rarity of labeled samples. Fortunately, the leave-one-out cross-validation is utilized due to limited labeled data, which does not require a separate validation set and may not be heavily dependent on specific validation sets (Hastie et al., 2009). The rest parameters for the other methods were set according to their corresponding references.

Table 6: Average MSE $\pm$ standard deviation on synthetic regression data with different label percentages ($r$) and noisy variable numbers ($p_n$). The upper and lower tables show the results on the Friedman data and the additive data. Notably, some deep SSL approaches provide better prediction performance under clean scenarios, i.e., when the number of noisy variables $p_n = 0$.

| Model | r = 5%, $p_n$ = 0 | | r = 5%, $p_n$ = 10 | | r = 10%, $p_n$ = 0 | | r = 10%, $p_n$ = 10 | |
|---|---|---|---|---|---|---|---|---|
| | Unlabeled | Test | Unlabeled | Test | Unlabeled | Test | Unlabeled | Test |
| Lasso | - | $15.579 \pm 12.396$ | - | $22.135 \pm 14.442$ | - | $8.684 \pm 2.393$ | - | $15.636 \pm 7.785$ |
| SpAM | - | $14.791 \pm 11.595$ | - | $21.055 \pm 13.744$ | - | $8.201 \pm 2.464$ | - | $14.706 \pm 7.577$ |
| DAN | - | $12.417 \pm 7.947$ | - | $23.350 \pm 7.074$ | - | $7.864 \pm 2.017$ | - | $17.392 \pm 5.283$ |
| LapRLS | $11.659 \pm 5.024$ | $11.678 \pm 5.125$ | $27.299 \pm 8.549$ | $27.588 \pm 8.779$ | $8.086 \pm 2.000$ | $8.103 \pm 1.970$ | $23.822 \pm 4.498$ | $23.918 \pm 4.457$ |
| VAE | $11.071 \pm 7.011$ | $11.499 \pm 7.971$ | $20.194 \pm 9.477$ | $20.860 \pm 9.977$ | $7.866 \pm 3.752$ | $7.950 \pm 4.873$ | $15.155 \pm 4.950$ | $15.809 \pm 5.134$ |
| COREG | $10.573 \pm 6.855$ | $\mathbf{10.730 \pm 6.946}$ | $19.011 \pm 7.644$ | $19.644 \pm 7.945$ | $7.801 \pm 3.011$ | $7.820 \pm 3.401$ | $15.305 \pm 4.117$ | $15.914 \pm 4.955$ |
| SSDKL | $\mathbf{10.144 \pm 6.917}$ | $10.744 \pm 7.301$ | $19.410 \pm 7.809$ | $19.655 \pm 8.137$ | $\mathbf{7.035 \pm 7.155}$ | $\mathbf{7.195 \pm 7.511}$ | $14.101 \pm 4.055$ | $14.731 \pm 4.773$ |
| S$^2$MAM (ours) | $10.837 \pm 4.355$ | $11.350 \pm 4.881$ | $\mathbf{12.274 \pm 5.101}$ | $\mathbf{12.941 \pm 5.807}$ | $7.204 \pm 2.591$ | $7.430 \pm 2.473$ | $\mathbf{8.418 \pm 3.140}$ | $\mathbf{8.701 \pm 3.433}$ |
| Lasso | - | $1.193 \pm 0.437$ | - | $2.706 \pm 3.174$ | - | $1.079 \pm 0.304$ | - | $2.102 \pm 0.705$ |
| SpAM | - | $1.122 \pm 0.422$ | - | $2.597 \pm 2.848$ | - | $1.033 \pm 0.301$ | - | $1.955 \pm 0.727$ |
| DAN | - | $1.217 \pm 0.346$ | - | $2.133 \pm 1.294$ | - | $1.014 \pm 0.232$ | - | $1.792 \pm 0.538$ |
| LapRLS | $1.025 \pm 0.121$ | $1.073 \pm 0.182$ | $3.571 \pm 0.138$ | $3.592 \pm 0.171$ | $0.986 \pm 0.136$ | $1.055 \pm 0.181$ | $3.101 \pm 0.104$ | $3.122 \pm 0.166$ |
| VAE | $1.117 \pm 0.569$ | $1.126 \pm 0.590$ | $1.433 \pm 0.622$ | $1.573 \pm 0.662$ | $0.991 \pm 0.233$ | $1.103 \pm 0.247$ | $1.341 \pm 0.305$ | $1.379 \pm 0.337$ |
| COREG | $\mathbf{0.959 \pm 0.237}$ | $\mathbf{0.974 \pm 0.295}$ | $1.137 \pm 0.306$ | $1.255 \pm 0.411$ | $\mathbf{0.937 \pm 0.209}$ | $\mathbf{0.961 \pm 0.104}$ | $1.059 \pm 0.287$ | $1.141 \pm 0.388$ |
| SSDKL | $0.992 \pm 0.221$ | $1.046 \pm 0.269$ | $1.312 \pm 0.411$ | $1.344 \pm 0.462$ | $0.959 \pm 0.210$ | $0.983 \pm 0.233$ | $1.247 \pm 0.359$ | $1.287 \pm 0.394$ |
| S$^2$MAM (ours) | $0.982 \pm 0.117$ | $1.027 \pm 0.162$ | $\mathbf{1.093 \pm 0.210}$ | $\mathbf{1.178 \pm 0.281}$ | $0.944 \pm 0.106$ | $0.970 \pm 0.146$ | $\mathbf{0.979 \pm 0.147}$ | $\mathbf{1.094 \pm 0.240}$ |

Table 7: Average Accuracy $\pm$ standard deviation (%) on synthetic classification data with fixed label percentages in each class ($r = 5\%$), uninformative variable ($p_u$) and noisy variable numbers ($p_n$). The upper and lower tables display the results of the moon data and additive data.

| Model | r = 5%, $p_u = p_n = 0$ | | r = 5%, $p_u = 10, p_n = 0$ | | r = 5%, $p_u = 0, p_n = 10$ | | r = 5%, $p_u = p_n = 10$ | |
|---|---|---|---|---|---|---|---|---|
| | Unlabeled | Test | Unlabeled | Test | Unlabeled | Test | Unlabeled | Test |
| $\ell_1$-SVM | - | $83.917 \pm 1.949$ | - | $78.631 \pm 6.737$ | - | $60.183 \pm 10.243$ | - | $55.872 \pm 8.377$ |
| SpAM | - | $84.122 \pm 1.626$ | - | $76.021 \pm 5.434$ | - | $62.307 \pm 9.590$ | - | $54.481 \pm 7.808$ |
| CSAM | - | $85.309 \pm 1.216$ | - | $77.611 \pm 4.790$ | - | $65.698 \pm 7.139$ | - | $64.714 \pm 7.211$ |
| TSpAM | - | $85.729 \pm 1.436$ | - | $79.183 \pm 4.260$ | - | $67.064 \pm 6.833$ | - | $65.592 \pm 7.148$ |
| LapSVM | $88.635 \pm 3.307$ | $86.395 \pm 2.825$ | $69.261 \pm 6.064$ | $69.670 \pm 5.941$ | $50.083 \pm 4.989$ | $51.011 \pm 5.001$ | $49.026 \pm 1.150$ | $50.000 \pm 0.000$ |
| f-FME | $89.201 \pm 1.955$ | $87.370 \pm 2.070$ | $71.631 \pm 5.255$ | $72.314 \pm 5.061$ | $53.083 \pm 5.109$ | $54.171 \pm 5.411$ | $51.026 \pm 6.598$ | $51.231 \pm 6.919$ |
| AWSSL | $\mathbf{93.171 \pm 1.801}$ | $92.395 \pm 1.977$ | $87.549 \pm 2.701$ | $87.106 \pm 2.844$ | $79.810 \pm 3.577$ | $79.901 \pm 3.650$ | $77.301 \pm 3.944$ | $77.368 \pm 4.050$ |
| RGL | $91.127 \pm 2.497$ | $90.804 \pm 2.781$ | $88.311 \pm 3.030$ | $87.914 \pm 3.152$ | $81.706 \pm 3.951$ | $81.254 \pm 4.077$ | $79.176 \pm 4.511$ | $78.679 \pm 4.989$ |
| SALE | $91.104 \pm 2.060$ | $90.799 \pm 2.135$ | $88.915 \pm 2.944$ | $88.193 \pm 3.029$ | $82.791 \pm 3.464$ | $82.199 \pm 3.891$ | $80.988 \pm 5.066$ | $80.489 \pm 5.066$ |
| SSNP | $92.720 \pm 2.184$ | $\mathbf{92.437 \pm 2.237}$ | $88.642 \pm 2.847$ | $\mathbf{88.306 \pm 3.195}$ | $81.244 \pm 4.230$ | $80.859 \pm 4.406$ | $79.287 \pm 5.026$ | $79.310 \pm 5.211$ |
| S$^2$MAM (ours) | $91.195 \pm 1.919$ | $91.877 \pm 2.207$ | $\mathbf{89.704 \pm 2.414}$ | $88.255 \pm 2.873$ | $\mathbf{83.013 \pm 4.097}$ | $\mathbf{83.454 \pm 4.388}$ | $\mathbf{81.636 \pm 4.240}$ | $\mathbf{81.950 \pm 4.713}$ |
| $\ell_1$-SVM | - | $83.914 \pm 6.410$ | - | $62.713 \pm 6.098$ | - | $62.261 \pm 6.550$ | - | $54.791 \pm 6.951$ |
| SpAM | - | $84.150 \pm 6.104$ | - | $65.091 \pm 5.917$ | - | $64.814 \pm 6.039$ | - | $54.413 \pm 6.295$ |
| CSAM | - | $86.597 \pm 5.424$ | - | $69.717 \pm 5.101$ | - | $65.178 \pm 5.255$ | - | $61.980 \pm 5.701$ |
| TSpAM | - | $86.993 \pm 5.340$ | - | $71.044 \pm 5.079$ | - | $67.340 \pm 4.959$ | - | $63.145 \pm 5.130$ |
| LapSVM | $88.814 \pm 5.398$ | $88.850 \pm 5.269$ | $59.992 \pm 5.259$ | $60.325 \pm 5.184$ | $55.630 \pm 8.213$ | $55.957 \pm 8.292$ | $55.137 \pm 8.414$ | $55.203 \pm 8.496$ |
| f-FME | $89.141 \pm 3.172$ | $89.305 \pm 3.359$ | $64.495 \pm 4.033$ | $64.611 \pm 4.208$ | $59.671 \pm 6.473$ | $59.801 \pm 6.655$ | $59.311 \pm 6.602$ | $59.407 \pm 6.659$ |
| AWSSL | $91.259 \pm 2.871$ | $90.211 \pm 3.077$ | $83.691 \pm 3.423$ | $83.950 \pm 3.519$ | $73.701 \pm 4.105$ | $73.859 \pm 4.322$ | $72.255 \pm 4.211$ | $72.370 \pm 4.428$ |
| RGL | $90.422 \pm 2.909$ | $90.026 \pm 3.477$ | $84.065 \pm 4.501$ | $84.879 \pm 4.711$ | $77.726 \pm 4.591$ | $78.041 \pm 4.510$ | $75.155 \pm 4.965$ | $75.413 \pm 4.708$ |
| SALE | $89.717 \pm 2.811$ | $90.149 \pm 2.665$ | $85.742 \pm 4.132$ | $85.971 \pm 4.018$ | $79.071 \pm 4.709$ | $79.844 \pm 4.277$ | $77.201 \pm 4.697$ | $77.891 \pm 4.431$ |
| SSNP | $\mathbf{90.492 \pm 3.059}$ | $89.871 \pm 3.218$ | $\mathbf{86.130 \pm 3.922}$ | $85.908 \pm 4.105$ | $78.250 \pm 4.294$ | $78.062 \pm 4.133$ | $77.462 \pm 4.412$ | $77.601 \pm 5.513$ |
| S$^2$MAM (ours) | $89.979 \pm 3.255$ | $\mathbf{90.309 \pm 3.409}$ | $85.517 \pm 3.481$ | $\mathbf{86.015 \pm 3.575}$ | $\mathbf{81.702 \pm 3.897}$ | $\mathbf{81.855 \pm 4.055}$ | $\mathbf{80.012 \pm 4.177}$ | $\mathbf{80.112 \pm 4.370}$ |

# C  ADDITIONAL EXPERIMENTS ON SYNTHETIC DATA

## C.1  EXPERIMENTS ON SYNTHETIC DATA

**Semi-supervised Regression:** The Friedman dataset (Friedman, 1991) owns $p^* = 5$ informative variables, and is generated by $y = 10\sin(\pi x^{(1)} x^{(2)}) + 20(x^{(3)} - 0.5)^2 + 10x^{(4)} + 5x^{(5)} + \epsilon$, where each $x^{(j)} \sim U(0, 1)$ and $\epsilon \sim \mathcal{N}(0, 1)$.

The additive data (Ravikumar et al., 2009; Chen et al., 2020; Wang et al., 2023) is generated from $y = \sum_{j=1}^{8} f^{(j)}(x^{(j)}) + \epsilon$, where $f^{(1)}(u) = -2\sin(2u)$, $f^{(2)}(u) = 8u^2$, $f^{(3)}(u) = \frac{7\sin u}{2 - \sin u}$, $f^{(4)}(u) = 6e^{-u}$, $f^{(5)}(u) = u^3 + \frac{3}{2}(u-1)^2$, $f^{(6)}(u) = 5u$, $f^{(7)}(u) = 10\sin(e^{-u/2})$, $f^{(8)}(u) = -10\widetilde{\phi}(u, \frac{1}{2}, \frac{4}{5})$. Here $\widetilde{\phi}$ stands for the normal cumulative distribution with mean of $\frac{1}{2}$ and the standard deviation of $\frac{4}{5}$. We generate $n = 200$ samples with $p^* = 8$ ($p^* = 5$) informative variables and $p_u = 92$ ($p_u = 95$) uninformative variables following $\mathcal{N}(0, 1)$ for the additive data (the Friedman data). To illustrate the impact of noisy variables, an additional $p_n = 10$ variables are designed as noisy variables following $\mathcal{N}(100, 100)$ for simplicity. The entire dataset is then split equally into training and testing sets, where only 10% or 20% of the samples retain their labels in the training set.

As shown in Table 6, S$^2$MAM enjoys competitive or even the best performance over the baselines. Under clean scenarios without corruption, some deep SSL baselines may perform slightly better,

which is understandable due to their strong approximation ability and reliance on high-quality training data. Especially under variable corruptions, our model has the smallest MSE and standard deviation, which implies that S$^2$MAM can identify most of the truly active variables by assigning the right mask. As validated in the extended experiments, these supervised baselines require larger labeled counterparts.

**Semi-supervised Classification:** Following the experimental design in (Chen et al., 2020; Wang et al., 2023), we consider the additive discriminant function $f^*(x_i) = (x_i^{(1)} - 0.5)^2 + (x_i^{(2)} - 0.5)^2 - 0.08$, where $x_i^{(j)} = (W_{ij} + U_i)/2$. $W_{ij}$ and $U_i$ are independently from $U(0,1)$ for $i = 1, \cdots, 200$, $j = 1, \cdots, 100$. The label satisfies $y_i = 0$ when $f(x_i) \leq 0$ and 1 otherwise.

To evaluate the robustness of S$^2$MAM, $p_n$ irrelevant variables are designed as noisy variables following $\mathcal{N}(100, 100)$. After equally dividing the entire dataset into training and testing sets, 5% or 10% samples for each class from the training set are randomly selected as the labeled set. As shown in Table 7, our method often enjoys better performance than the other baselines, especially in the case of noisy variables.

## C.2 ABLATION ANALYSIS

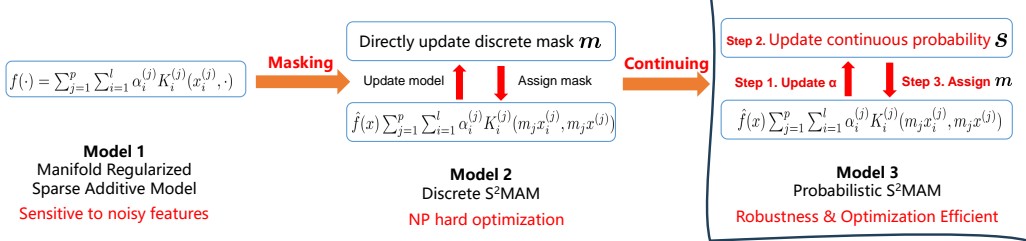

Figure 2: Connections among three models introduced in Sections 2&3. The third model with the black box is the final optimized bilevel model, probabilistic S$^2$MAM. The parameter update procedure relevant to the bilevel scheme is also illustrated.

This subsection investigated the effects of the manifold regularization, the probabilistic bilevel optimization method, and the additive modeling strategy. Firstly, we illustrate the relationship among the three models in Figure 2:

- Manifold Regularized Sparse Additive Model in Section 2.1,
- Discrete Bilevel Framework for S$^2$MAM in Section 2.2,
- Probabilistic Bilevel Framework for S$^2$MAM in Section 2.3.

We've further conducted extended ablation experiments by:

- removing the manifold regularization term ($\mathbf{f}^T \boldsymbol{L} \mathbf{f}$), named Supervised Meta Additive Model (SMAM);
- removing the upper-level problem (bilevel optimization), called Semi-supervised Additive Model (S$^2$AM);
- removing the additive strategy, named Semi-supervised Meta-based Model (S$^2$MM).

The experiments on the synthetic Friedman data and 3 real-world UCI datasets are shown below:

From the results in Tables 8 and 9, one can see that 1) SMAM has the worst performance with few labeled samples and even noisy variables. 2) Without feature corruptions, SSAM has similar performance to S$^2$MAM. Otherwise, S$^2$MM breaks down. 3) Both S$^2$MM and S$^2$MAM are robust to feature corruptions. And S$^2$MAM performs slightly better than S$^2$MM.

It implies that 1) The manifold regularization helps to use the unlabeled samples to learn better prediction functions. 2) The employed bilevel scheme for automatically assigning variable masks is vital to deal with noisy variables. 3) The additive strategy can improve the non-linear approximation

Table 8: Average MSE of extended ablation experiments on Friedman data by 1) removing the manifold regularization term; 2) removing the upper-level problem (bilevel optimization); 3) removing the additive strategy.

| Models | $r = 10\% \ \& \ p_n = 0$ | $r = 10\% \ \& \ p_n = 10$ |
|---|---|---|
| 1) SMAM | 8.319±2.740 | 10.291±3.511 |
| 2) $S^2$AM | 8.041±1.862 | 21.328±4.108 |
| 3) $S^2$MM | 7.861±2.611 | 8.913±3.811 |
| $S^2$MAM | 7.820±2.473 | 8.701±3.433 |

Table 9: Average R2 score of extended ablation experiments on UCI Datasets.

| Model | Buzz-Regression | | Boston House | | Ozone | |
|---|---|---|---|---|---|---|
| | $r = 0.1, p_n = 0$ | $r = 0.1, p_n = 10$ | $r = 0.1, p_n = 0$ | $r = 0.1, p_n = 10$ | $r = 0.1, p_n = 0$ | $r = 0.1, p_n = 10$ |
| 1) SMAM | $0.004 \pm 3.290$ | $-0.077 \pm 4.584$ | $-0.161 \pm 3.702$ | $-0.199 \pm 3.962$ | $-0.147 \pm 3.157$ | $-0.293 \pm 3.542$ |
| 2) $S^2$AM | $0.584 \pm 1.940$ | $0.553 \pm 2.514$ | $0.439 \pm 1.702$ | $0.421 \pm 1.962$ | $0.443 \pm 1.157$ | $0.397 \pm 1.472$ |
| 3) $S^2$MM | $0.684 \pm 1.390$ | $0.653 \pm 1.684$ | $0.539 \pm 0.952$ | $0.521 \pm 1.132$ | $0.543 \pm 0.357$ | $0.497 \pm 0.642$ |
| $S^2$MAM | $0.704 \pm 1.240$ | $0.673 \pm 1.534$ | $0.559 \pm 0.802$ | $0.541 \pm 0.982$ | $0.563 \pm 0.207$ | $0.517 \pm 0.492$ |

ability. SSMM fails to illustrate the prediction curve of each input variable, as the additive model is crucial for improving interpretability.

**Remark 5** *The above results also suggest that, after filtering out practical features using $S^2$MAM, the extracted data can be applied to downstream tasks under an adaptive bandwidth strategy, which can adapt to complex data distributions, such as imbalanced categories.*

## C.3 EMPIRICAL VALIDATION ON SENSITIVITY & CONVERGENCE

### C.3.1 IMPACT OF THE NUMBER OF LABELED SAMPLES

Based on the synthetic additive regression data, we first conduct a sensitivity analysis for the proposed $S^2$MAM on the size of the training set $n$, involving $l$ labeled samples and $u$ unlabeled ones.

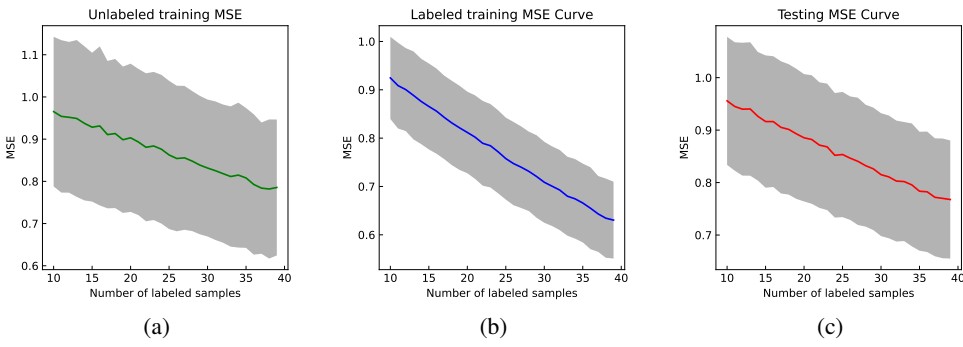

(a)  (b)  (c)

Figure 3: Average prediction MSE with standard deviation with different numbers of labeled samples. (a), (b) and (c) represent the results of the unlabeled training set, labeled training set as well as the testing set, respectively.

As shown in Figures 3, we find that larger size of labeled training data helps to improve the performance of semi-supervised model, which is consistent with our theoretical findings on the generalization error bounds, as well as some existing conclusions of statistical learning theory for supervised learning (Christmann & Zhou, 2016; Chen et al., 2020) and semi-supervised learning (Liu & Chen, 2018).

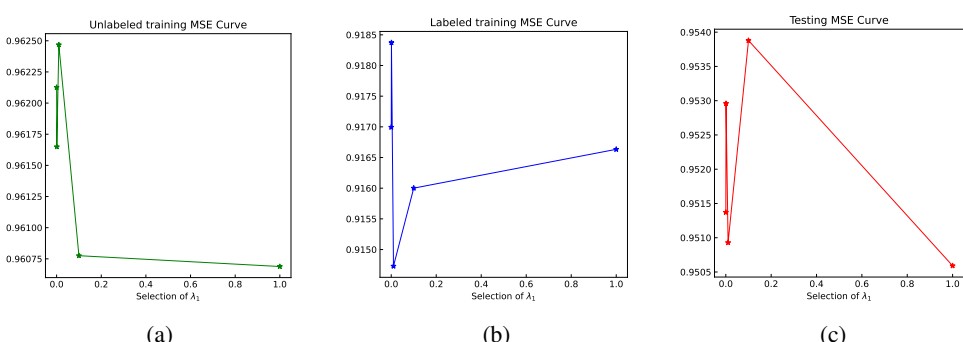

Figure 4: Average prediction MSE with different settings of $\lambda_1$. (a), (b) and (c) represent the results of the unlabeled training set, labeled training set as well as the testing set, respectively.

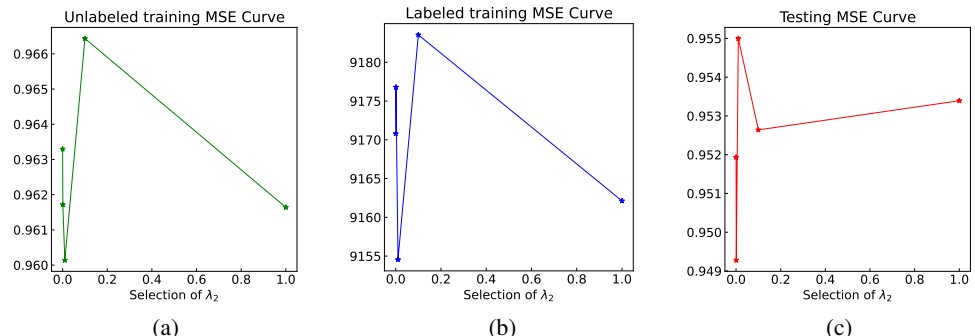

Figure 5: Average prediction MSE with different settings of $\lambda_2$. (a), (b) and (c) represent the results of the unlabeled training set, labeled training set as well as the testing set, respectively.

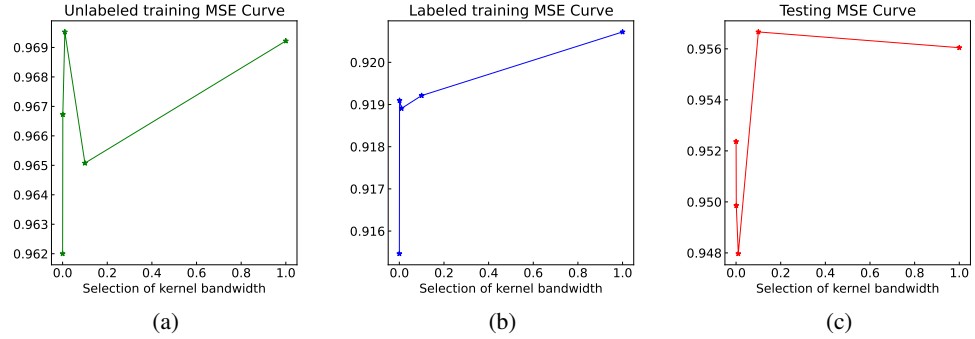

Figure 6: Average prediction MSE with different settings of Gaussian kernel bandwidth for computing similarity matrix. (a), (b) and (c) represent the results of the unlabeled training set, labeled training set as well as the testing set, respectively.

### C.3.2 IMPACT OF REGULARIZATION COEFFICIENTS AND GAUSSIAN KERNEL BANDWIDTH

Here, we focus on the impact of regularization coefficients $\lambda_1, \lambda_2$ as well as the Gaussian kernel bandwidth on the prediction performance.

Initially, we set $\lambda_1 = \lambda_2 = 10^{-3}$ as default. By changing merely a single parameter and fixing the left one, we draw the sensitive curves in Figures 4, 5, and 6. From practical experiments, we find

that too large $\lambda_1$ may introduce excessive sparsity, where truly informative variables could also be assigned relatively small weights. And $\lambda_2$ directly determines the degree of bias in the model towards unlabeled samples. The kernel bandwidth controls the similarity matrix, where values that are too small or too large can hinder the presentation of similarity between labeled and unlabeled samples. Properly selected parameters enable the model to investigate information from unlabeled data more effectively.

### C.3.3 IMPACT OF SELECTED CORE SIZE C

Now we start to analyze the sensitivity of core size C on the performance. Following the same settings as in the previous subsection, the sensitive curves for varying C, using the Friedman regression data and synthetic additive regression data, are plotted in Figure 7. The labeled rate is 5% in the training set. The average MSE and standard deviation are reported after 20 repeated experiments.

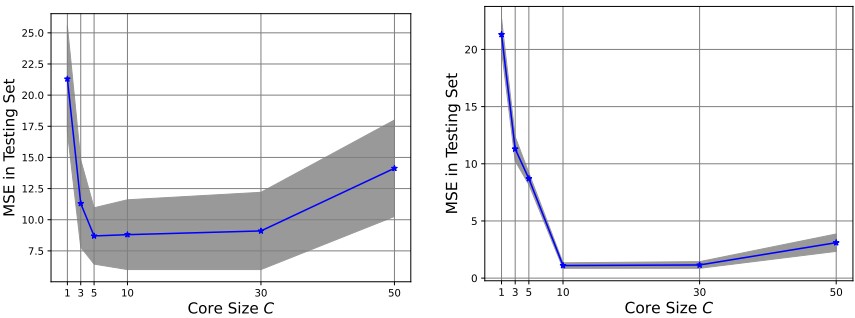

Figure 7: Average prediction MSE with different settings of parameter $C$. The left and right panels present the results on Friedman data (with 5/95/10 informative/redundant/noisy features) and synthetic additive regression data (with 8/92/10 informative/redundant/noisy features), respectively.

The empirical results show that the size of core variables $C$ is also a crucial parameter of S$^2$MAM in assigning proper masks to informative variables. In some high-dimensional real-world data without prior knowledge of instrumental variables, the binary (half-interval) searching method is suggested for setting $C$. Moreover, developing another level of problem to search for the proper $C$ automatically is also an enjoyable and meaningful direction, while the computation cost might also increase.

**Remark 6** *Practically, this binary search process was repeated individually for each baseline* $(C_1, \cdots)$ *to find the 75-quantile* $\{C_1, \cdots\}^{0.75}$ *as the choice* for sharing with all baselines requiring maximum core features, *rather than relying solely on a single model. Both the searching range and the final value are shared for all baselines. The coreset size $C$ for useful variables could be set slightly larger than the ground truth due to the sparsity constraint with $\ell$-1 regularization. Moreover, a too large $C$ may introduce unnecessary variables or even noisy variables, which could degrade the prediction performance.*

When it comes to determining the value of $C$ within the confines of the constraint set $\mathcal{C}_s$, which is defined by:
$$\mathcal{C}_s = \{\boldsymbol{s} : 0 \preceq \boldsymbol{s}_i \preceq 1, \|\boldsymbol{s}\|_1 \leq C, i = 1, 2, \cdots, p\},$$
we take the overall dimension $d$ as the starting point, setting $C$ equal to $d$. To streamline the process, in the initial stage, we identify the most suitable value for $C$, denoted as $\hat{C}$, by examining a sequence that starts at $d$ and decreases by factors of two down to 1, i.e., $[d, d/2, d/4, \ldots, 2, 1]$. Fortunately, our practical tests have shown that S$^2$MAM is capable of pinpointing the correct dimensions with high accuracy right from the outset, thereby significantly easing the burden of manually identifying key features.

### C.3.4 CONVERGENCE OF UPPER LEVEL PROBLEM

We then analyze the convergence performance of the mask learner at the upper level by plotting the curve of the upper-level objective function value with respect to iteration $t$ in Figure 8.

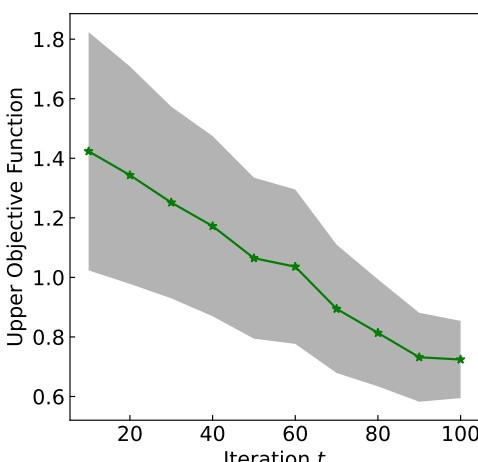

Figure 8: Convergence curve of the upper level problem of $S^2$MAM.

The synthetic additive regression data with noisy feature corruptions is used in this study. With fewer than 100 iterations, our method almost realizes convergence. However, compared to some existing SSL methods, the proposed $S^2$MAM may introduce higher computation and space complexity due to the additional computation required for the masks.

## C.4    INTERPRETABILITY AND VISUALIZATION

Additive models, including our proposed $S^2$MAM, have strong interpretability, where the component function of each input variable can be explicitly formulated and directly visualized. Here, we also give an example with our synthetic additive regression data, where the ground truth function is merely relevant to the first eight input variables:

$$Y = f^*(X) + \epsilon = \sum_{j=1}^{8} f^{(j)*}(X^{(j)}) + \epsilon, \tag{10}$$

where $f^{(1)*}(u) = -2\sin(2u)$, $f^{(2)*}(u) = 8u^2$, $f^{(3)*}(u) = \frac{7\sin u}{2-\sin u}$, $f^{(4)*}(u) = 6e^{-u}$, $f^{(5)*}(u) = u^3 + \frac{3}{2}(u-1)^2$, $f^{(6)*}(u) = 5u$, $f^{(7)*}(u) = 10\sin(e^{-u/2})$, $f^{(8)*}(u) = -10\widetilde{\phi}(u, \frac{1}{2}, \frac{4}{5}^2)$.

For simplicity, we present the prediction components of $\hat{f}^{(1)}$ and $\hat{f}^{(2)}$ as well as their ground truth $f^{(1)*}$ and $f^{(2)*}$ in Figure 9. We generate the input uniformly among $[-1, 1]$, which is further transformed into the Gram matrix of the corresponding component ($\mathbf{K}^{(1)}$ and $\mathbf{K}^{(2)}$). By multiplying with the model coefficients $\alpha^{(1)}$ and $\alpha^{(2)}$, one can directly obtain the outputs. As shown in Figure 9, the prediction results of $S^2$MAM for each input variable are close to the ground truth, which better validates the effectiveness. Additionally, the other components can also be formulated or visualized, where we omit them here for brevity.

**Remark 7** *In some relevant works, the high-dimensional observations can be regarded as the mixture of hidden information from an unknown manifold and ambient noise (Yao et al., 2024). In many realistic settings, including those with redundant, useless, or noisy variables, real-world data can also be corrupted by some noisy labels. To achieve robustness against such corruptions, a commonly considered approach is to replace the loss function with a robust one (e.g., the widely used robust Huber loss function (Wang et al., 2022b) for regression tasks). Simple modifications may help to improve the models' robustness against noisy labels. Extensions of $S^2$MAM from other perspectives are interesting directions for future study.*

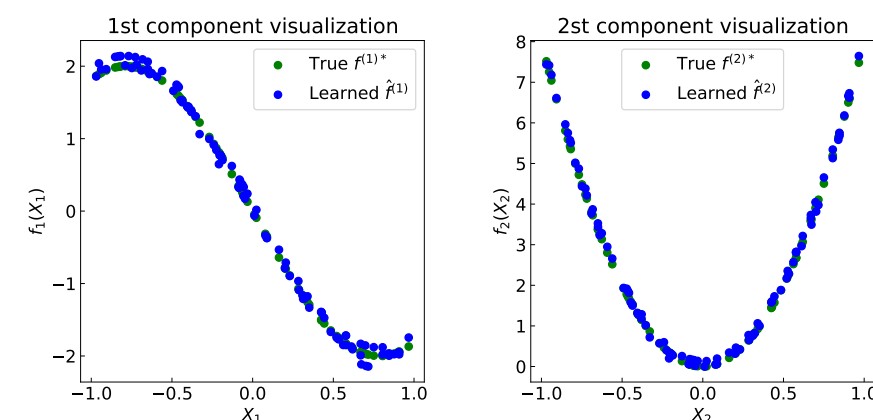

Figure 9: Visualization of the first two components. $f^*$ : ground truth; $\hat{f}$ : results predicted by S$^2$MAM.

### C.5 EXPLANATION FOR TOY EXAMPLE IN FIGURE 1

To better illustrate the negative impact of noisy variables on SSL models, we conduct semi-supervised binary classification experiments on moon data (Nie et al., 2019). For simplicity, here we generate a total of 200 samples involving 99 unlabeled points and 1 labeled point for each class. The original moon data involves two inputs ($X$ and $y$) and a single label ($-1$ or $1$). To highlight the robustness, we further add a noisy input variable ($X_n \sim \mathcal{N}(100, 100)$). Thus, the corrupted sample involves three inputs and a single output, where the $i$-th sample includes input variables $x_i = (X_i, y_i, (X_n)_i)$ and true label -1 or 1.

As shown in Figure 1, both LapSVM and our proposal, S$^2$MAM, perform well on the clean moon data without corruptions (Figure 1a). In the 2D plot in Figure 1 (b) and 3D plot in Figure 1 (d), the noisy variable directly causes negative impact on the Laplacian matrix $\mathbf{W}$, whose calculation relies on all input variables $W_{ij} = \exp\{-\|x_i - x_j\|/2\mu^2\}$ with bandwidth $\mu$.

And as present in Figure 1 (d), our proposed S$^2$MAM, with learned mask $\boldsymbol{m} = (1, 1, 0)$ assigned on inputs $(X, y, X_n)$, is robust with masked similarity $W_{ij} = \exp\{-\|\boldsymbol{m} \odot x_i - \boldsymbol{m} \odot x_j\|/\mu^2\}$, since noisy variable $X_n$ is suppressed with mask 0.

## D ADDITIONAL EXPERIMENTS ON UCI DATASET

Here we further present the additional empirical results of some baselines and S$^2$MAM on SSL learning problems. Following similar strategies for hyperparameter selection, we conduct additional experiments on 8 UCI datasets by assigning a few samples with actual labels, as well as some samples without labels, and treating the remaining points as testing sets. To better highlight the robustness of S$^2$MAM against noisy variables, the original input $X$ is corrupted by 10 noisy variables following $\mathcal{N}(100, 100)$.

Table 10 presents the experimental results on UCI datasets by varying the number of labeled training samples $l$, unlabeled training samples $u$, and noisy variables $p_n$. Since the data sizes of different classes may vary, we fixed the size of the training samples and adjusted only the labeled data size. The remaining samples are the unlabeled data sets. Because some datasets are extensive, we repeat each method 100 (or 10) times on each dataset, and list the average results as well as the standard deviation information.

Additionally, these algorithms perform better with an increasing number of labeled samples. Instead of the MSE and accuracy results, we further consider the R-squared score as the criterion to measure the performance of these methods on complex real-world data (involving a few labeled samples and unknown noise). Moreover, our proposed S$^2$MAM enjoys competitive or even better performance than these supervised or semi-supervised baselines, especially when noisy variables corrupt the data.

Table 10: Average R-squared score $\pm$ standard deviation on UCI data. The four tables from top to bottom represent the regression results under settings of $\{l = 50/20/10/50, u = 450/180/40/450, p_n = 0\}$, $\{l = 50/20/10/50, u = 450/180/40/450, p_n = 10\}$, $\{l = 100/40/20/100, u = 400/160/30/400, p_n = 0\}$ and $\{l = 100/40/20/100, u = 400/160/30/400, p_n = 10\}$, respectively.

| Model | Buzz-Regression | | Boston House | | Ozone | | SkillCraft | |
|---|---|---|---|---|---|---|---|---|
| | Unlabeled | Test | Unlabeled | Test | Unlabeled | Test | Unlabeled | Test |
| Lasso | - | -0.146 ± 12.345 | - | 0.045 ± 3.135 | - | 0.324 ± 0.822 | - | 0.467 ± 0.220 |
| SpAM | - | 0.559 ± 1.969 | - | 0.322 ± 3.693 | - | 0.340 ± 0.278 | - | 0.504 ± 0.173 |
| LapRLS | 0.631 ± 0.236 | 0.632 ± 0.240 | 0.513 ± 0.196 | 0.482 ± 0.219 | 0.557 ± 0.178 | 0.550 ± 0.192 | 0.509 ± 0.125 | 0.506 ± 0.141 |
| VAE | 0.659 ± 2.406 | 0.641 ± 2.711 | 0.525 ± 1.213 | 0.519 ± 1.301 | 0.562 ± 1.043 | 0.557 ± 1.260 | 0.512 ± 0.460 | 0.504 ± 0.475 |
| COREG | 0.691 ± 1.733 | 0.684 ± 1.851 | **0.565 ± 0.981** | 0.557 ± 1.020 | **0.573 ± 0.958** | **0.566 ± 1.030** | 0.540 ± 0.376 | 0.532 ± 0.386 |
| SSDKL | **0.717 ± 2.307** | **0.709 ± 2.434** | 0.534 ± 2.107 | 0.527 ± 2.195 | 0.569 ± 1.424 | 0.562 ± 1.472 | 0.524 ± 0.560 | 0.512 ± 0.581 |
| S²MAM (ours) | 0.712 ± 1.055 | 0.704 ± 1.240 | 0.563 ± 0.737 | **0.559 ± 0.802** | 0.568 ± 0.194 | 0.563 ± 0.207 | **0.542 ± 0.217** | **0.535 ± 0.240** |
| Lasso | - | -3.364 ± 137.251 | - | -0.358 ± 3.329 | - | -0.719 ± 4.627 | - | 0.322 ± 0.564 |
| SpAM | - | 0.364 ± 2.596 | - | -0.023 ± 0.370 | - | -0.028 ± 0.078 | - | 0.375 ± 0.438 |
| LapRLS | 0.581 ± 0.244 | 0.574 ± 0.251 | 0.473 ± 0.223 | 0.461 ± 0.247 | 0.362 ± 0.347 | 0.357 ± 0.378 | 0.485 ± 0.138 | 0.477 ± 0.146 |
| VAE | 0.573 ± 3.107 | 0.566 ± 3.211 | 0.492 ± 4.683 | 0.487 ± 4.820 | 0.485 ± 2.177 | 0.463 ± 2.305 | 0.503 ± 0.870 | 0.494 ± 0.891 |
| COREG | 0.595 ± 2.422 | 0.581 ± 2.507 | 0.511 ± 3.328 | 0.509 ± 3.511 | 0.492 ± 1.560 | 0.481 ± 1.633 | 0.517 ± 0.644 | 0.512 ± 0.671 |
| SSDKL | 0.517 ± 3.924 | 0.504 ± 3.955 | 0.502 ± 3.730 | 0.501 ± 3.795 | 0.483 ± 1.866 | 0.475 ± 1.947 | 0.511 ± 1.104 | 0.506 ± 1.193 |
| S²MAM (ours) | **0.687 ± 1.401** | **0.673 ± 1.534** | **0.549 ± 0.947** | **0.541 ± 0.982** | **0.529 ± 0.471** | **0.517 ± 0.492** | **0.523 ± 0.424** | **0.520 ± 0.439** |
| Lasso | - | 0.817 ± 0.115 | - | 0.552 ± 0.309 | - | 0.619 ± 0.331 | - | 0.524 ± 0.141 |
| SpAM | - | 0.804 ± 0.177 | - | 0.554 ± 0.335 | - | 0.631 ± 0.314 | - | 0.529 ± 0.102 |
| LapRLS | 0.841 ± 0.149 | 0.822 ± 0.205 | 0.612 ± 0.161 | 0.607 ± 0.170 | 0.650 ± 1.273 | 0.642 ± 1.311 | 0.536 ± 0.102 | 0.531 ± 0.125 |
| VAE | 0.817 ± 0.346 | 0.812 ± 0.355 | 0.631 ± 0.971 | 0.627 ± 0.990 | 0.664 ± 0.913 | 0.657 ± 0.930 | 0.542 ± 0.310 | 0.538 ± 0.318 |
| COREG | 0.881 ± 0.311 | 0.869 ± 0.320 | 0.646 ± 0.730 | **0.642 ± 0.762** | 0.673 ± 0.731 | 0.662 ± 0.760 | 0.548 ± 0.261 | 0.541 ± 0.275 |
| SSDKL | **0.911 ± 0.395** | **0.905 ± 0.418** | 0.634 ± 1.625 | 0.627 ± 1.692 | **0.679 ± 1.105** | 0.670 ± 1.231 | **0.569 ± 0.462** | **0.560 ± 0.471** |
| S²MAM (ours) | 0.901 ± 0.211 | 0.891 ± 0.180 | **0.650 ± 0.510** | 0.641 ± 0.522 | 0.677 ± 0.143 | **0.672 ± 0.159** | 0.563 ± 0.135 | 0.558 ± 0.146 |
| Lasso | - | 0.773 ± 0.433 | - | 0.526 ± 0.571 | - | -1.025 ± 3.630 | - | 0.515 ± 0.149 |
| SpAM | - | 0.747 ± 0.542 | - | 0.530 ± 0.672 | - | 0.324 ± 3.395 | - | 0.522 ± 0.191 |
| LapRLS | 0.711 ± 0.377 | 0.702 ± 0.392 | 0.522 ± 0.193 | 0.510 ± 0.217 | 0.574 ± 0.278 | 0.563 ± 0.304 | 0.504 ± 0.127 | 0.498 ± 0.132 |
| VAE | 0.742 ± 2.871 | 0.736 ± 2.951 | 0.546 ± 3.720 | 0.541 ± 2.807 | 0.591 ± 2.041 | 0.584 ± 2.259 | 0.529 ± 0.511 | 0.522 ± 0.519 |
| COREG | 0.771 ± 2.142 | 0.761 ± 2.216 | 0.565 ± 1.836 | 0.561 ± 1.862 | 0.595 ± 1.320 | 0.589 ± 1.452 | 0.538 ± 0.431 | 0.530 ± 0.438 |
| SSDKL | 0.764 ± 3.104 | 0.749 ± 3.277 | 0.537 ± 2.541 | 0.522 ± 2.679 | 0.602 ± 1.655 | 0.590 ± 1.712 | 0.546 ± 0.831 | 0.541 ± 0.840 |
| S²MAM (ours) | **0.812 ± 1.255** | **0.804 ± 1.278** | **0.621 ± 0.866** | **0.610 ± 0.879** | **0.644 ± 0.386** | **0.631 ± 0.397** | **0.558 ± 0.265** | **0.551 ± 0.271** |

Table 11: Average Accuracy $\pm$ standard deviation (%) on synthetic additive data under some extreme scenarios, i.e., label percentages in each class ($r = 5\%/50\%$) and noisy variable numbers ($p_n = 0/100$), $\{l = 100/100/100/40, u = 400/200/200/110, p_n = 0\}$ and $\{l = 100/100/100/40, u = 400/200/200/110, p_n = 10\}$, respectively.

| Model | r = 5%, $p_n$ = 0 | | r = 5%, $p_n$ = 100 | | r = 50%, $p_n$ = 0 | | r = 50%, $p_n$ = 100 | |
|---|---|---|---|---|---|---|---|---|
| | Unlabeled | Test | Unlabeled | Test | Unlabeled | Test | Unlabeled | Test |
| $\ell_1$-SVM | - | 83.914 ± 6.410 | - | 53.471 ± 8.427 | - | 93.644 ± 5.171 | - | 88.474 ± 6.209 |
| SpAM | - | 84.150 ± 6.104 | - | 51.308 ± 7.242 | - | 94.020 ± 4.255 | - | 90.201 ± 5.330 |
| CSAM | - | 86.597 ± 5.424 | - | 56.410 ± 8.781 | - | 94.973 ± 4.955 | - | 91.210 ± 5.237 |
| TSpAM | - | 86.993 ± 5.340 | - | 56.811 ± 7.570 | - | 95.031 ± 4.601 | - | **91.244 ± 5.197** |
| LapSVM | 88.814 ± 5.398 | 88.850 ± 5.269 | 37.174 ± 10.244 | 38.208 ± 10.959 | 93.899 ± 4.860 | 94.101 ± 4.571 | 41.177 ± 9.814 | 41.490 ± 9.202 |
| f-FME | 89.141 ± 3.172 | 89.305 ± 3.359 | 60.276 ± 8.427 | 59.771 ± 8.610 | 94.505 ± 2.871 | 94.893 ± 2.747 | 71.038 ± 7.979 | 70.875 ± 8.201 |
| AWSSL | **91.259 ± 2.871** | 90.211 ± 3.077 | 62.707 ± 8.660 | 62.842 ± 8.290 | 95.410 ± 3.229 | 95.601 ± 3.073 | 69.071 ± 7.759 | 69.368 ± 7.831 |
| RGL | 90.422 ± 2.909 | 90.026 ± 3.477 | 64.371 ± 8.391 | 65.011 ± 8.140 | **95.973 ± 2.417** | 96.027 ± 2.289 | 71.462 ± 7.141 | 71.511 ± 7.062 |
| SALE | 89.717 ± 2.811 | 90.149 ± 2.665 | 65.805 ± 8.106 | 65.887 ± 8.010 | 95.402 ± 2.311 | 95.427 ± 2.268 | 71.855 ± 6.947 | 71.913 ± 6.850 |
| S²MAM (ours) | 89.979 ± 3.255 | **90.309 ± 3.409** | **73.420 ± 6.177** | **73.641 ± 6.020** | 95.941 ± 2.031 | **96.147 ± 1.954** | **76.518 ± 5.326** | 76.560 ± 5.244 |

As shown in the above results, S²MAM achieves competitive or even superior performance under most settings, particularly when features are corrupted. However, when the synthetic data is clean (without noisy variables), some deep SSL methods (COREG and SSDKL) may perform better than S²MAM.

This is understandable, as the proposed S²MAM is built on kernels, and deep neural networks typically have a stronger fitting ability under clean data (Ghorbani et al., 2020; Agarwal et al., 2021; Yang et al., 2020). These deep SSL methods, along with the well-trained S²MAM, utilize all the informative input variables. While still achieving competitive prediction accuracy compared to Deep SSL methods, S²MAM further provides explainable predictions. Please refer to Figures 10 and 9, which include visual examples, where a tradeoff between interpretability and accuracy may exist (Rudin, 2019).

We further consider more settings of noisy variables, e.g., $\mathcal{N}(0, 100)$, $\mathcal{N}(50, 100)$, Student T distribution (with freedom of 2/5/10) and Chi-square noise (with freedom of 2/5/10), where the results are analogous to the setting ($X_n \in \mathcal{N}(100, 100)$). Thus, the extremely large random noise, following a $\mathcal{N}(100, 100)$ distribution, is employed throughout the entire paper for simplicity and consistency.

To make a comprehensive comparison, we further consider the data settings of 5% and 50% labeled samples, as well as $p_n = 0$ and $p_n = 100$ noisy features, on the synthetic additive data. The results are summarized in Table 11. The empirical results show that:

- At a 5% labeling rate, S²MAM is capable of assigning suitable masks, effectively utilizing the input from 95% unlabeled data to boost the model's predictive accuracy.

- At a 50% labeling rate, these supervised baselines typically yield better sparse regression estimators than S²MAM. The empirical observations are natural since the labeled data in this setting is often sufficient to identify the predictor, and supervised methods should be suggested.

### D.1 VISUALIZED LEARNING DYNAMIC PROCESS OF S²MAM

Here, we further present the visualization for the learning process of S²MAM, which shows the importance of assigning proper masks for (high-dimensional) semi-supervised modeling.

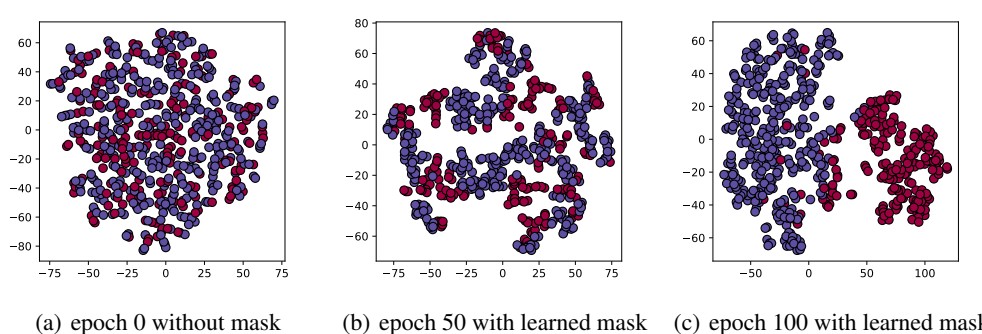

    (a) epoch 0 without mask    (b) epoch 50 with learned mask    (c) epoch 100 with learned mask

Figure 10: 2d t-SNE visualization for masked Breast Cancer data corrupted by 10 noisy features during the training process of S²MAM at epoch 0, 50, and 100, respectively. Dots with different colors represent different classes.

In Figure 10, we present the visualization of masked Breast Cancer data based on the t-SNE technique (Van der Maaten & Hinton, 2008), where the masks are updated gradually and can almost reach the ground truth after 100 epochs.

Especially under the noisy scenario, the masks sometimes exhibit fluctuations in the early stage (e.g., the first 20 epochs on the Breast Cancer data), which may be attributed to the initial settings of large step sizes and an all-one mask, as well as the high-variance gradient estimation on limited labeled data. Thanks to the decay of step size ($\eta^t = 1/\sqrt{t}$ in practice and in Theorem 1) and $\ell_{2,1}$ sparsity penalty, the learned masks tend to be stable and reach convergence among 50 to 100 epochs (as in Figure 10).

Fortunately, we observe that the coresize C could be slightly larger than the ground truth in practice. Along with the ablation studies, the $\ell_{2,1}$ penalty also helps to stabilize the training process.

## E EXTENSION TO IMAGE DATA

### E.1 PRETRAINING CNN FOR FEATURE EXTRACTION

Inspired by supervised (Su et al., 2023) and semi-supervised works (Qiu et al., 2018; Nie et al., 2019; Kang et al., 2020; Nie et al., 2021), an interesting approach for dealing with high-dimensional data, such as images, is to extract the variable vectors first.

Following (Bao et al., 2024), we first use a CNN to learn the vectors with 32 features for each image, which realizes rough dimensional reduction. However, this step may not remove those irrelevant or even noisy variables (Nie et al., 2019; 2021). Thus, it's still necessary to employ robust methods

before building semi-supervised models. Similar preprocessing methods for dimensional reduction also apply to larger (image) datasets.

### E.1.1 DETAILS ON CNN STRUCTURE AND PRETRAINING

**Structure** We use a lightweight CNN with two convolutional layers followed by three fully connected layers. This architecture is initialized randomly and explicitly trained for feature extraction.

As the dataset scales, some minor adjustments to the size of the input and hidden layers of the CNN for feature extraction are required. However, different from $S^2$MAM and RER, PLF and SemiReward are initially designed with a specific (generative) network structure to deal directly with raw images. To enable a controlled comparison with every other semi-supervised baseline, the following noise is injected into all raw inputs, rather than the noisy dimensions as in Tables 4 and 8.

**Optimization.** Only labeled raw data are used for rough training of the CNN. The CNN is optimized using the Cross-Entropy loss and the Adam optimizer with an initial learning rate of 0.001 and exponential decay. After 50 epochs, the CNN parameters are frozen.

**Extraction.** The new data shared for all baselines is extracted from the first fully connected layer of the frozen CNN.

### E.1.2 EMPIRICAL VALIDATION ON THE IMPACT OF CNN PREPROCESS ON $S^2$MAM

We conducted additional experiments based on the experimental settings in Table 12. We compared the current CNN model (CNN-1) with a more complex network model (CNN-2) that includes two convolutional layers, three local convolutional layers, and three fully connected layers (Wen et al., 2016). The changes of feature extraction bring slight differences on $S^2$MAM (Accuracy arises 0.42 with COIL-20 $r = 30\%$, $p_n = 0$)

### E.2 TIME COST ANALYSIS ON IMAGES

The following experiments are conducted for classifying the 12th and 13th objects in the COIL-20 image data.

Firstly, we conduct experiments on the clean processed feature matrix. The results are present in Table 4. Secondly, following the settings in (Bao et al., 2024), we simulate pixel-level corruption in images by manually adding five noisy variables, drawn from $\mathcal{N}(100, 100)$, to the processed 32 dimensions. The results are presented in Table 12.

Table 12: Extended experiments with average accuracy (%) $\pm$ standard deviation (SD) and training time cost (minutes) on (the 12th and 13th objects of) the corrupted COIL20 image data, which involves five manually added noisy variables (Bao et al., 2024). For simplicity, the competitors used here are all designed for SSL. Notably, $S^2$MAM and $S^2$MAM-F stand for the original strategy and the Fourier accelerated strategy (Wang et al., 2023), respectively. The upper and lower panels correspond to scenarios of original and scaled features.

| Models | LapSVM | f-FME | AWSSL | RGL | SALE | SSNP | $S^2$MAM | $S^2$MAM-F |
|---|---|---|---|---|---|---|---|---|
| Accuracy | 57.026 | 76.464 | 74.034 | 74.217 | 75.109 | 77.629 | 78.917 | **79.020**($\uparrow$) |
| SD | 7.192 | 4.106 | 3.226 | **3.011** | 4.049 | 4.310 | 3.601 | 3.473($\downarrow$) |
| Time | **0.6** | 1.5 | 2.8 | 3.0 | 2.2 | 4.1 | 2.4 | 1.7($\downarrow$) |
| Accuracy | 64.433 | 78.815 | 75.682 | 75.729 | 76.811 | 78.796 | 79.167 | **79.341**($\uparrow$) |
| SD | 7.029 | 3.972 | 3.041 | **2.870** | 3.792 | 4.067 | 3.380 | 3.209($\downarrow$) |
| Time | **0.5** | 1.3 | 2.4 | 2.7 | 2.0 | 3.7 | 2.2 | 1.4($\downarrow$) |

Moreover, inspired by (Rahimi & Recht, 2007; Wang et al., 2023), we also consider some efficient approaches for accelerating the optimization process of our $S^2$MAM, especially under the kernel hypothesis. These results empirically verify that $S^2$MAM-F ($S^2$MAM with RFF) largely retains accuracy while reducing time costs from 2.4 minutes to 1.7 minutes, confirming the practical scalability of the proposed framework.

### E.3 COMPARISONS TO DEEP SSL BASELINES WITH PIXEL CORRUPTIONS

The following comparisons are conducted on COIL images, the higher-dimensional CelebA-HQ images and AgeDB images, focusing the comparisons of $S^2$MAM and those deep SSL baselines, including AWSSL (Nie et al., 2019), SSNP (Wang et al., 2022a), RER (Bao et al., 2024), SemiReward (Li et al., 2024), PLF (Jo et al., 2024) and Flexmatch (Zhang et al., 2021). Both the regression and classification scenarios are considered, evaluating the prediction accuracy and the training time cost.

**Experimental Settings.** All new results were produced on the identical hardware platform and a similar CNN for feature extraction as in Section C.3 of the main experiments. As the dataset scales, some minor modifications to the size of the input and hidden layers of the CNN are required.

However, unlike $S^2$MAM and RER (Bao et al., 2024), PLF (Jo et al., 2024) and SemiReward (Li et al., 2024) are initially designed with a specific network structure to handle raw images as inputs directly. To enable a fair comparison with all semi-supervised baselines, the following noise (instead of the noisy dimensions as in Tables 4 and 8) is injected into all raw inputs, which is done before feature extraction via CNN.

**Noise Injection with Image Blocks.** To assess robustness and the capacity for feature selection, we employed the pixel-level corruption protocol introduced by RER (See their Figure 5(a) with 10x10 block occlusions).

The following six tables in Tables 13-18, report the average testing results and training time (minutes) on COIL, CelebA-HQ (on gender recognition), and AgeDB (age regression). Different sizes of occlusion blocks are injected, respectively, according to their image sizes.

Table 13: Classification estimation of accuracy (%) and training time on COIL ($r = 30\%, p_n = 0$, with no blocks)

| Models | AWSSL | SSNP | RER | SemiReward | Flexmatch | $S^2$MAM | $S^2$MAM-F | $S^2$MAM-N |
|---|---|---|---|---|---|---|---|---|
| Accuracy | 84.921 | 83.470 | 86.391 | 90.262 | 88.509 | 88.513 | 88.410 | 89.106 |
| SD | 0.420 | 0.430 | 0.461 | 0.390 | 0.377 | 0.439 | 0.417 | 0.381 |
| Time | 2.7 | 4.0 | 1.7 | 7.8 | 3.2 | 2.5 | 1.6 | 2.4 |

Table 14: Classification estimation of accuracy (%) and training time on COIL ($r = 30\%, p_n = 0$, with block=20x20)

| Models | AWSSL | SSNP | RER | SemiReward | Flexmatch | $S^2$MAM | $S^2$MAM-F | $S^2$MAM-N |
|---|---|---|---|---|---|---|---|---|
| Accuracy | 78.812 | 79.361 | 80.280 | 82.672 | 80.466 | 83.403 | 83.115 | 83.710 |
| SD | 0.941 | 3.439 | 3.461 | 3.890 | 3.515 | 3.429 | 3.737 | 3.371 |
| Time | 3.0 | 4.4 | 1.9 | 8.1 | 3.4 | 2.6 | 1.9 | 2.5 |

Table 15: Classification estimation of accuracy (%) and training time on CelebA-HQ ($r = 0.5\%$, no blocks)

| Models | AWSSL | SSNP | RER | SemiReward | Flexmatch | $S^2$MAM | $S^2$MAM-F | $S^2$MAM-N |
|---|---|---|---|---|---|---|---|---|
| Accuracy | 80.102 | 79.720 | 83.593 | 86.172 | 86.049 | 85.960 | 85.572 | 86.218 |
| SD | 5.423 | 5.485 | 3.622 | 2.281 | 2.325 | 2.516 | 2.410 | 2.374 |
| Time | 20.7 | 24.0 | 17.0 | 39.2 | 29.3 | 22.5 | 16.8 | 20.8 |

Based on the above results, we find that the utilized dimensionality reduction (via a pretrained CNN as in RER) combined with Fourier acceleration is both effective and accuracy-preserving for enabling $S^2$MAM to deal with relatively large-scale datasets.

In the absence of block occlusion, $S^2$MAM underperforms PLF and SemiReward, which are designed with specialized architectures that incorporate generative pseudo-label reward networks and task-specific penalties. After introducing block occlusion, however, $S^2$MAM and $S^2$MAM-F achieve marginally superior performance. Practically, when reproducing PLF and SemiReward, the noisy pixel blocks appear to compromise the similarity principle and degrade their capabilities in pseudo-label generation and filtering.

Beyond accuracy, $S^2$MAM is an interpretable additive model. The selected features can be mapped back to the frozen CNN (originally for feature extraction). Their relevance can be visualized through

Table 16: Classification estimation of accuracy (%) and training time on CelebA-HQ ($r = 0.5\%$, with block=200x200)

| Models | AWSSL | SSNP | RER | SemiReward | Flexmatch | S$^2$MAM | S$^2$MAM-F | S$^2$MAM-N |
|---|---|---|---|---|---|---|---|---|
| Accuracy | 73.102 | 72.720 | 76.593 | 76.172 | 77.416 | 78.960 | 78.572 | 79.180 |
| SD | 7.423 | 7.485 | 5.622 | 7.281 | 8.017 | 4.516 | 4.410 | 4.433 |
| Time | 21.0 | 24.1 | 17.3 | 40.2 | 34.5 | 22.8 | 17.0 | 21.2 |

Table 17: Regression estimation of root mean square error (RMSE) on AgeDB ($r = 0.5\%$, with no blocks)

| Models | COREG | SSDKL | PLF | SemiReward | S$^2$MAM | S$^2$MAM-F | S$^2$MAM-N |
|---|---|---|---|---|---|---|---|
| RMSE | 17.456 | 17.728 | 17.025 | 16.215 | 16.515 | 16.808 | 16.328 |
| SD | 2.121 | 2.305 | 1.565 | 0.650 | 0.805 | 0.870 | 0.841 |
| Time | 13.8 | 9.1 | 10.8 | 22.1 | 12.4 | 11.2 | 11.7 |

Table 18: Regression estimation of RMSE on AgeDB ($r = 0.5\%$, with block=50x50)

| Models | COREG | SSDKL | PLF | SemiReward | S$^2$MAM | S$^2$MAM-F | S$^2$MAM-N |
|---|---|---|---|---|---|---|---|
| RMSE | 19.931 | 22.713 | 19.435 | 18.302 | 17.012 | 17.317 | 16.941 |
| SD | 2.426 | 2.710 | 2.271 | 1.344 | 1.101 | 1.573 | 1.160 |
| Time | 14.0 | 9.3 | 11.0 | 22.3 | 12.6 | 11.4 | 12.2 |

Table 19: Accuracy $\pm$ standard deviation on the fast implementation experiments on COIL $r = 30\%$, $p_n = 0$, where $\downarrow 1$ implies the accuracy degrades 1 point compared to the fine-tuned model.

| | Fine-tuned | With A | With B | With C | With ALL |
|---|---|---|---|---|---|
| S$^2$MAM | $88.317 \pm 0.412$ | $86.123 \pm 0.576$ ($\downarrow$2.194) | $87.045 \pm 0.543$ ($\downarrow$1.272) | $86.789 \pm 0.567$ ($\downarrow$1.528) | $85.642 \pm 0.675$ ($\downarrow$2.675) |
| S$^2$MAM-F | $88.162 \pm 0.401$ | $86.034 \pm 0.612$ ($\downarrow$2.128) | $86.871 \pm 0.559$ ($\downarrow$1.291) | $86.543 \pm 0.584$ ($\downarrow$1.619) | $85.487 \pm 0.689$ ($\downarrow$2.675) |

Table 20: Accuracy $\pm$ standard deviation on the fast implementation experiments on COIL $r = 30\%$, $p_n = 5$.

| | Fine-tuned | With A | With B | With C | With ALL |
|---|---|---|---|---|---|
| S$^2$MAM | $78.937 \pm 3.572$ | $74.567 \pm 4.200$ ($\downarrow$4.37) | $77.234 \pm 3.950$ ($\downarrow$1.70) | $74.892 \pm 4.180$ ($\downarrow$4.05) | $71.876 \pm 4.800$ ($\downarrow$7.06) |
| S$^2$MAM-F | $79.029 \pm 3.440$ | $75.012 \pm 4.150$ ($\downarrow$4.02) | $77.345 \pm 3.920$ ($\downarrow$1.68) | $75.023 \pm 4.140$ ($\downarrow$4.01) | $72.123 \pm 4.750$ ($\downarrow$6.91) |

heat maps that highlight the corresponding pixel regions in the raw image. This yields interpretable classification (on COIL and CelebA-HQ) and regression (on AgeDB).

### E.4 TUNING-FREE TRAINING FOR FASTER IMPLEMENTATION

Parameter tuning can sometimes be time-consuming, especially in image semi-supervised classification or regression tasks. In practice, parameter selection can be accelerated and simplified efficiently by the following strategies.

**Strategy A: As for mask constraint C**, we initialize it by the first kink of the Lasso path (Chichignoud et al., 2016; Dalalyan et al., 2017), which provides a fast yet near-optimal starting point $C_0 (C_0 \ll C)$. Then, the binary selection process begins with $C_0$.

**Strategy B: For the bandwidth** $\mu$, we replicate the algorithm from (Cheng & Wu, 2022) to adaptively adjust the Laplacian kernel bandwidth.

**Strategy C: Penalty coefficients** are fixed with $\lambda_1 = \frac{u}{l+u}$, $\frac{l}{l+u}$ following similar strategies in (Ren et al., 2020; Liu et al., 2022b).

We conduct a fine-grained ablation study on COIL image data below in Tables 19 and 20 to verify the influence of the above strategies individually.

The results summarized in Tables 19 and 20 suggest that the proposal can be implemented quickly in specific tasks through these three strategies, while maintaining relatively competitive prediction and robustness compared to full fine-tuning.

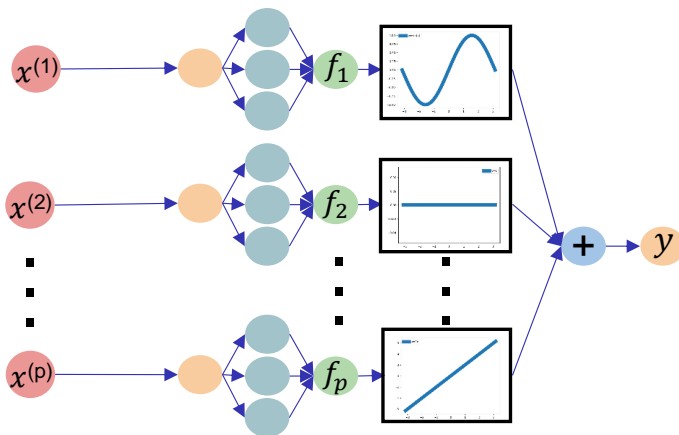

Figure 11: The lower additive model structure of neural S$^2$MAM (S$^2$MAM-F).

### E.5 EXTENSIONS TO NON-CONVEX TASKS

To better validate the bilevel strategy in non-convex scenarios, we've further proposed the neural S$^2$MAM (called S$^2$MAM-N), where each component $f_j$ is based on an individual MLP Yang et al. (2020) as shown in Figure 11. The squared loss and cross-entropy loss are utilized in practice. The extended experiments are conducted on the COIL image dataset, where the results are summarized in Tables 12-16.

Theoretically, the generalization guarantees for neural semi-supervised meta additive models are insightful and challenging, which is listed as a learning topic in future research.

## F GENERALIZATION ERROR ANALYSIS (PROOF OF THEOREM 2)

To better illustrate the proof process, we summarize the major steps and lemmas in Figure 12.

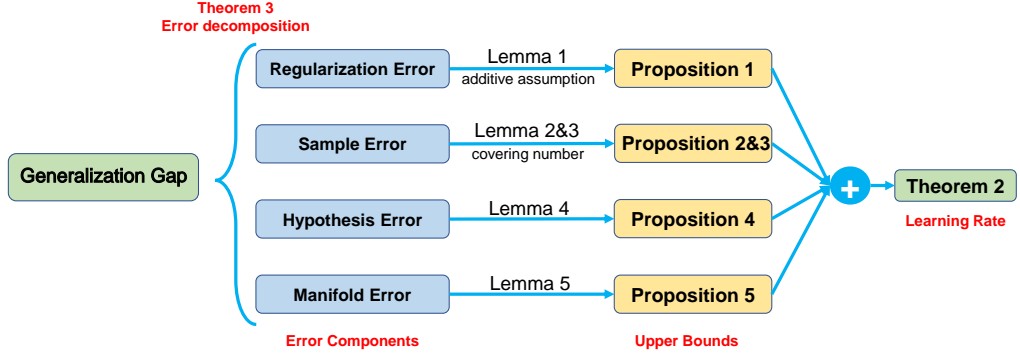

Figure 12: Sketch of the theoretical proofs for the generalization bound.

### F.1 ERROR DECOMPOSITION

Now we are in the position to recall the semi-supervised algorithm with $\ell_2$ regularizer in the additive hypothesis space

$$f_{\mathbf{z}} = \arg\min_{f \in \mathcal{H}} \left\{ \mathcal{E}_{\mathbf{z}}(f) + \lambda_1 \Omega_{\mathbf{z}}(f) + \frac{\lambda_2}{(l+u)^2} \mathbf{f}^T \mathbf{L} \mathbf{f} \right\}. \tag{11}$$

For simplicity, the semi-supervised regression task with squared loss under a kernel-based framework is considered here. Denote $\mathbf{z} = \{\mathbf{z}_l, \mathbf{z}_u\}$ as the labeled data $\mathbf{z}_l = \{x_i, y_i\}_{i=1}^l$ and unlabeled

data $\mathbf{z}_u = \{x_i\}_{i=l+1}^{l+u}$ together. Denote $\mathbf{f} = (f(x_1), \ldots, f(x_{l+u}))^T$, which involves predicting both the labeled and unlabeled data. $\lambda_1 > 0$ and $\lambda_2 > 0$ are regularization parameters. Series $\{\tau_j\}_{j=1}^p$ are weights to different input variables. For feasibility, define the Gram matrix $\mathbf{K}_i = \left(\mathbf{K}_i^{(1)}, \ldots, \mathbf{K}_i^{(p)}\right)^T \in \mathbb{R}^{(l+u)\times p}$, $\mathbf{K}^{(j)} = \left(\mathbf{K}_1^{(j)}, \ldots, \mathbf{K}_{l+u}^{(j)}\right)^T \in \mathbb{R}^{(l+u)\times(l+u)}$ with $\mathbf{K}_i^{(j)} = \left(K^{(j)}(x_1^{(j)}, x_i^{(j)}), \ldots, K^{(j)}(x_{l+u}^{(j)}, x_i^{(j)})\right)^T \in \mathbb{R}^{l+u}$ and the coefficient $\boldsymbol{\alpha} = \left(\boldsymbol{\alpha}^{(1)}, \ldots, \boldsymbol{\alpha}^{(p)}\right)^T \in \mathbb{R}^{(l+u)\times p}$ with $\boldsymbol{\alpha}^{(j)} = \left(\alpha_1^{(j)}, \ldots, \alpha_{l+u}^{(j)}\right)^T \in \mathbb{R}^{l+u}$.

The manifold regularized additive model in equation 11 can be formulated as

$$f_{\mathbf{z}} = \underset{f=\sum_{j=1}^p f^{(j)} \in \mathcal{H}}{\arg\min} \left\{\mathcal{E}_{\mathbf{z}}(f) + \lambda_1 \Omega_{\mathbf{z}}(f) + \frac{\lambda_2}{(l+u)^2}\mathbf{f}^T L \mathbf{f}\right\}, \tag{12}$$

where

$$\mathcal{E}_{\mathbf{z}}(f) = \frac{1}{l}\sum_{i=1}^l (f(x_i) - y_i)^2 = \frac{1}{l}\sum_{i=1}^l \left(\sum_{j=1}^p (\mathbf{K}_i^{(j)})^T \alpha^{(j)} - y_i\right)^2. \tag{13}$$

If the $j$-th variable is not truly informative, we expect that $\hat{\alpha}_{\mathbf{z}}^{(j)} = \left(\hat{\alpha}_{\mathbf{z},1}^{(j)}, \ldots, \hat{\alpha}_{\mathbf{z},l+u}^{(j)}\right)^T \in \mathbb{R}^{l+u}$ satisfies $\left\|\hat{\alpha}_{\mathbf{z}}^{(j)}\right\|_2 = \left(\sum_{i=1}^{l+u}\left|\hat{\alpha}_{\mathbf{z},i}^{(j)}\right|^2\right)^{(1/2)} = 0$. Inspired by this, we introduce the $\ell_{2,1}$-regularizer

$$\Omega_{\mathbf{z}}(f) = \inf\left\{\sum_{j=1}^p \tau_j \left\|\alpha^{(j)}\right\|_2 : f = \sum_{j=1}^p \sum_{i=1}^{l+u} \alpha^{(j)} K^{(j)}\left(x_i^{(j)}, \cdot\right), \alpha^{(j)} \in \mathbb{R}^{l+u}\right\} \tag{14}$$

as the penalty to address the sparsity of the output functions.

**Definition 1** *Define an operator* $L_\omega : L_{\rho_X}^2 \to L_{\rho_X}^2$ *by* $(L_\omega f)(x) = f(x)p(x) - \int_X K(x, x')f(x')d\rho_X(x')$, *with* $p(x) = \int_X K(x, x')d\rho_X(x')$. *Then we have*

$$\langle f, L_\omega f\rangle_2 = \frac{1}{2}\iint (f(x) - f(x'))^2 W(x, x')d\rho_X(x)d\rho_X(x').$$

Suppose that $\rho$ is a fixed (but unknown) probability distribution on $Z := X \times Y$. Define $f^{(j)} = (\mathbf{K}^{(j)})^T \alpha^{(j)}$. Similarly, now we introduce a regularizing function as

$$f_\lambda = \underset{f=\sum_{j=1}^p f^{(j)} \in \mathcal{H}}{\arg\min} \left\{\mathcal{E}(f) + \lambda_1 \Omega(f) + \lambda_2 \langle f, L_\omega f\rangle_2\right\}, \tag{15}$$

where

$$\mathcal{E}(f) = \int_{\mathbf{z}} (f(x) - y)^2 d\rho, \tag{16}$$

and

$$\Omega(f) = \sum_{j=1}^p \tau_j \|f^{(j)}\|_{K^{(j)}}^2. \tag{17}$$

Before presenting the error analysis, we give some basic definitions throughout this paper.

**Definition 2** *Define* $\kappa = \sup_{j,u}\left(K^{(j)}(u, u)\right)^{1/2} < \infty$. *For* $f_{\mathbf{z}}$ *defined above, there holds*

$$\|f_{\mathbf{z}}\|_K \leq \kappa \sum_{j=1}^p \sum_{i=1}^{l+u}\left|\alpha_{\mathbf{z},i}^{(j)}\right| \leq \kappa \sum_{j=1}^p \left(\sum_{i=1}^{l+u} 1^{1-\frac{1}{q}}\right)^{1-\frac{1}{q}}\left(\sum_{i=1}^{l+u}\left|\alpha_{\mathbf{z},i}^{(j)}\right|^q\right)^{\frac{1}{q}} \leq \kappa\sqrt{l+u}\sum_{j=1}^p \left\|\alpha_{\mathbf{z}}^{(j)}\right\|_2, \tag{18}$$

*where the last inequality is obtained from the Hölder inequality with positive constant* $q = 2$.

**Remark 8** *Based on the definition of $\kappa$ and $\Omega_{\mathbf{z}}(f)$, we can further obtain $\|f\|_\infty \leq \kappa\|f\|_K$ for any $f \in \mathcal{H}_K$ (Mukherjee et al., 2006; Chen et al., 2018).*

**Definition 3** *For any measurable function $f : X \to \mathbb{R}$, define the following clipping function:*

$$\pi(f) = \left\{ \begin{array}{ll} M & f(x) > M \\ -M & f(x) < -M \\ f(x) & otherwise \end{array} \right. . \tag{19}$$

**Theorem 3** *Let $f_{\mathbf{z}}$ be defined by (11) and $\pi(f)$ defined in (19). Then for $\lambda > 0$, we have*

$$\mathcal{E}\left(\pi\left(f_{\mathbf{z}}\right)\right) - \mathcal{E}\left(f_\rho\right) \leq \mathcal{D}(\lambda) + \mathcal{S}(\mathbf{s}, \lambda) + \mathcal{H}(\mathbf{s}, \lambda) + \mathcal{M}(\mathbf{s}, \lambda), \tag{20}$$

*where the regularization error, sample error, hypothesis error, and manifold error can be defined, respectively, as*

$$\mathcal{D}(\lambda) = \mathcal{E}\left(f_\lambda\right) - \mathcal{E}\left(f_\rho\right) + \lambda_1 \sum_{j=1}^{p} \tau_j \left\| f_\lambda^{(j)} \right\|_{K^{(j)}}^2 + \lambda_2 \sum_{j=1}^{p} \left\langle f_\lambda^{(j)}, L_\omega f_\lambda^{(j)} \right\rangle_2,$$

$$\mathcal{S}(\mathbf{z}, \lambda) = \mathcal{E}\left(\pi\left(f_{\mathbf{z}}\right)\right) - \mathcal{E}_{\mathbf{z}}\left(\pi\left(f_{\mathbf{z}}\right)\right) + \mathcal{E}_{\mathbf{z}}\left(f_\lambda\right) - \mathcal{E}\left(f_\lambda\right),$$

$$\mathcal{H}(\mathbf{z}, \lambda) = \mathcal{E}_{\mathbf{z}}\left(\pi\left(f_{\mathbf{z}}\right)\right) + \lambda_1 \Omega\left(f_{\mathbf{z}}\right) + \frac{\lambda_2}{(l+u)^2} \sum_{j=1}^{p} (\mathbf{f}_{\mathbf{z}}^{(j)})^T L_j \mathbf{f}_{\mathbf{z}}^{(j)}$$

$$- \left\{ \mathcal{E}_{\mathbf{z}}\left(f_\lambda\right) + \lambda_1 \sum_{j=1}^{p} \tau_j \|f_\lambda^{(j)}\|_{K^{(j)}}^2 + \frac{\lambda_2}{(l+u)^2} \sum_{j=1}^{p} (\mathbf{f}_\lambda^{(j)})^T L_j \mathbf{f}_\lambda^{(j)} \right\}, \tag{21}$$

$$\mathcal{M}(\mathbf{z}, \lambda) = \frac{\lambda_2}{(l+u)^2} \sum_{j=1}^{p} (\mathbf{f}_{\mathbf{z}}^{(j)})^T L_j \mathbf{f}_{\mathbf{z}}^{(j)} - \lambda_2 \sum_{j=1}^{p} \left\langle f_\lambda^{(j)}, L_\omega f_\lambda^{(j)} \right\rangle_2.$$

**Proof 1** *Based on the definition of $f_{\mathbf{z}}$ and $\pi(f)$, we have*

$$\mathcal{E}\left(\pi\left(f_{\mathbf{z}}\right)\right) - \mathcal{E}\left(f_{\rho}\right)$$

$$\leq \mathcal{E}\left(\pi\left(f_{\mathbf{z}}\right)\right) - \mathcal{E}\left(f_{\rho}\right) + \lambda_1 \Omega(f_{\mathbf{z}}) + \frac{\lambda_2}{(l+u)^2} \sum_{j=1}^{p} (\mathbf{f}_{\mathbf{z}}^{(j)})^T L_j \mathbf{f}_{\mathbf{z}}^{(j)}$$

$$\leq \mathcal{E}\left(\pi\left(f_{\mathbf{z}}\right)\right) - \mathcal{E}_{\mathbf{z}}(\pi\left(f_{\mathbf{z}}\right)) + \mathcal{E}_{\mathbf{z}}(\pi\left(f_{\mathbf{z}}\right)) + \lambda_1 \Omega((f_{\mathbf{z}})) + \frac{\lambda_2}{(l+u)^2} \sum_{j=1}^{p} (\mathbf{f}_{\mathbf{z}}^{(j)})^T L_j \mathbf{f}_{\mathbf{z}}^{(j)}$$

$$- \left\{ \mathcal{E}_{\mathbf{z}}\left(f_{\lambda}\right) + \lambda_1 \sum_{j=1}^{p} \tau_j \|f_{\lambda}^{(j)}\|_{K^{(j)}}^2 + \lambda_2 \sum_{j=1}^{p} \left\langle f_{\lambda}^{(j)}, L_{\omega} f_{\lambda}^{(j)} \right\rangle_2 \right\}$$

$$+ \left\{ \mathcal{E}_{\mathbf{z}}\left(f_{\lambda}\right) + \lambda_1 \sum_{j=1}^{p} \tau_j \|f_{\lambda}^{(j)}\|_{K^{(j)}}^2 + \lambda_2 \sum_{j=1}^{p} \left\langle f_{\lambda}^{(j)}, L_{\omega} f_{\lambda}^{(j)} \right\rangle_2 \right\}$$

$$- \mathcal{E}\left(f_{\lambda}\right) + \mathcal{E}\left(f_{\lambda}\right) - \mathcal{E}\left(f_{\rho}\right) + \frac{\lambda_2}{(l+u)^2} \sum_{j=1}^{p} (\mathbf{f}_{\lambda}^{(j)})^T L_j \mathbf{f}_{\lambda}^{(j)} - \frac{\lambda_2}{(l+u)^2} \sum_{j=1}^{p} (\mathbf{f}_{\lambda}^{(j)})^T L_j \mathbf{f}_{\lambda}^{(j)}$$

$$\leq \underbrace{\mathcal{E}\left(f_{\lambda}\right) - \mathcal{E}\left(f_{\rho}\right) + \lambda_1 \sum_{j=1}^{p} \tau_j \left\| f_{\lambda}^{(j)} \right\|_{K^{(j)}}^2 + \lambda_2 \sum_{j=1}^{p} \left\langle f_{\lambda}^{(j)}, L_{\omega} f_{\lambda}^{(j)} \right\rangle_2}_{\mathcal{D}(\lambda)}$$

$$+ \underbrace{\mathcal{E}\left(\pi\left(f_{\mathbf{z}}\right)\right) - \mathcal{E}_{\mathbf{z}}\left(\pi\left(f_{\mathbf{z}}\right)\right) + \mathcal{E}_{\mathbf{z}}\left(f_{\lambda}\right) - \mathcal{E}\left(f_{\lambda}\right)}_{\mathcal{S}(\mathbf{z},\lambda)}$$

$$+ \underbrace{\mathcal{E}_{\mathbf{z}}\left(\pi\left(f_{\mathbf{z}}\right)\right) + \lambda_1 \Omega\left(f_{\mathbf{z}}\right) + \frac{\lambda_2}{(l+u)^2} \sum_{j=1}^{p} (\mathbf{f}_{\mathbf{z}}^{(j)})^T L_j \mathbf{f}_{\mathbf{z}}^{(j)} - \left\{ \mathcal{E}_{\mathbf{z}}\left(f_{\lambda}\right) + \lambda_1 \sum_{j=1}^{p} \tau_j \|f_{\lambda}^{(j)}\|_{K^{(j)}}^2 + \frac{\lambda_2}{(l+u)^2} \sum_{j=1}^{p} (\mathbf{f}_{\lambda}^{(j)})^T L_j \mathbf{f}_{\lambda}^{(j)} \right\}}_{\mathcal{H}(\mathbf{z},\lambda)}$$

$$+ \underbrace{\frac{\lambda_2}{(l+u)^2} \sum_{j=1}^{p} (\mathbf{f}_{\lambda}^{(j)})^T L_j \mathbf{f}_{\lambda}^{(j)} - \lambda_2 \sum_{j=1}^{p} \left\langle f_{\lambda}^{(j)}, L_{\omega} f_{\lambda}^{(j)} \right\rangle_2}_{\mathcal{M}(\mathbf{z},\lambda)},$$

*where $\mathcal{D}(\lambda)$, $\mathcal{S}(\mathbf{z}, \lambda)$, $\mathcal{H}(\mathbf{z}, \lambda)$ and $\mathcal{M}(\mathbf{z}, \lambda)$ stand for the regularization error, sample error, hypothesis error, and manifold error, respectively. The proof is completed.*

### F.2 BOUNDING REGULARIZATION ERROR $\mathcal{D}(\lambda)$

In this section, we present the theoretical results under specific assumptions on $f_{\rho}$ for bounding the regularization error of manifold-regularized additive models. Inspired by the supervised work (Christmann & Zhou, 2016), we give some necessary assumptions and lemmas before deriving the bound under the additive space.

As defined in Section 2, we denote $\rho_{\mathcal{X}}$ as the marginal distribution with respect to $\mathcal{X}$. Here we further introduce $\rho_{\mathcal{X}^{(j)}}$ for $\mathcal{X}^{(j)}$, which is the $j$-th component of $\mathcal{X}$ (Christmann & Zhou, 2016; Chen et al., 2020). For completeness, we restate the settings in Assumption 2.

**Assumption 5** *Assume $f_{\rho} \in L_{\infty}\left(\rho_{\mathcal{X}}\right)$ and $f_{\rho} = f_{\rho}^{(1)} + f_{\rho}^{(2)} + \ldots + f_{\rho}^{(p)}$ where for some $0 < r \leq \frac{1}{2}$ and for each $j \in \{1, \ldots, p\}$, the $j$-th component function $f_{\rho}^{(j)} : \mathcal{X}^{(j)} \to \mathbb{R}$ is a mapping: $f_{\rho}^{(j)} = L_{K^{(j)}}^r \left(g_j^*\right)$ with some $g_j^* \in L_2\left(\rho_{\mathcal{X}^{(j)}}\right)$.*

The case $r = \frac{1}{2}$ of Assumption 5 means each $f_{\rho}^{(j)}$ lies in the RKHS $K^{(j)}$. Here, the operator $L_K$ is defined by

$$L_K(f)\left(X^{(1)}, \ldots, X^{(p)}\right)$$

$$= \int_{\mathcal{X}} \left( \sum_{j=1}^{p} K^{(j)}\left(X^{(j)}, X^{(j)\prime}\right) \right) f\left(X^{(1)\prime}, \ldots, X^{(p)\prime}\right) d\rho_{\mathcal{X}}\left(X^{(1)\prime}, \ldots, X^{(p)\prime}\right).$$

**Lemma 1** *(Christmann & Zhou, 2016) Let $j \in \{1, \ldots, p\}$ and $0 < r \leq \frac{1}{2}$. Assume the j-th component function $f_\rho^{(j)} = L_{K^{(j)}}^r \left(g_j^*\right)$ for some $g_j^* \in L_2\left(\rho_{\mathcal{X}^{(j)}}\right)$. Define an intermediate function $f_\lambda^{(j)}$ on $\mathcal{X}^{(j)}$ by*

$$f_\lambda^{(j)} = (L_{K^{(j)}} + \lambda I)^{-1} L_{K^{(j)}} \left(f_\rho^{(j)}\right).$$

*Then we have*

$$\left\| f_\lambda^{(j)} - f_\rho^{(j)} \right\|_{L_2\left(\rho_{X^{(j)}}\right)}^2 + \lambda \left\| f_\lambda^{(j)} \right\|_{K^{(j)}}^2 \leq \lambda^{2r} \left\| g_j^* \right\|_{L_2\left(\rho_{X^{(j)}}\right)}^2.$$

**Proposition 1** *Under Assumption 5 and $\lambda_2 = \lambda_1^{1-r}$ where $0 < r \leq 1/2$, we have*

$$\mathcal{D}(\lambda) \leq C\lambda_1^r \quad \forall 0 < \lambda_1 \leq 1,$$

*where $C$ is the constant given by*

$$C = \sum_{j=1}^p \left( L \left\| g_j^* \right\|_{L_2\left(\rho_{\mathcal{X}^{(j)}}\right)} + \left( 2\omega\kappa^2 + \max_j\{\tau_j\} \right) \left\| g_j^* \right\|_{L_2\left(\rho_{\mathcal{X}^{(j)}}\right)}^2 \right).$$

**Proof 2** *Observe that $f_\lambda^{(j)} \in H_{K^{(j)}}$ and $\sum_j^p f_\lambda^{(j)} \in H_K$. The definition of the regularization error means that*

$$\mathcal{D}(\lambda) = \mathcal{E}\left(f_\lambda\right) - \mathcal{E}\left(f_\rho\right) + \lambda_1 \sum_{j=1}^p \tau_j \left\| f_\lambda^{(j)} \right\|_{K^{(j)}}^2 + \lambda_2 \sum_{j=1}^p \left\langle f_\lambda^{(j)}, L_\omega f_\lambda^{(j)} \right\rangle_2$$

*Denote*

$$\mathcal{D}_1(\lambda) = \mathcal{E}\left(f_\lambda\right) - \mathcal{E}\left(f_\rho\right) + \lambda_1 \sum_{j=1}^p \tau_j \left\| f_\lambda^{(j)} \right\|_{K^{(j)}}^2.$$

*By Theorem 1 of (Christmann & Zhou, 2016), based on the additive hypothesis with p components in Assumption 1 and the L-Lipschitz property, we can rewrite*

$$\mathcal{E}\left(f_\lambda\right) - \mathcal{E}\left(f_\rho\right) = \mathcal{E}\left(f_\lambda^{(1)} + \cdots + f_\lambda^{(p)}\right) - \mathcal{E}\left(f_\rho^{(1)} + \cdots + f_\rho^{(p)}\right)$$

$$\leq L \sum_{j=1}^p \int_{\mathcal{X}^{(j)}} \left| f_\lambda^{(j)}\left(X^{(j)}\right) - f_\rho^{(j)}\left(X^{(j)}\right) \right| d\rho_{\mathcal{X}^{(j)}}\left(X^{(j)}\right)$$

$$\leq L \left\| f_\lambda^{(j)} - f_\rho^{(j)} \right\|_{L_2\left(\rho_{\mathcal{X}^{(j)}}\right)}.$$

*With Lemma 1, we can further derive that*

$$\left\| f_\lambda^{(j)} - f_\rho^{(j)} \right\|_{L_2\left(\rho_{\mathcal{X}^{(j)}}\right)}^2 \leq \lambda_1^{2r} \left\| g_j^* \right\|_{L_2\left(\rho_{\mathcal{X}^{(j)}}\right)}^2,$$

*and*

$$\lambda_1 \left\| f_\lambda^{(j)} \right\|_{K^{(j)}}^2 \leq \lambda_1^{2r} \left\| g_j^* \right\|_{L_2\left(\rho_{\mathcal{X}^{(j)}}\right)}^2.$$

*Thus*

$$\mathcal{D}(\lambda) \leq \mathcal{D}_1(\lambda) + \lambda_2 \sum_{j=1}^p \left\langle f_\lambda^{(j)}, L_\omega f_\lambda^{(j)} \right\rangle_2,$$

*where $0 \leq \lambda_1 \leq 1$, $0 < r \leq 1/2$ and*

$$\mathcal{D}_1(\lambda) \leq \sum_{j=1}^p \left( L\lambda_1^r \left\| g_j^* \right\|_{L_2\left(\rho_{\mathcal{X}^{(j)}}\right)} + \lambda_1^{2r} \max_j\{\tau_j\} \left\| g_j^* \right\|_{L_2\left(\rho_{\mathcal{X}^{(j)}}\right)}^2 \right)$$

$$\leq \lambda_1^r \sum_{j=1}^p \left( L \left\| g_j^* \right\|_{L_2\left(\rho_{\mathcal{X}^{(j)}}\right)} + \max_j\{\tau_j\} \left\| g_j^* \right\|_{L_2\left(\rho_{\mathcal{X}^{(j)}}\right)}^2 \right).$$

*From the fact that* $(f_\lambda(x) - f_\lambda(x'))^2 W(x,x') \leq 4\omega \|f_\lambda\|_\infty^2$ *and* $\|f_\lambda\|_\infty \leq \kappa \|f_\lambda\|_K$. *Furthermore, according to* $\langle f, L_\omega f \rangle_2 = \frac{1}{2} \iint (f(x) - f(x'))^2 W(x,x') d\rho_X(x) d\rho_X(x')$, *we have*

$$\|f_\lambda\|_K^2 \leq \sum_{j=1}^p \left\|f_\lambda^{(j)}\right\|_{K^{(j)}}^2 \leq \lambda_1^{2r-1} \sum_{j=1}^p \left\|g_j^*\right\|_{L_2\left(\rho_{\mathcal{X}^{(j)}}\right)}^2.$$

*By setting* $\lambda_2 = \lambda_1^{1-r}$ *where* $0 < r \leq 1/2$, *we can derive*

$$\lambda_2 \langle f_\lambda, L_\omega f_\lambda \rangle_2 \leq 2\omega\kappa^2 \lambda_2 \lambda_1^{2r-1} \sum_{j=1}^p \left\|g_j^*\right\|_{L_2\left(\rho_{\mathcal{X}^{(j)}}\right)}^2 \leq 2\omega\kappa^2 \lambda_1^r \sum_{j=1}^p \left\|g_j^*\right\|_{L_2\left(\rho_{\mathcal{X}^{(j)}}\right)}^2.$$

*The desired bound is derived by combining the above inequalities.*

### F.3 BOUNDING SAMPLE ERROR $\mathcal{S}(\mathbf{z}, \lambda)$

This section aims to bound the sample error term, which could be written as

$$\mathcal{S}(\mathbf{z}, \lambda) = \mathcal{S}_1(\mathbf{z}, \lambda) + \mathcal{S}_2(\mathbf{z}, \lambda),$$

where

$$\mathcal{S}_1(\mathbf{z}, \lambda) = \{\mathcal{E}(\pi(f_\mathbf{z})) - \mathcal{E}(f_\rho)\} - \{\mathcal{E}_\mathbf{z}(\pi(f_\mathbf{z})) - \mathcal{E}_\mathbf{z}(f_\rho)\} \tag{22}$$

and

$$\mathcal{S}_2(\mathbf{z}, \lambda) = \{\mathcal{E}_\mathbf{z}(f_\lambda) - \mathcal{E}_\mathbf{z}(f_\rho)\} - \{\mathcal{E}(f_\lambda) - \mathcal{E}(f_\rho)\}. \tag{23}$$

Before bounding above $\mathcal{S}_1(\mathbf{z}, \lambda)$ and $\mathcal{S}_2(\mathbf{z}, \lambda)$, we introduce the following definitions and lemmas.

**Definition 4** *Define the ball* $\mathcal{B}_r$ *associated with the function space* $\mathcal{H}_K$ *as*

$$\mathcal{B}_r = \{f \in \mathcal{H}_K : \|f\|_K \leq r\}.$$

**Definition 5** *Let* $C^v$ *be a* $v$-*times continuously differentiable function set. Then, for* $K^{(j)} \in C^\nu\left(\mathcal{X}^{(j)} \times \mathcal{X}^{(j)}\right), j \in \{1, \ldots, p\}$, *define*

$$\zeta = \begin{cases} \frac{2}{1+2v}, & v \in (0,1] \\ \frac{2}{1+v}, & v \in (1,3/2] \\ \frac{1}{v}, & v \in (3/2, \infty). \end{cases}$$

Now, we introduce the empirical covering number to measure the capacity of $\mathcal{B}_r$.

**Definition 6** *Let* $\mathcal{F}$ *be a set of measurable functions on* $\mathcal{X}$ *and* $\mathbf{x} = \{x_1, x_2, \ldots, x_n\} \subset \mathcal{X}$. *The* $\ell_2$-*empirical metric for* $f_1, f_2 \in \mathcal{F}$ *is* $d_{2,\mathbf{x}}(f_1, f_2) = \sqrt{\frac{1}{n} \sum_{i=1}^n (f_1(x_i) - f_2(x_i))^2}$. *Then the* $\ell_2$-*empirical covering number of* $\mathcal{F}$ *is defined as*

$$\mathcal{N}_2(\mathcal{F}, \epsilon) = \sup_{n \in \mathbb{N}} \sup_\mathbf{x} \mathcal{N}_{2,\mathbf{x}}(\mathcal{F}, \epsilon), \forall \epsilon > 0,$$

*where*

$$\mathcal{N}_{2,\mathbf{x}}(\mathcal{F}, \epsilon) = \inf\left\{m \in \mathbb{N} : \exists \left\{f^{(j)}\right\}_{j=1}^m \subset \mathcal{F}, s.t., \mathcal{F} \subset \bigcup_{j=1}^m \left\{f \in \mathcal{F} : d_{2,\mathbf{x}}\left(f, f^{(j)}\right) < \epsilon\right\}\right\}.$$

Indeed, the empirical covering number of $\mathcal{B}_r$ has been investigated extensively in learning theory literature (Steinwart & Christmann, 2008; Shi et al., 2011; Shi, 2013; Guo & Zhou, 2013; Chen et al., 2020).

The following concentration inequality established in (Wu et al., 2007) is used for our sample error estimation.

**Lemma 2** *(Wu et al., 2007) Let $\mathcal{G}$ be a measurable function set on $\mathcal{Z}$. Assume that there are constants $B, c, a > 0$ and $\theta \in [0,1]$ such that $\|g\|_\infty \leqslant B, \mathrm{E}g^2 \leqslant c(\mathrm{E}g)^\theta$ for each $g \in \mathcal{G}$. If for $0 < \zeta < 2, \log \mathcal{N}_2(\mathcal{G}, \epsilon) \leqslant a\epsilon^{-\zeta}, \forall \epsilon > 0$, then for any $\delta \in (0,1)$ and i.i.d observations $\{z_i\}_{i=1}^n \subset \mathcal{Z}$, there holds*

$$\mathrm{E}g - \frac{1}{n}\sum_{i=1}^n g(z_i) \leqslant \frac{1}{2}\gamma^{1-\theta}(\mathrm{E}g)^\theta + C_\zeta \gamma + 2\left(\frac{c\log(1/\delta)}{n}\right)^{\frac{1}{2-\theta}} + \frac{18B\log(1/\delta)}{n}, \forall g \in \mathcal{G}$$

*with confidence at least $1 - \delta$, where $C_\zeta$ is a constant depending only on $\zeta$ and*

$$\gamma = \max\left\{c^{\frac{2-\zeta}{4-2\theta+\zeta\theta}}(a/n)^{\frac{2}{4-2\theta+\zeta\theta}}, B^{\frac{2-\zeta}{2+\zeta}}(a/n)^{\frac{2}{2+\zeta}}\right\}.$$

**Lemma 3** *Let $\xi$ be a random variable on a probability space $\mathcal{Z}$ satisfying $|\xi(z) - E\xi| \leq M_\xi$ for some constant $M_\xi$ and variance $\sigma_\xi$. Then, for any $\delta \in (0,1)$, there holds*

$$\frac{1}{n}\sum_{i=1}^n \xi(z_i) - \mathrm{E}\xi \leq \frac{2M_\xi \log(1/\delta)}{3n} + \sqrt{\frac{2\sigma_\xi^2 \log(1/\delta)}{n}}$$

*with confidence at least $1 - \delta$.*

### F.3.1 Bounding $\mathcal{S}_1(\mathbf{z}, \lambda)$ in equation 22

**Proposition 2** *If for $0 < \zeta < 2, \log \mathcal{N}_2(\mathcal{G}, \epsilon) \leqslant a\epsilon^{-\zeta}, \forall \epsilon > 0$, then for any $\delta \in (0,1)$ and i.i.d observations $\{z_i\}_{i=1}^{l+u} \subset \mathcal{Z}$, under Assumptions 2, 3 and 4, there holds*

$$\mathcal{S}_1(\mathbf{z}, \lambda) \leqslant \frac{1}{2}\left(\mathcal{E}(\pi(f_\mathbf{z})) - \mathcal{E}(f_\rho)\right) + C_\zeta \gamma + \frac{32M^2\log(4/\delta)}{l+u} + \frac{144M^2\log(4/\delta)}{l+u}, \forall g \in \mathcal{G}$$

*with confidence at least $1 - \delta/4$, where $C_\zeta$ is a constant depending only on $\zeta$ and*

$$\gamma = \max\left\{(16M^2)^{\frac{2-\zeta}{2+\zeta}}(C_\zeta p^{1+\zeta}(4Mr)^\zeta/(l+u))^{\frac{2}{2+\zeta}}, (8M^2)^{\frac{2-\zeta}{2+\zeta}}(C_\zeta p^{1+\zeta}(4Mr)^\zeta/(l+u))^{\frac{2}{2+\zeta}}\right\}.$$

**Proof 3** *Step 1: Bounding $f_\mathbf{z}$.*

*Since $f_\mathbf{z}$ is dependent on the training sample set $\mathbf{z}$, we first need to find a function set containing $f_\mathbf{z}$.*

$$\lambda_1 \sum_{j=1}^p \tau_j\|\alpha_\mathbf{z}^{(j)}\|_2 = \lambda_1 \Omega_\mathbf{z}(f_\mathbf{z}) \leq \mathcal{E}_\mathbf{z}(f_\mathbf{z}) + \lambda_1 \Omega_\mathbf{z}(f_\mathbf{z}) + \frac{\lambda_2}{(l+u)^2}\sum_{j=1}^p (f_\mathbf{z}^{(j)})^T L_j f_\mathbf{z}^{(j)} \leq \mathcal{E}_\mathbf{z}(0) \leq M^2.$$

*Hence we have*

$$\sum_{j=1}^p \|\alpha_\mathbf{z}^{(j)}\|_2 \leq \frac{M^2}{\lambda_1 \min_j \tau_j}.$$

*Furthermore, based on the Cauchy inequality, we can obtain*

$$\|f_\mathbf{z}\|_K = \left\|\sum_{j=1}^p \sum_{i=1}^{l+n} \alpha_{\mathbf{z},i}^{(j)} K^{(j)}\left(x_i^{(j)}, \cdot\right)\right\|_K \leq \kappa \sum_{j=1}^p \sum_{i=1}^{l+u} |\alpha_{\mathbf{z},i}^{(j)}| \leq \kappa \sum_{j=1}^p \sqrt{l+u}\sqrt{\sum_{i=1}^{l+u} \|\alpha_{\mathbf{z},i}^{(j)}\|^2}$$

$$= \kappa\sqrt{l+u}\sum_{j=1}^p \|\alpha_\mathbf{z}^{(j)}\|_2.$$

*Therefore, $f_\mathbf{z}$ belongs to $B_r$ with $r = \kappa\sqrt{l+u}\sum_{j=1}^p \|\alpha_\mathbf{z}^{(j)}\|_2 \leq \frac{\kappa\sqrt{l+u}M^2}{\lambda_1 \min_j \tau_j}$.*

*Step 2: Bounding $\mathcal{S}_1(\mathbf{z}, \lambda)$ in equation 22.*

Consider the function set

$$\mathcal{G} = \left\{ g(z) = (y - \pi(f)(x))^2 - (y - f_p(x))^2 , f \in B_r, z = (x, y) \in \mathcal{Z} \right\}.$$

For any $f_1, f_2 \in \mathcal{B}_r$, we have

$$
\begin{aligned}
g(z_1) - g(z_2) &= (y - \pi(f_1)(x))^2 - (y - \pi(f_2)(x))^2 \\
&\leq |(2y - \pi(f_1)(x) - \pi(f_2)(x))(\pi(f_1)(x) - \pi(f_2)(x))| \\
&\leq 4M|\pi(f_1)(x) - \pi(f_2)(x)|.
\end{aligned}
$$

Hence for each $K^{(j)} \in C^v(x_j, x_j)$, $j = 1, \cdots, p$, we have

$$\log \mathcal{N}_2(\mathcal{G}, \epsilon) \leqslant \log \mathcal{N}_2\left(\mathcal{B}_r, \frac{\epsilon}{4M}\right) \leqslant \log \mathcal{N}_2\left(\mathcal{B}_1, \frac{\epsilon}{4Mr}\right) \leqslant C_s p^{1+\zeta}(4Mr)^\zeta \epsilon^{-\zeta}, \tag{24}$$

where $\zeta$ is defined in Definition 5, and the last inequality follows from the covering number bounds for $\mathcal{H}_{K^{(j)}}$ with $K^{(j)} \in C^v$ (see (Shi, 2013; Shi et al., 2011; Wang et al., 2021)).

Considering $0 \leq (y - \pi(f)(x))^2 \leq 4M^2$ and $0 \leq (y - f_\rho(x))^2 \leq 4M^2$, we have

$$|g(z)| \leq 8M^2, \quad |g(z) - \mathrm{E}(g)| \leq 16M^2,$$

and

$$\mathrm{E}g^2 = \int (2y - \pi(f)(x) - f_p(x))^2 (\pi(f)(x) - f_p(x))^2 \, d\rho \leqslant 16M^2 \mathrm{E}(g).$$

By applying Lemma 2 with $a = C_\zeta p^{1+\zeta}(4Mr)^\zeta$, $B = 8M^2$, $c = 16M^2$ and $\theta = 1$, $C_\zeta$ is the constant depending only on $\zeta$.

Therefore, we have the desired results for bounding $S_1$ with confidence of $1 - \delta/4$.

### F.3.2 Bounding $\mathcal{S}_2(\mathbf{z}, \lambda)$ in equation 23

**Proposition 3** *Let Assumptions 2 and 3 hold, then for any $\delta > 0$, there holds*

$$
\begin{aligned}
\mathcal{S}_2(\mathbf{z}, \lambda) &\leq \frac{2M_\xi \log(4/\delta)}{3(l + u)} + \sqrt{\frac{2Var(\xi)^2 \log(4/\delta)d}{l + u}} \\
&\leq \frac{4\left(3M + \kappa\sqrt{\frac{\mathcal{D}(\lambda)}{\lambda_1 \min_j\{\tau_j\}}}\right)^2 \log(4/\delta)}{3(l + u)} + \sqrt{\frac{2\log(4/\delta)}{l + u}} \left(3M + \kappa\sqrt{\frac{\mathcal{D}(\lambda)}{\lambda_1 \min_j\{\tau_j\}}}\right)^3 \mathcal{D}(\lambda)
\end{aligned}
$$

*with confidence at least $1 - \delta/4$.*

**Proof 4** *From the definition of $\mathcal{D}(\lambda)$ and $f_\lambda$, we can deduce that*

$$\|f_\lambda\|_K^2 \leq \frac{\mathcal{D}(\lambda)}{\lambda_1 \min_j\{\tau_j\}},$$

*and*

$$\|f_\lambda\|_\infty \leq \kappa\|f_\lambda\|_K \leq \kappa\sqrt{\frac{\mathcal{D}(\lambda)}{\lambda_1 \min_j\{\tau_j\}}}.$$

*Denote $\xi(z) = (y - f_\lambda(z))^2 - (y - f_\rho(x))^2$, we have*

$$|\xi(z)| = |2y - f_\lambda(x) - f_\rho(x)| \cdot |f_\lambda(x) - f_\rho(x)| \leq \left(3M + \kappa\sqrt{\frac{\mathcal{D}(\lambda)}{\lambda_1 \min_j\{\tau_j\}}}\right)^2 := d$$

*Then*

$$|\xi(z) - \mathrm{E}\xi| \leq 2d := M_\xi,$$

*and*

$$\mathrm{E}\xi^2 = \int |2y - f_\lambda(x) - f_\rho(x)|^2 \cdot |f_\lambda(x) - f_\rho(x)|^2 d\rho_x$$

$$\leq \left(3M + \kappa\sqrt{\frac{\mathcal{D}(\lambda)}{\lambda_1 \min_j\{\tau_j\}}}\right)^2 \|f_\lambda(x) - f_\rho(x)\|_{\rho_x}^2$$

$$\leq d(\mathcal{E}(f_\lambda) - \mathcal{E}(f_\rho))$$

$$\leq d\mathcal{D}(\lambda).$$

*Moreover,*

$$\mathrm{Var}(\xi) \leq \mathrm{E}(\xi^2) \leq d\mathcal{D}(\lambda).$$

*Applying the one side Bernstein inequality in Lemma 3 with $M_\xi = 2d$, $Var(\xi) \leq d\mathcal{D}(\lambda)$ and $d = \left(3M + \kappa\sqrt{\frac{\mathcal{D}(\lambda)}{\lambda_1 \min_j\{\tau_j\}}}\right)^2$, we get*

$$\mathcal{S}_2(\mathbf{z}, \lambda) \leq \frac{2M_\xi \log(4/\delta)}{3(l+u)} + \sqrt{\frac{2Var(\xi)^2 \log(4/\delta)d}{l+u}}$$

$$\leq \frac{4\left(3M + \kappa\sqrt{\frac{\mathcal{D}(\lambda)}{\lambda_1 \min_j\{\tau_j\}}}\right)^2 \log(4/\delta)}{3(l+u)} + \sqrt{\frac{2\log(4/\delta)}{l+u}}\left(3M + \kappa\sqrt{\frac{\mathcal{D}(\lambda)}{\lambda_1 \min_j\{\tau_j\}}}\right)^3 \mathcal{D}(\lambda)$$

*with confidence at least $1 - \delta/4$.*

The desired upper bound of $S$ is obtained by combining the above estimations for $S_1$ and $S_2$.

### F.4 BOUNDING HYPOTHESIS ERROR $\mathcal{H}(\mathbf{z}, \lambda)$

Before bounding $\mathcal{H}(\mathbf{z}, \lambda)$, we first introduce the auxiliary function

$$f_{\mathbf{z},\lambda} = \operatorname*{arg\,min}_{f=\sum_{j=1}^p f^{(j)} \in \mathcal{H}} \left\{\frac{1}{l}\sum_{i=1}^l (y_i - f(x_i))^2 + \lambda_1 \sum_{j=1}^p \tau_j\|f^{(j)}\|_{K^{(j)}}^2 + \frac{\lambda_2}{(l+u)^2}\mathbf{f}^T L\mathbf{f}\right\}, \quad (25)$$

which enjoys the representation

$$f_{\mathbf{z},\lambda}(x_i) = \sum_{j=1}^p (\mathbf{K}_i^{(j)})^T \hat{\alpha}_{\mathbf{z}}^{(j)}.$$

Here $\mathbf{K}_i^{(j)} = (K^{(j)}(x_1^{(j)}, x_i^{(j)}), K^{(j)}(x_2^{(j)}, x_i^{(j)}), \cdots, K^{(j)}(x_{l+u}^{(j)}, x_i^{(j)})) \in \mathbb{R}^{l+u}$ and $\hat{\alpha}_{\mathbf{z}}^{(j)} = (\hat{\alpha}_{\mathbf{z},1}^{(j)}, \cdots, \hat{\alpha}_{\mathbf{z},l+u}^{(j)}) \in \mathbb{R}^{l+u}$.

**Remark 9** *Based on the assumptions of boundedness (Assumption 2), we can naturally obtain that the introduced function $\mathbf{f}_{\mathbf{z},\lambda}$ in (25) has a bounded output, where the corresponding proof could be found at Lemma 4 in (Liu & Chen, 2018). By the definition of $\mathbf{f}_{\mathbf{z},\lambda}$ in (25) for $f = 0$, we have $\lambda_1\|f_{z,\lambda}\| \leq M^2$. That is, $\|\mathbf{f}_{\mathbf{z},\lambda}\|_\infty \leq M^2/\lambda_1 \leq \infty$.*

Inspired by Lemma 4 of (Chen et al., 2020) and Lemma 5 of (Wang et al., 2023), we further build the following key lemma for deriving the upper bound of hypothesis error.

**Lemma 4** *For $f_{\mathbf{z},\lambda}$ defined in (25), there exists*

$$\tau_j\|\hat{\alpha}_{\mathbf{z}}^{(j)}\|_2 \leq \frac{M + \|f_{\mathbf{z},\lambda}\|_\infty}{\lambda_1\sqrt{l}} + \frac{\lambda_2 w\|\mathbf{f}_{\mathbf{z},\lambda}^{(j)}\|_\infty}{\lambda_1(l+u)}.$$

**Proof 5** *Based the definition of $f_{\mathbf{z},\lambda}$, we can deduce that*

$$
\frac{\partial f_{\mathbf{z},\lambda}}{\partial \alpha^{(j)}} = \frac{2}{l} \sum_{i=1}^{l} (y_i - f_{\mathbf{z},\lambda}(x_i)(-(\mathbf{K}_i^{(j)})^T)) + 2\lambda_1 \tau_j (\hat{\alpha}_{\mathbf{z}}^{(j)})^T \mathbf{K}^{(j)} + \frac{\lambda_2 L_j \mathbf{f}_{\mathbf{z},\lambda}^{(j)} \mathbf{K}^{(j)}}{(l+u)^2}
$$

$$
= \frac{2}{l} \left( \underbrace{y_1 - f_{\mathbf{z},\lambda}(x_1), \cdots, y_l - f_{\mathbf{z},\lambda}(x_l)}_{l \quad Items}, \underbrace{0, \quad \cdots, 0}_{u \quad Items} \right)^T (-\mathbf{K}^{(j)}) + 2\lambda_1 \tau_j (\hat{\alpha}_{\mathbf{z}}^{(j)})^T \mathbf{K}^{(j)}
$$

$$
+ \frac{2\lambda_2 L_j \mathbf{f}_{\mathbf{z},\lambda}^{(j)} \mathbf{K}^{(j)}}{(l+u)^2},
$$

*where $\mathbf{K}^{(j)} = (K^{(j)}(x_a^{(j)}, x_b^{(j)}))_{a,b=1}^{l+u} \in \mathbb{R}^{(l+u)\times(l+u)}$.*

*When satisfying $\frac{\partial f_{\mathbf{z},\lambda}}{\partial \alpha^{(j)}} = 0$, we have*

$$
\tau_j (\hat{\alpha}_{\mathbf{z}}^{(j)})^T = \frac{1}{l\lambda_1} (y_1 - f_{\mathbf{z},\lambda}(x_1), \cdots, y_l - f_{\mathbf{z},\lambda}(x_l), 0, \cdots, 0)^T - \frac{\lambda_2 L_j \mathbf{f}_{\mathbf{z},\lambda}^{(j)}}{\lambda_1(l+u)^2}.
$$

*Then it follows for any $j \in \{1, \cdots, p\}$,*

$$
\tau_j \|\hat{\alpha}_{\mathbf{z}}^{(j)}\|_2 \leq \frac{1}{l\lambda_1} \sqrt{\sum_{i=1}^{l} (y_i - f_{\mathbf{z},\lambda}(x_i))^2} + \frac{\lambda_2}{\lambda_1(l+u)^2} \|L_j \mathbf{f}_{\mathbf{z},\lambda}^{(j)}\|_2
$$

$$
\leq \frac{M + \|f_{\mathbf{z},\lambda}\|_\infty}{\lambda_1 \sqrt{l}} + \frac{\lambda_2 w}{\lambda_1(l+u)^{3/2}} \|\mathbf{f}_{\mathbf{z},\lambda}^{(j)}\|_\infty,
$$

*where $L_j \mathbf{f}_{\mathbf{z},\lambda}^{(j)}$ could also be rewritten as the sum of $l + u$ components.*

Based on the above conclusions, we give the proof for bounding $\mathcal{H}(\mathbf{z}, \lambda)$.

**Proposition 4** *The hypothesis error $\mathcal{H}(\mathbf{z}, \lambda)$ defined in Theorem 3 could be bounded by*

$$
\mathcal{H}(\mathbf{z}, \lambda) \leq p \left( \frac{(M + \|f_{\mathbf{z},\lambda}\|_\infty)}{\sqrt{l}} + \frac{\lambda_2 w \|\mathbf{f}_{\mathbf{z},\lambda}\|_\infty}{(l+u)^{3/2}} \right),
$$

*where $f_{\mathbf{z},\lambda}$ is defined in equation 25.*

**Proof 6** *Recall the definitions of $f_{\mathbf{z}}$, $f_\lambda$ and $f_{\mathbf{z},\lambda}$, we have*

$$
\mathcal{E}_{\mathbf{z}}(f_{\mathbf{z}}) \leq \mathcal{E}_{\mathbf{z}}(f_{\mathbf{z}}) + \lambda_1 \Omega(f_{\mathbf{z}}) + \frac{\lambda_2}{(l+u)^2} \sum_{j=1}^{p} (\mathbf{f}_{\mathbf{z}}^{(j)})^T L_j \mathbf{f}_{\mathbf{z}}^{(j)}
$$

$$
\leq \mathcal{E}_{\mathbf{z}}(f_{\mathbf{z},\lambda}) + \lambda_1 \Omega(f_{\mathbf{z},\lambda}) + \frac{\lambda_2}{(l+u)^2} \sum_{j=1}^{p} (\mathbf{f}_{\mathbf{z},\lambda}^{(j)})^T L_j \mathbf{f}_{\mathbf{z},\lambda}^{(j)},
$$

*and*

$$
\mathcal{E}_{\mathbf{z}}(f_{\mathbf{z},\lambda}) + \lambda_1 \sum_{j=1}^{p} \tau_j \|f_{\mathbf{z},\lambda}^{(j)}\|_{K^{(j)}}^2 + \frac{\lambda_2}{(l+u)^2} \sum_{j=1}^{p} (\mathbf{f}_{\mathbf{z},\lambda}^{(j)})^T L_j \mathbf{f}_{\mathbf{z},\lambda}^{(j)}
$$

$$
\leq \mathcal{E}_{\mathbf{z}}(f_\lambda) + \lambda_1 \sum_{j=1}^{p} \tau_j \|f_\lambda^{(j)}\|_{K^{(j)}}^2 + \frac{\lambda_2}{(l+u)^2} \sum_{j=1}^{p} (\mathbf{f}_\lambda^{(j)})^T L_j \mathbf{f}_\lambda^{(j)}.
$$

*Then based on the definition of $\mathcal{H}(\mathbf{z}, \lambda)$, we can derive that*

$$
\begin{aligned}
\mathcal{H}(\mathbf{z}, \lambda) =& \mathcal{E}_{\mathbf{z}}\left(\pi(f_{\mathbf{z}})\right) + \lambda_1 \Omega(f_{\mathbf{z}}) + \frac{\lambda_2}{(l+u)^2} \sum_{j=1}^{p} (\mathbf{f}_{\mathbf{z}}^{(j)})^T L_j \mathbf{f}_{\mathbf{z}}^{(j)} \\
& - \left\{ \mathcal{E}_{\mathbf{z}}\left(f_{\lambda}\right) + \lambda_1 \sum_{j=1}^{p} \tau_j \|f_{\lambda}^{(j)}\|_{K^{(j)}}^2 + \frac{\lambda_2}{(l+u)^2} \sum_{j=1}^{p} (\mathbf{f}_{\lambda}^{(j)})^T L_j \mathbf{f}_{\lambda}^{(j)} \right\} \\
\leq& \mathcal{E}_{\mathbf{z}}\left(f_{\mathbf{z},\lambda}\right) + \lambda_1 \Omega(f_{\mathbf{z},\lambda}) + \frac{\lambda_2}{(l+u)^2} \sum_{j=1}^{p} (\mathbf{f}_{\mathbf{z},\lambda}^{(j)})^T L_j \mathbf{f}_{\mathbf{z},\lambda}^{(j)} \\
& - \left\{ \mathcal{E}_{\mathbf{z}}\left(f_{\mathbf{z},\lambda}\right) + \lambda_1 \sum_{j=1}^{p} \tau_j \|f_{\mathbf{z},\lambda}^{(j)}\|_{K^{(j)}}^2 + \frac{\lambda_2}{(l+u)^2} \sum_{j=1}^{p} (\mathbf{f}_{\mathbf{z},\lambda}^{(j)})^T L_j \mathbf{f}_{\mathbf{z},\lambda}^{(j)} \right\} \\
\leq& \lambda_1 \Omega(f_{\mathbf{z},\lambda}),
\end{aligned}
$$

*and based on Lemma 4, we have*

$$
\lambda_1 \Omega(f_{\mathbf{z},\lambda}) = \lambda_1 \sum_{j=1}^{p} \tau_j \|\hat{\alpha}_{\mathbf{z}}^{(j)}\|_2 \leq p \left( \frac{M + \|f_{\mathbf{z},\lambda}\|_{\infty}}{\sqrt{l}} + \frac{\lambda_2 w \max\limits_{j=1,\cdots,p} \|\mathbf{f}_{\mathbf{z},\lambda}^{(j)}\|_{\infty}}{(l+u)^{3/2}} \right).
$$

*The desired results can be obtained by combining the above inequalities.*

## F.5 BOUNDING MANIFOLD ERROR $\mathcal{M}(\mathbf{z}, \lambda)$

Recall the definition of $\mathcal{M}(\mathbf{z}, \lambda)$, we have

$$
\mathcal{M}(\mathbf{z}, \lambda) = \frac{\lambda_2}{(l+u)^2} \sum_{j=1}^{p} (\mathbf{f}_{\lambda}^{(j)})^T L_j \mathbf{f}_{\lambda}^{(j)} - \lambda_2 \sum_{j=1}^{p} \left\langle f_{\lambda}^{(j)}, L_{\omega} f_{\lambda}^{(j)} \right\rangle_2.
$$

The manifold error can be derived by bounding each of the terms with a reasonable assumption that the random variables on the similarity measure $\mathcal{W}(\cdot, x)$ lies in the additive space of RKHS. Thus, we further divide the manifold error into the following four parts:

$$
\mathcal{M}(\mathbf{z}, \lambda) = \mathcal{M}_1(\mathbf{z}, \lambda) + \mathcal{M}_2(\mathbf{z}, \lambda) + \mathcal{M}_3(\mathbf{z}, \lambda) + \mathcal{M}_4(\mathbf{z}, \lambda),
$$

where

$$
\mathcal{M}_1(\mathbf{z}, \lambda) = \frac{\lambda_2}{l+u} \sum_{i=1}^{l+u} \left( \frac{1}{l+u} \sum_{k=1}^{l+u} f_{\lambda}^2(x_k)\mathcal{W}(x_k, x_i) - \int f_{\lambda}^2(x)\mathcal{W}(x, x_i)d\rho_{\mathcal{X}}(x) \right), \quad (26)
$$

$$
\mathcal{M}_2(\mathbf{z}, \lambda) = \lambda_2 \int f_{\lambda}^2(x) \left( \frac{1}{l+u} \sum_{i=1}^{l+u} \mathcal{W}(x, x_i) - \int \mathcal{W}(x, x')d\rho_{\mathcal{X}}(x') \right) d\rho_{\mathcal{X}}(x), \quad (27)
$$

$$
\mathcal{M}_3(\mathbf{z}, \lambda) = \frac{\lambda_2}{l+u} \sum_{i=1}^{l+u} f_{\lambda}(x_i) \left( \int f_{\lambda}(x)\mathcal{W}(x, x_i)d\rho_{\mathcal{X}}(x) - \frac{1}{l+u} \sum_{k=1}^{l+u} f_{\lambda}(x)\mathcal{W}(x_k, x_i) \right), \quad (28)
$$

and

$$
\mathcal{M}_4(\mathbf{z}, \lambda) = \lambda_2 \int f_{\lambda}(x) \left( \int f_{\lambda}(x')\mathcal{W}(x, x')d\rho_{\mathcal{X}}(x') - \frac{1}{l+u} \sum_{i=1}^{l+u} f_{\lambda}(x_i)\mathcal{W}(x, x_i) \right) d\rho_{\mathcal{X}}(x). \tag{29}
$$

To analyze the above terms, we introduce the following lemma.

**Lemma 5** *(Smale & Zhou, 2007) Let $\xi$ be a random variable on $\mathcal{Z}$ in a Hilbert space $\mathcal{H}$, which satisfies $\|\xi\| \le M_\xi$. Denote $Var(\xi) = \sigma_\xi^2 = \mathrm{E}(\|\xi\|^2)$. Then for any $\delta \in (0,1)$, there holds*

$$\|\frac{1}{l+u} \sum_{i=1}^{l+u} [\xi_i - \mathrm{E}(\xi)]\| \le \frac{2M_\xi \log(\frac{2}{\delta})}{l+u} + \left( \frac{2\sigma_\xi^2 \log(\frac{2}{\delta})}{l+u} \right)^{\frac{1}{2}}$$

*with confidence $1 - \delta$.*

**Proposition 5** *For all $\delta \in (0,1)$, with confidence at least $1 - \delta$, there holds*

$$\mathcal{M}(\mathbf{z}, \lambda) \le \frac{8w\lambda_2 \kappa^2 \mathcal{D}(\lambda) \log(8/\delta)}{\lambda_1 \min_j \{\tau_j\}} (l+u)^{-\frac{1}{2}}.$$

**Proof 7** *Step 1: Bounding $\mathcal{M}_1(\mathbf{z}, \lambda)$ in equation 26. Based on the definition of $f_\lambda$, we have*

$$\|f_\lambda^2(x)\mathcal{W}(x, \cdot)\|_\infty \le w\|f_\lambda\|_\infty^2$$

*since $\|f_\lambda\|_\infty \le \kappa\|f_\lambda\|_K \le \kappa\sqrt{\frac{\mathcal{D}(\lambda)}{\lambda_1 \min_j \{\tau_j\}}}$.*

*Thus we have*

$$M_\xi \le \|f_\lambda^2(x)\mathcal{W}(x, \cdot)\|_\infty \le \frac{w\kappa^2 \mathcal{D}(\lambda)}{\lambda_1 \min_j \{\tau_j\}}.$$

*and*

$$\sigma_\xi^2 \le E[\|f_\lambda^2(x)\mathcal{W}(x, \cdot)\|_\infty^2] \le \frac{w^2 \kappa^4 \mathcal{D}^2(\lambda)}{\lambda_1^2 \min_j \{\tau_j\}^2}.$$

*Applying Lemma 5, we can derive that*

$$\mathcal{M}_1(\mathbf{z}, \lambda) \le \lambda_2 \left( \frac{2\log(\frac{8}{\delta})}{l+u} \frac{w\kappa^2 \mathcal{D}(\lambda)}{\lambda_1 \min_j \tau_j} + \sqrt{\frac{2\log(\frac{8}{\delta})}{l+u} \frac{w\kappa^2 \mathcal{D}(\lambda)}{\lambda_1 \min_j \{\tau_j\}}} \right)$$

$$\le \frac{\lambda_2 w\kappa^2 \mathcal{D}(\lambda)}{\lambda_1 \min_j \tau_j} \left( \frac{2\log(\frac{8}{\delta})}{l+u} + \sqrt{\frac{2\log(\frac{8}{\delta})}{l+u}} \right)$$

$$\le \frac{4\lambda_2 w\kappa^2 \mathcal{D}(\lambda) \log(\frac{8}{\delta})}{\sqrt{l+u}\lambda_1 \min_j \tau_j}$$

*with confidence of $1 - \delta/4$.*

*Step 2: Bounding $\mathcal{M}_2(\mathbf{z}, \lambda)$ in equation 27. Note that $\|\mathcal{W}(\cdot, x)\| \le w$, $E[\|\mathcal{W}(\cdot, x)\|^2] \le w^2$.*

*Then, with confidence of $1 - \frac{\delta}{4}$, we have*

$$\mathcal{M}_2(\mathbf{z}, \lambda) \le \lambda_2 \int f_\lambda^2(x)w \left( \frac{2\log(8/\delta)}{l+u} + \sqrt{\frac{2\log(8/\delta)}{l+u}} \right) d\rho_{\mathcal{X}}(x)$$

$$\le \lambda_2 w \left( \frac{2\log(8/\delta)}{l+u} + \sqrt{\frac{2\log(8/\delta)}{l+u}} \right) \int f_\lambda^2(x) d\rho_{\mathcal{X}}(x)$$

$$\le \lambda_2 w \frac{4\log(8/\delta)}{\sqrt{l+u}} \frac{w\kappa^2 \mathcal{D}(\lambda)}{\lambda_1 \min_j \tau_j}$$

$$\le \frac{4\lambda_2 w\kappa^2 \mathcal{D}(\lambda)}{\sqrt{l+u}\lambda_1 \min_j \tau_j} \log(\frac{8}{\delta}).$$

*Step 3: Bounding $\mathcal{M}_3(\mathbf{z}, \lambda)$ in equation 28. It is easy to deduce that*

$$\|f_\lambda(x)\mathcal{W}(\cdot, x)\| \le w\kappa\sqrt{\frac{\mathcal{D}(\lambda)}{\lambda_1 \min_j \tau_j}},$$

*and*

$$E[\|f_\lambda(x)\mathcal{W}(\cdot,x)\|^2] \leq w^2\kappa^2 \frac{\mathcal{D}(\lambda)}{\lambda_1 \min_j \tau_j}.$$

*Then, with confidence of $1 - \frac{\delta}{4}$, we can derive that*

$$\mathcal{M}_3(\mathbf{z},\lambda) = \frac{\lambda_2}{l+u} \sum_{i=1}^{l+u} f_\lambda(x_i) \left( \int f_\lambda(x)\mathcal{W}(x,x_i)d\rho_\mathcal{X}(x) - \frac{1}{l+u} \sum_{k=1}^{l+u} f_\lambda(x)\mathcal{W}(x_k,x_i) \right)$$

$$\leq \frac{\lambda_2}{l+u} \sum_{i=1}^{l+u} f_\lambda(x_i) w\kappa \sqrt{\frac{\mathcal{D}(\lambda)}{\lambda_1 \min_j \tau_j}} \left( \frac{2\log(\frac{8}{\delta})}{l+u} + \sqrt{\frac{2\log(\frac{8}{\delta})}{l+u}} \right)$$

$$\leq \lambda_2 w\kappa^2 \frac{\mathcal{D}(\lambda)}{\lambda_1 \min_j \tau_j} \frac{4\log(\frac{8}{\delta})}{\sqrt{l+u}}$$

$$\leq \frac{4\lambda_2 w\kappa^2 \mathcal{D}(\lambda)}{\sqrt{l+u}\lambda_1 \min_j \tau_j} \log(\frac{8}{\delta}).$$

*Step 4: Bounding $\mathcal{M}_4(\mathbf{z},\lambda)$ in equation 29. Finally, we can deduce that with confidence of $1 - \delta/4$,*

$$\mathcal{M}_4(\mathbf{z},\lambda) \leq \lambda_2 \int f_\lambda(x) w\kappa \sqrt{\frac{\mathcal{D}(\lambda)}{\lambda_1 \min_j \tau_j}} \left( \frac{2\log(\frac{8}{\delta})}{l+u} + \sqrt{\frac{2\log(\frac{8}{\delta})}{l+u}} \right) d\rho_\mathcal{X}(x)$$

$$\leq \lambda_2 w\kappa \sqrt{\frac{\mathcal{D}(\lambda)}{\lambda_1 \min_j \tau_j}} 2\frac{2\log(\frac{8}{\delta})}{\sqrt{l+u}} \int f_\lambda(x)d\rho_\mathcal{X}(x)$$

$$\leq \frac{4\lambda_2 w\kappa^2 \mathcal{D}(\lambda)}{\sqrt{l+u}\lambda_1 \min_j \tau_j} \log(\frac{8}{\delta}).$$

*The desired result follows by combining the above estimations.*

### F.6 PROOF OF THEOREM 2

Then we summarize the above conclusions and analyze the learning rate under mild assumptions.

**Proposition 6** *Let Assumptions 2-4 be true. For any $\delta \in (0, 1/2)$, with confidence $1 - 2\delta$ there holds*

$$\begin{aligned}
&\mathcal{E}(\pi(f_\mathbf{z})) - \mathcal{E}(f_\rho) \\
\leq\ & \mathcal{D}(\lambda) + \mathcal{S}(\mathbf{z},\lambda) + \mathcal{H}(\mathbf{z},\lambda) + \mathcal{M}(\mathbf{z},\lambda) \\
\leq\ & C_r\lambda_1^r + \frac{1}{2}\left(\mathcal{E}(\pi(f_\mathbf{z})) - \mathcal{E}(f_\rho)\right) + C_\zeta\gamma + \frac{32M^2\log(4/\delta)}{l+u} + \frac{144M^2\log(4/\delta)}{l+u} \\
& + \frac{4\left(3M + \kappa\sqrt{\frac{\mathcal{D}(\lambda)}{\lambda_1 \min_j\{\tau_j\}}}\right)^2 \log(4/\delta)}{3(l+u)} + \sqrt{\frac{2\log(4/\delta)}{l+u}} \left(3M + \kappa\sqrt{\frac{\mathcal{D}(\lambda)}{\lambda_1 \min_j\{\tau_j\}}}\right)^3 \mathcal{D}(\lambda) \\
& + p\left(\frac{(M + \|f_{\mathbf{z},\lambda}\|_\infty)}{\sqrt{l}} + \frac{\lambda_2 w\|\mathbf{f}_{\mathbf{z},\lambda}^{(j)}\|_\infty}{(l+u)^{3/2}}\right) + \frac{16\lambda_2 w\kappa^2\mathcal{D}(\lambda)}{\sqrt{l+u}\lambda_1 \min_j \tau_j} \log(\frac{8}{\delta}),
\end{aligned}$$

*where*

$$C_r = \sum_{j=1}^p \left( L\left\|g_j^*\right\|_{L_2\left(\rho_{\mathcal{X}^{(j)}}\right)} + \left(2\omega\kappa^2 + \max_j\{\tau_j\}\right)\left\|g_j^*\right\|_{L_2\left(\rho_{\mathcal{X}^{(j)}}\right)}^2 \right),$$

$$\gamma = \max\left\{ (16M^2)^{\frac{2-\zeta}{2+\zeta}} (C_\zeta p^{1+\zeta}(4Mr)^\zeta/(l+u))^{\frac{2}{2+\zeta}}, (8M^2)^{\frac{2-\zeta}{2+\zeta}} (C_\zeta p^{1+\zeta}(4Mr)^\zeta/(l+u))^{\frac{2}{2+\zeta}} \right\},$$

*$C_\zeta$ is a constant, $0 < r \leq 1/2$, $0 < \zeta < 2$ and $f_{\mathbf{z},\lambda}$ is defined in equation 25.*

**Proof 8** *The above results can be obtained by directly combining the results of Theorem 3 and Propositions 1-5.*

Now, we present the proof of Theorem 22.

**Proof 9** *Let $\lambda_1 = (l+u)^{-\Delta}$ and $\lambda_2 = \lambda_1^{1-r} = (l+u)^{-\Delta(1-r)}$, where $0 < r \leq 1/2$. According to Proposition 6 and the properties of $\mathcal{D}(\lambda)$ and $\mathbf{f}_{\mathbf{z},\lambda}$, we have*

$$
\mathcal{E}\left(\pi\left(f_{\mathbf{z}}\right)\right) - \mathcal{E}\left(f_\rho\right)
$$

$$
\leq C_1(l+u)^{-\Delta r} + C_2(l+u)^{-2/(2+\xi)} + C_3 \log(4/\delta)(l+u)^{-1}
$$

$$
+ C_4 \log(4/\delta)(l+u)^{\Delta(1-r)-1} + C_5 \sqrt{\log(4/\delta)}(l+u)^{-\Delta(5r/2-3/2)-1/2} + C_6 l^{-1/2}
$$

$$
+ C_7(l+u)^{\Delta r - 3/2} + C_8 \log(8/\delta)(l+u)^{-1/2}
$$

$$
\leq C_9 \log(8/\delta)\left((l+u)^{-\Delta r} + (l+u)^{-2/(2+\xi)} + (l+u)^{-1} + (l+u)^{\Delta(1-r)-1}\right.
$$

$$
\left. + (l+u)^{-\Delta(5r/2-3/2)-1/2} + (l+u)^{\Delta r - 3/2} + (l+u)^{-1/2}\right) + l^{-1/2})
$$

$$
\leq C_{10} \log(8/\delta)\left((l+u)^{-\Theta} + l^{-1/2}\right),
$$

*where*

$$
\Theta = \min\{\Delta r, 2/(2+\zeta), 1, 1+\Delta(r-1), \Delta(5r/2-3/2)+1/2, 3/2-\Delta r, 1/2\}
$$

$$
= \min\{\Delta r, 1+\Delta(r-1), \Delta(5r/2-3/2)+1/2, 3/2-\Delta r, 1/2\},
$$

*and $\Delta > 0$, $0 < r \leq 1/2$, $0 < \zeta < 2$. And $C_1, \cdots, C_{10}$ are positive constants independently of $l, u, \delta$ and $r$.*

*With $\Delta = 1$ and $r = 1/2$, the following holds*

$$
\mathcal{E}\left(\pi\left(f_{\mathbf{z}}\right)\right) - \mathcal{E}\left(f_\rho\right) \leq \max\left\{\mathcal{O}\left((l+u)^{-1/4}\right), \mathcal{O}\left(l^{-1/2}\right)\right\}.
$$

*This completes the proof.*

## G    CONVERGENCE ANALYSIS (PROOF OF THEOREM 1)

As described in the main paper, the masks on all features are learned at the upper level of $\text{S}^2\text{MAM}$, where a project operation is employed to limit informative variables. Thus, we mainly focus on the corresponding convergence performance of the upper level of $\text{S}^2\text{MAM}$.

Notice that the update rule for variable $s$ in practice can be formulated by

$$
\boldsymbol{s}^{t+1} = \mathcal{P}_\mathcal{C}\left(\boldsymbol{s}^t - \eta^t \mathcal{L}_\mathcal{B}\left(\boldsymbol{\alpha}^*(\boldsymbol{m})\right) \nabla_{\boldsymbol{s}} \ln p\left(\boldsymbol{m} \mid \boldsymbol{s}^t\right)\right), \tag{30}
$$

where $\mathcal{L}_\mathcal{B}$ is the loss on selected sample batch $\mathcal{B}$.

Furthermore, denote the update rules with stochastic and deterministic gradient mappings as

$$
\boldsymbol{s}^{t+1} = \boldsymbol{s}^t - \eta^t \hat{\mathcal{G}}^t = \mathcal{P}_\mathcal{C}\left(\boldsymbol{s}^t - \eta^t \mathcal{L}_\mathcal{B}\left(\boldsymbol{\alpha}^*(\boldsymbol{m})\right) \nabla_{\boldsymbol{s}} \ln p\left(\boldsymbol{m} \mid \boldsymbol{s}^t\right)\right),
$$

$$
\boldsymbol{s}^{t+1} = \boldsymbol{s}^t - \eta^t \mathcal{G}^t = \mathcal{P}_\mathcal{C}\left(\boldsymbol{s}^t - \eta^t \nabla_s \Phi\left(\boldsymbol{s}^t\right)\right).
$$

That is to say

$$
\hat{\mathcal{G}}^t = \frac{1}{\eta^t}\left(\boldsymbol{s}^t - \mathcal{P}_\mathcal{C}\left(\boldsymbol{s}^t - \eta^t \mathcal{L}_\mathcal{B}\left(\boldsymbol{\alpha}^*(\boldsymbol{m})\right) \nabla_{\boldsymbol{s}} \ln p\left(\boldsymbol{m} \mid \boldsymbol{s}^t\right)\right)\right) = \frac{1}{\eta^t}\left(\boldsymbol{s}^t - \boldsymbol{s}^{t+1}\right),
$$

$$
\mathcal{G}^t = \frac{1}{\eta^t}\left(\boldsymbol{s}^t - \mathcal{P}_\mathcal{C}\left(\boldsymbol{s}^t - \eta^t \nabla_s \Phi\left(\boldsymbol{s}^t\right)\right)\right).
$$

Firstly, we recall some necessary assumptions and definitions for projection operation, which have been used in existing works on algorithmic convergence analysis on projection optimization for single-level problems (Bauschke et al., 2012) and bilevel ones (Pedregosa, 2016).

Inspired by some research on bilevel optimization problems (Pedregosa, 2016; Shu et al., 2023; Zhao et al., 2023) with mini-batch settings, this paper adopts the independently and identically distributed (i.i.d.) random variables induced by the mini-batch. Notice that $\xi^{(t)} = \mathcal{L}_\mathcal{B}\left(\boldsymbol{\alpha}^*(\boldsymbol{m})\right)\nabla_{\boldsymbol{s}}\ln p(\boldsymbol{m} \mid s^t) - \nabla_{\boldsymbol{s}}\Phi(s^t)$ for $t \in [1, 2, \cdots, T]$ are i.i.d random variables with finite variance $\sigma^2$, since the mini-batch are drawn i.i.d with a finite number of samples. Furthermore, $\mathbb{E}\left[\xi^{(t)}\right] = 0$ since samples are drawn uniformly at random.

**Lemma 6** *Given a compact convex set $\mathcal{C} \subset \mathbb{R}^d$ and let $\mathcal{P}_\mathcal{C}(\cdot)$ be the projection operator on $\mathcal{C}$, then for any $\boldsymbol{u} \in \mathbb{R}^d$ and $\boldsymbol{v} \in \mathbb{R}^d$, we have*

$$\|\mathcal{P}_\mathcal{C}(\boldsymbol{u}) - \mathcal{P}_\mathcal{C}(\boldsymbol{v})\|^2 \leq (\boldsymbol{u} - \boldsymbol{v})^\top (\mathcal{P}_\mathcal{C}(\boldsymbol{u}) - \mathcal{P}_\mathcal{C}(\boldsymbol{v}))$$

**Lemma 7** *Given a compact convex set $\mathcal{C} \subset \mathbb{R}^d$ and let $\mathcal{P}_\mathcal{C}(\cdot)$ be the projection operator on $\mathcal{C}$, then for any $\boldsymbol{c} \in \mathcal{C}$ and $\boldsymbol{u} \in \mathbb{R}^d, \boldsymbol{v} \in \mathbb{R}^d$, we have*

$$\|\mathcal{P}_\mathcal{C}(\boldsymbol{c} + \boldsymbol{u}) - \mathcal{P}_\mathcal{C}(\boldsymbol{c} + \boldsymbol{v})\| \leq \|\boldsymbol{u} - \boldsymbol{v}\|.$$

**Remark 10** *Considering $\boldsymbol{c} = s^t$, $\boldsymbol{u} = \eta^t \mathcal{L}_\mathcal{B}\left(\boldsymbol{\alpha}^*(\boldsymbol{m})\right)\nabla_{\boldsymbol{s}}\ln p(\boldsymbol{m} \mid s^t)$ and $\boldsymbol{v} = \nabla_{\boldsymbol{s}}\Phi(s^t)$, we can easily obtain that*

$$\|\hat{\mathcal{G}}^t - \mathcal{G}^t\| \leq \|\mathcal{L}_\mathcal{B}\left(\boldsymbol{\alpha}^*(\boldsymbol{m})\right)\nabla_{\boldsymbol{s}}\ln p(\boldsymbol{m} \mid s^t) - \nabla_{\boldsymbol{s}}\Phi(s^t)\| := \|\xi^{(t)}\|.$$

In the following, we present the corresponding proof for Theorem 1.

**Proof 10** *Inspired from Theorem 2 in (Pedregosa, 2016), the following holds with Lemma 6 by setting $\boldsymbol{u} = s^t$ and $\boldsymbol{v} = s^t - \eta^t g^t$,*

$$\|s^t - s^{t+1}\|^2 \leq \eta^t (\mathcal{L}_\mathcal{B}\left(\boldsymbol{\alpha}^*(\boldsymbol{m})\right)\nabla_{\boldsymbol{s}}\ln p(\boldsymbol{m} \mid s^t))^T (s^t - s^{t+1}) = \eta^t (\mathcal{L}_\mathcal{B}\left(\boldsymbol{\alpha}^*(\boldsymbol{m})\right)\nabla_{\boldsymbol{s}}\ln p(\boldsymbol{m} \mid s^t))^T \hat{\mathcal{G}}^t.$$

*Thus we have*

$$\|\hat{\mathcal{G}}^t\|^2 \leq \left\langle \mathcal{L}_\mathcal{B}\left(\boldsymbol{\alpha}^*(\boldsymbol{m})\right)\nabla_{\boldsymbol{s}}\ln p(\boldsymbol{m} \mid s^t), \hat{\mathcal{G}}^t \right\rangle.$$

*Recall the random variable $\xi^{(t)} = \mathcal{L}_\mathcal{B}\left(\boldsymbol{\alpha}^*(\boldsymbol{m})\right)\nabla_{\boldsymbol{s}}\ln p(\boldsymbol{m} \mid s^t) - \nabla_{\boldsymbol{s}}\Phi(s^t)$ for $t \in [1, 2, \cdots, T]$. Based on the definitions of the stochastic gradient mapping $\hat{\mathcal{G}}^t$ and the $L$ smoothness of $\Phi$, we have*

$$\begin{aligned}
\Phi\left(s^{t+1}\right) - \Phi\left(s^t\right) &\leq \frac{L}{2}\left\|s^{t+1} - s^t\right\|^2 - \left\langle \nabla_s\Phi\left(s^t\right), s^t - s^{t+1}\right\rangle \\
&= \frac{L(\eta^t)^2}{2}\left\|\hat{\mathcal{G}}^t\right\|^2 - \eta^t\left\langle \mathcal{L}_\mathcal{B}\left(\boldsymbol{\alpha}^*(\boldsymbol{m})\right)\nabla_{\boldsymbol{s}}\ln p(\boldsymbol{m} \mid s^t) - \xi^{(t)}, \hat{\mathcal{G}}^t \right\rangle \\
&= \frac{L(\eta^t)^2}{2}\left\|\hat{\mathcal{G}}^t\right\|^2 - \eta^t\left\langle \mathcal{L}_\mathcal{B}\left(\boldsymbol{\alpha}^*(\boldsymbol{m})\right)\nabla_{\boldsymbol{s}}\ln p(\boldsymbol{m} \mid s^t), \hat{\mathcal{G}}^t \right\rangle + \eta^t\left\langle \xi^{(t)}, \hat{\mathcal{G}}^t \right\rangle \\
&\leq (\frac{L(\eta^t)^2}{2} - \eta^t)\left\|\hat{\mathcal{G}}^t\right\|^2 + \eta^t\left\langle \xi^{(t)}, \mathcal{G}^t \right\rangle + \eta^t\left\langle \xi^{(t)}, \hat{\mathcal{G}}^t - \mathcal{G}^t \right\rangle \\
&\leq (\frac{L(\eta^t)^2}{2} - \eta^t)\left\|\hat{\mathcal{G}}^t\right\|^2 + \eta^t\left\langle \xi^{(t)}, \mathcal{G}^t \right\rangle + \eta^t\|\xi^{(t)}\|^2 \\
&\leq (L(\eta^t)^2 - 2\eta^t)(\|\mathcal{G}^t\|^2 + \|\xi^{(t)}\|^2) + \eta^t\left\langle \xi^{(t)}, \mathcal{G}^t \right\rangle + \eta^t\|\xi^{(t)}\|^2,
\end{aligned}$$

*where the last line is obtained with Lemma 7 and $\left\|\hat{\mathcal{G}}^t\right\|^2 \leq 2(\|\mathcal{G}^t\|^2 + \|\xi^{(t)}\|^2)$.*

*By summing up from $t = 1$ to $T$, we derive that*

$$\sum_{t=1}^{T}\left(2\eta^t - L(\eta^t)^2\right)\left\|\mathcal{G}^t\right\|^2 \leq \Phi\left(s^1\right) - \Phi\left(s^{T+1}\right) + \sum_{t=1}^{T}\left(\eta^t\left\langle \xi^{(t)}, \mathcal{G}^t \right\rangle + (L(\eta^t)^2 - \eta^t)\left\|\xi^{(t)}\right\|^2\right).$$

*Since $\eta^t = \frac{c}{\sqrt{t}} \leq \frac{1}{L}$, we have $2\eta^t - L\eta^t \geq \eta^t \geq 0$. Denote $(\eta^t)' = \min\{\eta^t, t = 1, \cdots, T\} = \frac{c}{\sqrt{T}}$.*

*Then we can derive*

$$\sum_{t=1}^{T} \left(2\eta^t - L(\eta^t)^2\right) \geq \sum_{t=1}^{T} \eta^t,$$

*and*

$$\frac{1}{\sum_{t=1}^{T} \left(2\eta^t - L(\eta^t)^2\right)} \leq \frac{1}{\sum_{t=1}^{T} \eta^t} \leq \frac{1}{T(\eta^t)'} = \frac{1}{c\sqrt{T}}.$$

*Under the assumptions on $\mathbb{E}[\xi^{(t)}] = 0$ and $\mathbb{E}\|\xi^{(t)}\|^2 \leq \sigma^2$, we have*

$$\min_{1 \leq t \leq T} \mathbb{E}\left\|\mathcal{G}^t\right\|^2 \leq \frac{\sum_{t=1}^{T} \left(2\eta^t - L(\eta^t)^2\right)\left\|\mathcal{G}^t\right\|^2}{\sum_{t=1}^{T} \left(2\eta^t - L(\eta^t)^2\right)} \leq \frac{\Phi\left(s^1\right) - \Phi\left(s^{T+1}\right) + \sum_{t=1}^{T}(L(\eta^t)^2 - \eta^t)\sigma^2}{c\sqrt{T}}$$

$$\leq \frac{\Phi\left(s^1\right) - \Phi\left(s^{T+1}\right)}{c\sqrt{T}},$$

*where last inequality is obtained by $\eta^t \leq 1/L$ and $L(\eta^t)^2 - \eta^t \leq 0$.*

*Finally, it can be obtained that*

$$\min_{1 \leq t \leq T} \mathbb{E}\left\|\mathcal{G}^t\right\|^2 \lesssim \mathcal{O}\left(\frac{1}{\sqrt{T}}\right).$$

**Remark 11** *Zhou et al. (2022) demonstrate that with assumed variance $\sigma$, smoothness parameter $\ell$ and learning rate $\eta \leq \frac{2}{\ell}$, the average gradient $\frac{1}{T}\sum_{t=1}^{T} \mathbb{E}\left\|\mathcal{G}^t\right\|^2$ converges to a small constant $\frac{8-2\ell\eta}{2-\ell\eta}\sigma^2$, when $T \to \infty$.*

*Differently, we further adopt the learning rate $\eta = \frac{c}{t} \leq \frac{1}{L}$ ($c > 0$), and new inequalities to further derive an improved convergence rate, $\mathcal{O}(\frac{1}{\sqrt{T}})$, which converges to zero with $T \to \infty$.*

## H  OPTIMIZATION DETAILS

### H.1  DISCRETE MASKS $m$ TO CONTINUOUS PROBABILITY $s$

As introduced in (Zhou et al., 2022), the probabilistic bilevel problem is indeed a tight relaxation (although not equivalent) of the original discrete problem. For completeness, we summarize the reasons for such a transformation:

- The discrete masks $m = 0/1$ can be represented as a particular stochastic one by letting $s_i = 0/1$, thus we have $\min_{s \in \mathcal{C}} \Phi(s) \leq \min_{m \in \tilde{\mathcal{C}}} \tilde{\Phi}(m)$;

- The constraint on $s$ with $\ell$-1 regularization within $[0, 1]$ guides the most components of the optimal $s$ either 0 or 1, which has already been empirically validated in (Zhou et al., 2022);

- The new probabilistic form can be optimized directly with the gradient-based method as follows,
$$\nabla_s \Phi(s) = \nabla_s \mathbb{E}_{p(m|s)} \mathcal{L}\left(\alpha^*(m)\right)$$
$$= \nabla_s \int \mathcal{L}\left(\alpha^*(m)\right) p(m \mid s)dm$$
$$= \int \mathcal{L}\left(\alpha^*(m)\right) \frac{\nabla_s p(m \mid s)}{p(m \mid s)} p(m \mid s)dm$$
$$= \int \mathcal{L}\left(\alpha^*(m)\right) \nabla_s \ln p(m \mid s)p(m \mid s)dm$$
$$= \mathbb{E}_{p(m|s)} \mathcal{L}\left(\alpha^*(m)\right) \nabla_s \ln p(m \mid s),$$
which obviously reduced the computation cost of bilevel problems.

---

**Algorithm 2:** Projection Operation $\mathcal{P}_\mathcal{C}(\boldsymbol{a})$

---

**Input**: Vector $\boldsymbol{a} \in \mathbb{R}^p$, core variables $C$, Domain $\mathcal{C} = \{\boldsymbol{s} : 0 \preceq \boldsymbol{s} \preceq 1, \|\boldsymbol{s}\|_1 \leq C\}$.

    1) Computing auxiliary variable $b$ satisfying:
        $\mathbf{1}^\top [\min(1, \max(0, \boldsymbol{a} - b \cdot \mathbf{1}))] - C = 0$
    2) Computing auxiliary variable $c$ satisfying:
        $c \leftarrow \max(0, b)$
    3) Update $\boldsymbol{a}$:
        $\boldsymbol{a}^* \leftarrow \min(1, \max(0, \boldsymbol{a} - c \cdot \mathbf{1}))$
**Output**: $\mathcal{P}_\mathcal{C}(\boldsymbol{a}) = \boldsymbol{a}^*$.

---

### H.2 PROJECT OPTIMIZATION FROM PROBABILITY $\boldsymbol{s}$ TO DOMAIN $\boldsymbol{C}$

Inspired from existing works (Zhao et al., 2023; Zhou et al., 2022), the algorithm for project operation from probability $\boldsymbol{s}$ to domain $\mathcal{C}$ is realized with projection operation $\mathcal{P}_\mathcal{C}(\boldsymbol{s})$, which is summarized in Algorithm 2. Indeed, the Lagrangian multiplier, as well as the bisection method, are employed in designing this algorithm, yielding a closed-form solution. The theoretical guarantee for learning masks on all samples $\boldsymbol{m} \in \mathbb{R}^N$ can be found at (Zhou et al., 2022). Moreover, this paper focuses on the masks on all variables $\boldsymbol{m} \in \mathbb{R}^p$. For completeness, we present the corresponding theoretical proof as follows.

**Proof 11** *Given variable $\boldsymbol{a} \in \mathbb{R}^p$, in order to project $\boldsymbol{a}$ to set $\mathcal{C}$, we introduce the following problem with constraints:*

$$\min_{\boldsymbol{s} \in \mathbb{R}^p} \frac{1}{2}\|\boldsymbol{s} - \boldsymbol{a}\|^2, \; s.t. \mathbf{1}^T \boldsymbol{s} \leq C \text{ and } 0 \leq \boldsymbol{s}_i \leq 1,$$

*where $\mathbf{1} = (1, 1, \cdots, 1) \in \mathbb{R}^p$ and $\boldsymbol{s}$ is the ideal output after projection.*

*The above problem can be resolved by the commonly used Lagrangian multiplier method formulated with:*

$$L(\boldsymbol{s}, b) = \frac{1}{2}\|\boldsymbol{s} - \boldsymbol{a}\|^2 + b\left(\mathbf{1}^\top \boldsymbol{s} - C\right) = \frac{1}{2}\|\boldsymbol{s} - (\boldsymbol{a} - b\mathbf{1})\|^2 + b\left(\mathbf{1}^\top \boldsymbol{a} - C\right) - \frac{n}{2}b^2. \quad (31)$$

*where the auxiliary variable $b \geq 0$ and $0 \leq s_i \leq 1$.*

*To minimize above problem equation 31 with respect to $\boldsymbol{s}$, we can derive that $\tilde{s} = \mathbf{1}_{\boldsymbol{a}-b\mathbf{1}\geq 1} + (\boldsymbol{a} - b\mathbf{1})_{1 > \boldsymbol{a}-b\mathbf{1}>0}$.*

*Then we can develop two auxiliary functions as follows:*

$$g(b) = L(\tilde{\boldsymbol{s}}, b) = \frac{1}{2}\left\|[\boldsymbol{a} - b\mathbf{1}]_- + [\boldsymbol{a} - (b+1)\mathbf{1}]_+\right\|^2 + b\left(\mathbf{1}^\top \boldsymbol{a} - s\right) - \frac{n}{2}b^2$$

$$= \frac{1}{2}\left\|[\boldsymbol{a} - b\mathbf{1}]_-\right\|^2 + \frac{1}{2}\left\|[\boldsymbol{a} - (b+1)\mathbf{1}]_+\right\|^2 + b\left(\mathbf{1}^\top \boldsymbol{a} - s\right) - \frac{n}{2}b^2, \text{for} \quad b \geq 0,$$

*and*

$$g'(b) = \mathbf{1}^\top [b\mathbf{1} - \boldsymbol{a}]_+ + \mathbf{1}^\top [(b+1)\mathbf{1} - \boldsymbol{a}]_- + \left(1^T \boldsymbol{a} - s\right) - nb = \mathbf{1}^\top \min(1, \max(0, \boldsymbol{a} - b\mathbf{1})) - C, \text{for} \quad b \geq 0.$$

*Finally, with the monotone decreasing property of $g'(b)$, a bisection method is exploited to solve the equation $g'(b) = 0$ with solution $b^*$. Because $g(b)$ increases in $(-\infty, b^*]$ and decreases in $[b^*, +\infty)$, we can conclude that the maximum of $g(b)$ is obtained at 0 if $b^* \leq 0$ and $b^*$ if $b^* > 0$.*

*Finally, by setting $c^* = \max(0, b^*)$, we have the output*

$$\boldsymbol{s}^* = \mathbf{1}_{\boldsymbol{a}-c^*\mathbf{1}\geq 1} + (\boldsymbol{a} - c^*\mathbf{1})_{1 > \boldsymbol{a}-c^*\mathbf{1}>0} = \min(1, \max(0, \boldsymbol{a} - c^*\mathbf{1})).$$

### H.3 OPTIMIZATION FOR UPPER-LEVEL PROBLEM

The detailed optimization steps for probabilistic S$^2$MAM have already been introduced in Section 2.4, which has been further summarized in Algorithm 1. Notably, this policy gradient estimation approach significantly improves the algorithmic efficiency by reducing the computational burden associated with the hypergradient of bilevel optimization problems.

### H.4 OPTIMIZATION FOR LOWER-LEVEL PROBLEM

Based on the principle of the Alternating Direction Method of Multipliers (ADMM), an optimization algorithm is designed to solve the manifold-regularized sparse additive problem at the lower level. For simplicity, merely the regression task with squared loss is presented here.

Here we generate the Gram matrix over labeled and unlabeled points $\mathbf{K} = \left(\mathbf{K}^{(1)}, \ldots, \mathbf{K}^{(p)}\right) \in \mathbb{R}^{(l+u) \times (l+u)p}$ with masked input $\boldsymbol{m} \odot x_i$ where $i \in [1, 2, \cdots, l+u]$, the model coefficient $\boldsymbol{\alpha} = \left(\alpha^{(1)^T}, \ldots \alpha^{(p)^T}\right)^T \in \mathbb{R}^{(l+u)p}$, and the label vector $Y = (y_1, \ldots, y_l, 0, \ldots, 0)^T \in \mathbb{R}^{l+u}$. Then, the lower-level problem can be reformulated as

$$\boldsymbol{\alpha}^* = \arg \min_{\boldsymbol{\alpha} \in \mathbb{R}^{(l+u)p}} \frac{1}{l}(Y - J\mathbf{K}\boldsymbol{\alpha})^T (Y - J\mathbf{K}\boldsymbol{\alpha}) + \lambda_1 \sum_{j=1}^{p} \tau_j \left\|\boldsymbol{\alpha}^{(j)}\right\|_2 + \frac{\lambda_2}{(l+u)^2} \boldsymbol{\alpha}^T \mathbf{K} L \mathbf{K} \boldsymbol{\alpha}, \quad (32)$$

where the matrix $J = \text{diag}(1, \ldots, 1, 0, \ldots, 0)$ is an $(l+u) \times (l+u)$ diagonal matrix with the first $l$ diagonal entries as 1 and the rest as 0 (Belkin et al., 2006).

By introducing the auxiliary variable $\vartheta = \left(\vartheta^{(1)^T}, \ldots, \vartheta^{(p)^T}\right)^T \in \mathbb{R}^{(l+u)p}, \vartheta^{(j)} = \left(\vartheta_1^{(j)}, \ldots, \vartheta_{l+u}^{(j)}\right) \in \mathbb{R}^{l+u}$, equation 32 can be rewritten as:

$$\min_{\alpha, \vartheta} \frac{1}{l}(Y - J\mathbf{K}\alpha)^T (Y - J\mathbf{K}\alpha) + \lambda_1 \sum_{j=1}^{p} \tau_j \left\|\vartheta^{(j)}\right\|_2 + \frac{\lambda_2}{(l+u)^2} \alpha^T \mathbf{K} L \mathbf{K} \alpha, \quad \text{s.t.} \quad \alpha - \vartheta = 0. \quad (33)$$

Hence, by introducing the auxiliary variable $\vartheta \in \mathbb{R}^{(l+u)p}$ and the Lagrange multiplier $\Lambda \in \mathbb{R}^{(l+u)p}$, the scaled augmented Lagrangian function of the primal problem equation 32 is

$$
\begin{aligned}
L(\alpha, \vartheta, \Lambda) = &\frac{1}{l}(Y - J\mathbf{K}\alpha)^T (Y - J\mathbf{K}\alpha) + \lambda_1 \sum_{j=1}^{p} \tau_j \left\|\vartheta^{(j)}\right\|_2 \\
&+ \frac{\lambda_2}{(l+u)^2} \alpha^T \mathbf{K} L \mathbf{K} \alpha + \frac{\varrho}{2} \|\alpha - \vartheta - \Lambda\|_2^2 - \frac{\varrho}{2} \|\Lambda\|_2^2,
\end{aligned}
\quad (34)
$$

where $\varrho > 0$ is a positive penalty coefficient.

Given initialized parameters $(\alpha^0, \vartheta^0, \Lambda^0)$ and convergence criterion $\epsilon$, the manifold regularized additive regression problem with squared loss can be solved by the following iterative steps:

(1) Fix $\vartheta^t$ and $\Lambda^t$, and update the model coefficient $\alpha^{t+1}$:

$$\alpha^{t+1} = \arg \min_{\alpha} \frac{1}{l}(Y - J\mathbf{K}\alpha)^T (Y - J\mathbf{K}\alpha) + \frac{\lambda_2}{(l+u)^2} \alpha^T \mathbf{K} L \mathbf{K} \alpha + \frac{\varrho}{2} \|\alpha - \vartheta^t - \Lambda^t\|_2^2.$$

$\alpha^{t+1}$ can be calculated by the derivative of the objective function, which vanishes at the minimizer:

$$\frac{1}{l}(Y - J\mathbf{K}\alpha)^T (-J\mathbf{K}) + \left(\frac{\lambda_2}{(l+u)^2} \mathbf{K} L \mathbf{K} + \varrho(\alpha - \vartheta^t - \Lambda^t)^T\right)\alpha = 0.$$

(2) Fix $\alpha^{t+1}$ and $\Lambda^t$, and update the auxiliary variable $\vartheta^{t+1}$:

$$\vartheta^{t+1} = \arg \min_{\vartheta} \frac{1}{2} \|\alpha^{t+1} - \vartheta + \Lambda^t\|_2^2 + \frac{\lambda_1}{\varrho} \sum_{j=1}^{p} \tau_j \left\|\vartheta^{(j)}\right\|_2. \quad (35)$$

With fixed $\alpha^{t+1}$ and $\Lambda^t$, equation 35 is equivalent to the following $p$ subproblems:

$$(\vartheta^{(j)})^{t+1} = \arg \min_{\vartheta^{(j)}} \frac{1}{2} \left\|(\alpha^{(j)})^{t+1} - \vartheta^{(j)} + (\Lambda^{(j)})^t\right\|_2^2 + \frac{\lambda_1 \tau_j}{\varrho} \left\|\vartheta^{(j)}\right\|_2.$$

Thanks to the soft thresholding operators (Boyd et al., 2011; Chen et al., 2020), we have

$$(\vartheta^{(j)})^{t+1} = S_{\lambda_1 \tau_j / \varrho} \left( (\alpha^{(j)})^{t+1} + (\Lambda^{(j)})^t \right), \quad j = 1, \ldots, p,$$

where the soft thresholding operator $S$ stands for

$$S_k(a) = (a - k/\|a\|_2)_+ \, a.$$

(3) Fix $\alpha^{t+1}$ and $\vartheta^{t+1}$, and update the Lagrange multiplier $\Lambda^{t+1}$:

$$\Lambda^{t+1} = \Lambda^t + \alpha^{t+1} - \vartheta^{t+1}.$$

Denote the objective function of lower level problem as $\mathcal{R}(\alpha)$ (standing for $\mathcal{R}(\boldsymbol{\alpha}; \boldsymbol{m}; \boldsymbol{L})$) parameterized by model coefficient $\alpha$ (and mask $\boldsymbol{m}$ learned by upper level problem, the Laplacian matrix $\boldsymbol{L}$). The above three iterative steps form a loop until the following convergence conditions are met at $(t+1)$-th iteration:

$$|\mathcal{R}(\alpha^{t+1}) - \mathcal{R}(\alpha^t)| \leq \epsilon. \tag{36}$$

Then the updating process stops and the output $\alpha^{t+1}$ can be considered as the desired model coefficient. Moreover, inspired by (Chen et al., 2020; Yuan et al., 2023), the early-stop condition in equation 36 could also be set as

$$\|\alpha^{t+1} - \alpha^t\|_\infty \leq \epsilon \quad \text{and} \quad \|\alpha^{t+1} - \vartheta^{t+1}\|_\infty \leq \epsilon.$$

### H.5 ANALYSIS ON COMPUTATION COMPLEXITY

With the $\epsilon$-stationary point defined in (Ji et al., 2021; Chu et al., 2024; Zhang et al., 2024), we conclude that the optimization for the upper problem requires at most $T = \mathcal{O}(\epsilon_1^{-2})$ iterations before reaching $\epsilon_1$-stationary based on Theorem 1. The lower level requires $\mathcal{O}(K(l+u))$ steps on gradient computations and $\mathcal{O}(p(l+u))$ assigning masks per outer iteration. Notice that K is the inner iteration and $p$ is the input dimension. The lower problem optimized by ADMM (Culp, 2011; Culp & Michailidis, 2008) exhibits a sublinear convergence rate $\mathcal{O}(1/K)$ with respect to the Nash point, provided the threshold $1/K \lesssim \epsilon_2$ is satisfied, when the lower problem satisfies the convexity condition. Please refer to (Wang & Zhao, 2022) for the corresponding proof of general ADMM optimization.

In summary, the computation complexity of S$^2$MAM reaches $\mathcal{O}\left(\frac{p(l+u)}{\epsilon_1^2 \epsilon_2}\right)$, which is competitive with some latest bilevel algorithms(Liu et al., 2022a; Xiao et al., 2023). Empirically, please refer to *Appendix G* for convergence analysis and Section E for some experimental comparisons on training time cost.

## I IMPACT, CHALLENGES, AND LIMITATIONS

### I.1 IMPACT STATEMENT

This paper presents work aimed at advancing the field of Machine Learning. We believe this work can deepen our understanding of the interplay between generalization and variable selection, and widen the applications of bilevel optimization for interpretable prediction.

### I.2 NOVELTY AND DIFFERENCE TO RELATED WORK

In the following, we summarize and restate the novelty and contributions of S$^2$MAM to our bilevel baseline, PBCS (Zhou et al., 2022), and another classical work on generalization theory (Cao & Chen, 2012), from the perspectives of algorithm design, learning region, and theoretical analysis.

#### I.2.1 MOTIVATION AND ALGORITHMIC DESIGN

While we adopt a similar policy gradient estimation (PGE) technique to avoid implicit differentiation (e.g., Hessian/Jacobian computations), significant technical hurdles arise when adapting PBCS to

semi-supervised feature selection with interpretable additive schemes. Below, we first recall the differences, clarify these challenges, and summarize our novel contributions to address them.

**PBCS (Supervised Coreset Selection)**: **(1) Upper-level**: Learns an $l$-dimensional sample mask using $l$ labeled points. **(2) Lower-level**: Trains a CNN on the same $l$ labeled points (masked subset) via standard backpropagation. **(3) Data usage**: Single labeled set for both levels.

**Our Work (Semi-Supervised Feature Selection)**: **(1) Upper-level**: Learns a $p$-dimensional feature mask using only $l$ labeled points (challenges arise with high-dimension $p$, e.g., $l < p$). **(2) Lower-level**: Solves a Laplacian-regularized sparse additive model over $l + u$ points (labeled + unlabeled): Mask impacts feature-wise additive terms ($f_j$, dimensions $p$, $j \in \{1, 2, \cdots, p\}$). **(3) Data usage**: Labeled data for the supervised upper problem and the supervision part of the lower problem; unlabeled data for lower-level manifold regularization.

### I.3 CORE TECHNICAL CHALLENGES IN ALGORITHM DESIGN

The following three challenges make the method in PBCS inapplicable to the semi-supervised learning (SSL) scenarios.

**(1) Challenge 1 (Mask Dimension Mismatch)**:

When $p > l$, learning a $p$-dimensional mask from $l$ labels is challenging (ill-posed (Friedman, 1989; Meng et al., 2014)). PBCS avoids this since its mask dimension ($l$) matches supervision.

**Our Solution**:

Introduce sparsity regularization (via $\ell_{2,1}$-norm) to stabilize mask learning.

For extremely high-dimensional data (e.g., 512×512 images from CelebA-HQ), employ a pre-trained and frozen CNN on limited labeled samples for feature extraction to enhance the capability of $S^2MAM$ in handling such SSL tasks.

**(2) Challenge 2 (Computational Cost of Laplacian)**:

Each mask update requires recomputing pairwise similarities across all $l + u$ samples, costing $\mathcal{O}((l + u)^2 p)$. For large $u$, this might dominate the training.

**Our Solution**: Accelerate with Random Fourier Features (RFF), reducing Laplacian-based cost to $O((l + u)d)$ (where $d \ll l + u$).

**(3) Challenge 3 (Specialized Lower-Level Solver)**:

PBCS uses a standard SGD for CNNs.

**Our Solution**: Our lower-level requires solving:

$$\min_{\boldsymbol{\alpha}} \underbrace{\frac{1}{l} \sum_{i=1}^{l} \ell(f(x_i \odot \boldsymbol{m}), y_i)}_{\text{supervised loss}} + \lambda_1 \underbrace{\sum_{j=1}^{p} \tau_j \|\boldsymbol{\alpha}^{(j)}\|_2}_{\text{Sparsity regularization}} + \underbrace{\frac{\lambda_2}{(l+u)^2} \mathbf{f}^T \boldsymbol{L} \mathbf{f}}_{\text{Laplacian regularization}} \ . \tag{37}$$

This kernel-based objective, regularized by two different penalties, demands a custom solver as introduced in our Appendix H.4.

Practically, this work naturally solves the question: ***How to retain interpretability while filtering redundant or corrupted features efficiently in semi-supervised settings?***

### I.4 THEORETICAL CHALLENGES AND NEW TECHNIQUES

**The hypothesis space** of $S^2MAM$ is additive and data-dependent, instead of the data-independent hypothesis in (Cao & Chen, 2012). Concretely, the Regularization error of $S^2MAM$ is derived under the assumption of generalized additive forms.

**Our excess-risk decomposition** contains four terms: Regularization, Sample, Manifold, and an extra Hypothesis error absent in (Cao & Chen, 2012), which requires new auxiliary functions and proof techniques.

**The manifold error** of S$^2$MAM is decomposed into p additive U-statistic deviations, bounded by a Hilbert-space Bernstein-type concentration inequality, rather than by the spectral-based lemma (Theorem 4.11 of (Cao & Chen, 2011)) in (Cao & Chen, 2012).

### I.5 LIMITATIONS AND DISCUSSIONS

#### LIMITATIONS AND EXISTING SOLUTIONS

This paper proposes a new bilevel manifold regularization approach for semi-supervised learning tasks, featuring an automatic feature masking mechanism. Theoretically, we establish the foundations of learning theory, including the computing convergence and the generalization error analysis. To the best of our knowledge, this is the first work to bound the excess risk of a semi-supervised additive model. Our results show better convergence performance than those in (Zhou et al., 2022). While inspired by the PGE technique (Zhou et al., 2022), our proposal S$^2$MAM addresses distinct challenges in semi-supervised feature selection on high-dimensional mask learning, Laplacian scalability, and specialized optimization. Our innovations (RFF acceleration, sparsity regularization, ADMM solver) enable robust performance where direct extension of PBCS fails. Empirically, we verify the effectiveness of the proposed approach using both synthetic and real-world datasets. We designed a novel optimization algorithm for the proposed manifold-regularized sparse additive model (see Appendix H.4). In the implemented codes, we further provide the settings of spline-based additive models. However, some limitations still exist, including computational difficulties with large-scale datasets and the assumption of bounded output.

Fortunately, as introduced in Appendix E, our proposal S$^2$MAM can also handle high-dimensional data with the aid of some preprocessing techniques. An interesting approach for dealing with high-dimensional data, such as images, is to extract the feature vectors first, which has been widely employed in various supervised (Su et al., 2023) and semi-supervised works (Qiu et al., 2018; Nie et al., 2019; Kang et al., 2020; Nie et al., 2021). The random Fourier technique (Rahimi & Recht, 2007; Wang et al., 2023) can also be considered to accelerate the computation process further. Theoretically, the bounded condition of the response can be relaxed to include the unbounded output, e.g., replacing it by the $1 + \epsilon$ moment bounded assumptions (Feng, 2021; Feng & Wu, 2022)). The neural additive modeling strategy (Agarwal et al., 2021; Yang et al., 2020) is another interesting and compelling direction for improving the non-linear approximation ability and prediction performance of S$^2$MAM. Furthermore, in the image-type experiments, mapping from mask importance back to image attributes is also meaningful to identify which of these extracted features are informative for the classification task. In addition, the current generalization analysis focuses on the basic model of S$^2$MAM, which can be further improved to match the bilevel manifold regularization tightly.

#### HOW DOES S$^2$MAM SCALE AND PERFORM WITH EXTREMELY HIGH DIMENSIONS?

**Common Challenges in S$^2$MAM and SSL when $n \ll p$.** When $n \ll p$, the sample-covariance matrix is singular and the graph-Laplacian $L \in \mathbb{R}^{n \times n}$ becomes rank-deficient, which violates the restricted strong convexity condition required by manifold regularization. Furthermore, the bilevel optimization becomes NP-hard, and the space complexity for directly computing on raw data increases to $\mathcal{O}(n^2 p)$.

**Our solutions and suggestions** As reported in Section 4 and the Appendix, the CNN-based feature extraction and random Fourier transformation have been successfully utilized to enhance computational efficiency on large-scale image datasets.

The semi-supervised modeling of extremely high-dimensional data remains a challenging task (Azriel et al., 2022; Mai & Couillet, 2021) in both practical and theoretical analyses, which is listed as a future research goal.

### DISCLOSURE OF GENERATIVE AI USAGE

GenAI tools were used during the editing (e.g., grammar, spelling, word choice). And the authors are fully accountable for the content.

