# OpenReview forum: "S$^2$MAM: Semi-supervised Meta Additive Model for Robust Estimation and Variable Selection"
_ICLR.cc/2026/Conference — Submitted to ICLR 2026_

### Official Review · Reviewer_rNn1 · 2025-10-29

**Soundness:** 3
**Presentation:** 3
**Contribution:** 3
**Rating:** 8
**Confidence:** 3

**Summary:**

This paper introduces a SSL method, that performs robust feature selection combined with manifold regularization. This makes the method also interpetable (because we can see which features are used). Feature selection is achieved through l2,1 regularization; but also a bi-level optimization routine is used on top, to update the masks of features and update the regularization terms. The bi-level optimization is relaxed into a probabilistic bilevel problem, which can be solved efficiently. Several proofs are given showing the algorithm converges and enjoys good generalization bounds. An extensive empirical evaluation is compared, showing the method obrtains state-of-the-art performance, especially when uninformative and noisy variables are included.

**Strengths:**

- Extensive empirical evaluation
- The proofs are a nice addition, but I did not check them carefully
- I like how the work is presented, and the comparison with the SOTA with the table makes the contribution very clear

**Weaknesses:**

- I found it hard to understand the bilevel optimization; I think there might be a few notational errors (see questions below), but maybe a figure may also help.

- I find it hard to evaluate whether the proposed method improves on state-of-the-art due its technical innovations, or whether it simply can remove improperly scaled features? Are features scaled prior to fitting models (also for competetitors)? See also questions.

- I think reflections on the weaknesses and future work would be better suited to the main body.

**Questions:**

1. Are the features scaled in the experiments? How? If not, why not? For me, the features that are normally distributed with a mean of 100 and std of 100, are clearly going to mess with algorithms that are not scaling features. In this case, your algorithm may just be favored because it will ignore unscaled features; meanwhile, I think your claim is that your algorithm is robust for other reasons. So it would be good to include an experiment where features are properly scaled (for all baselines).

2. I think in line 242 it should be alpha^*? For me, the bilevel optimization was not clear. Is it true that alpha^* is the result of an optimization? I am not sure what is the difference with alpha hat in line 251.

3. "It is usually intractable to solve the bilevel problem." This is a bit vague. Can you make this more concrete? (if possible). Furthermore, can you clarify why this is difficult?

4. I did not understand how C was set (remark 6). Can you explain in more detail? E.g. I don't understand over which data quantiles are taken, and why multiple models are trained. I checked the Appendix but I still was a bit lost.

Some other minor issues:

5. Why logistic loss for S^2 MAM? In principle we can plug in any loss, right?

6. Line 254; I think it would be good to more clearly indicate that L depends on m, e.g. write L(m)

7. Its unclear why D_{meta} was introduced and why its called D_{meta}. Isn't it simply the case that this is the labeled data? How it is written now, it comes accross as if D_{meta} could be instantiated in different ways.

8. Why is Tau needed? It is introduced, but Appendix B.2.3 states you always set all Tau equal to 1?

9. "As in Table 2 with ..."; do you mean "As can be seen in Table 2" ?

10. Line 295; are you now saying here that m^t is a sample from p(s|m) ? I found it a bit strange that this is a single sample; but I can imagine that if you iterate long enough it will still converge? E.g. you could also update L^t with multiple samples from m. Is this something you considered?

---

> ### Author Response · Authors · 2025-11-20
>
> We sincerely thank you for your insightful comments and recognition of our paper!
>
> Following your suggestions, we've made corresponding changes in the revised PDF, which could be downloaded by clicking the PDF button.
>
> # To Weakness
>
> Thanks for your suggestions on enhancing the readability by illustrating the bilevel mechanism with visualized plots.
>
>
>
> And we've drawn the relationship and **optimization procedure of our bilevel model in Figure 2 in Page 24 of Appendix C.2**.
>
>
>
>
>
> # To Questions
>
> ## 1)
>
> We appreciate your feedback regarding the scale operation.
>
>
>
> In our experiments, features were not explicitly scaled prior to model fitting for any methods, including competitors. We deliberately maintained the original feature scales to demonstrate how different algorithms handle raw data with noisy variables. The noisy variables drawn from N(100, 100) were intentionally designed to have substantially different scales to test robustness.
>
>
>
> Following your suggestions, **we've included an additional experiment in Table 12 (Page 32) of the revised PDF**, where all methods use scaled features to demonstrate that S2MAM's advantages extend beyond handling scale differences. The new results in the lower panel of Table 12 are colored in red.
>
>
>
> While you raise a valid point that this may favor algorithms that ignore unscaled features, our experiments in **Tables 2 and 3 without scaling** and **Table 12 with scaling show that S2MAM maintains superior performance** under noisy scenarios even when compared against methods that could theoretically handle scale differences internally.
>
>
>
> ## 2)
>
> Thank you for pointing out this notational confusion.
>
>
>
> We acknowledge that this distinction could be clearer in the paper by rewriting $\alpha\*$ to $\hat{\alpha}$, which has been corrected in Eq.(3) of the revised PDF.
>
>
>
> ## 3）
>
> The key difficulty arises from several factors:
>
>
>
>  (1) The discrete nature of the mask variables makes the problem NP-hard;
>
>
>
>  (2) the nested optimization structure requires solving an inner optimization problem for each evaluation of the outer objective;
>
>
>
>  (3) Computing gradients through the lower-level optimization typically requires expensive operations like calculating Hessian or Jacobian matrices.
>
>
> In our revision, we've expanded on these specific challenges in Section 2.2 and cite relevant literature [1] that discusses these computational obstacles in more detail.
>
>
> [1] Approximation methods for bilevel programming
>
>
> ## 4）
>
> We apologize for the confusion regarding parameter C selection.
>
> In Remark 6 (which appears in the full paper but not in the excerpt provided), individually optimal C_1 of each model is determined through a binary cross-validation procedure, where we evaluate model performance across different values of C (number of selected variables). This strategy balances the trade-off between model complexity and performance.
>
> Specifically, we select the quantiles of  C_2 derived from the preliminary models with all alternative C_1, and then train multiple models with C_2. For ensuring fairness, the selected C (=C_2) indeed is shared to all baselines requiring maximum selected features, which has been suggested by other reviewers before, where **the ground truth informative size C should be fixed in current data and irrelevant to models**.
>
> In the revision, we've expanded this explanation in both the main paper and Appendix to make the selection process clearer. **Please refer to Appendix C.3.3: Impact of selected core size C for more details on core size setting**.
>
> ## 5)
>
> You're correct that, S2MAM can accommodate various loss functions.
>
> We chose the logistic loss for classification tasks primarily because it provides a natural probabilistic interpretation and works well with the RKHS framework we employ. Additionally, logistic loss has been widely used in similar semi-supervised learning settings (e.g., LapSVM), allowing for fairer comparisons with baseline methods. This could be a comparison to our baseline TSpAM (Wang et al., ICML 2023), where this single-level supervised additive model is also equipped with logistic loss.
>
> For regression tasks, as mentioned in Section 4.1, we use the squared loss. The choice of loss function is indeed flexible in our framework, and we've clarified this point in the revision by explicitly noting that other loss functions could be substituted depending on the specific application requirements.
>
> ## 6)
>
> Thank you for this valuable suggestion. You're right that explicitly indicating the dependence of L on m would improve clarity.
>
> In line 254 and throughout the paper, the Laplacian matrix L is indeed computed based on the masked similarity matrix W, which depends on the mask m.
>
> In the revised PDF, we've modified the notation to **write $L_m$ instead of L to make this dependence explicit**. This change will help readers better understand the relationship between the variable masks and the manifold regularization component of our model.

---

> ### Author Response · Authors · 2025-11-20
>
> ## 7)
>
> Thanks for your comment!
>
>
>
> D_meta is indeed simply the labeled data, and we introduced this notation to emphasize its role in the meta-learning framework where the upper-level optimization is performed. The term "meta" here refers to the fact that we're learning the variable selection strategy (the masks) based on the performance on this labeled data, which is a meta-learning approach.
>
>
>
> We acknowledge that this terminology might be confusing without additional context. In the revision, we've clarified that D_meta represents the labeled dataset used for the upper-level optimization and explain why we use the "meta" designation in the context of our bilevel optimization framework.
>
>
>
> ## 8)
>
>
>
> Yes, you're right. In our experiments, we set all $\tau_j = 1$  for j  in  [1, 2, ..., p] as mentioned in Appendix B.2.3.
>
>
>
> **The $\tau$ parameter was included in the formulation to provide flexibility in assigning different weights to variables based on prior knowledge or importance.**
>
> While we didn't utilize this flexibility in our current experiments (treating all variables equally a priori), the parameter allows for potential extensions where domain knowledge could be incorporated. For instance, in applications where certain variables are known to be more reliable, $\tau_j$ could be adjusted accordingly. We've clarified this point in the revision, explaining that while τ provides theoretical flexibility, our current implementation uses uniform weights.
>
>
>
> We've added relevant discussions in the revision in Line 1183 with red color.
>
>
>
> ## 9)
>
> We appreciate your attention to detail in helping us improve the presentation of our work.
>
>
>
> The phrase "As in Table 2 with..." has been rewritten with "As present in Table 2, " in the revision.
>
>
>
> ## 10）
>
> Your observation about the sampling of $m^t$ is insightful.
>
>
>
> In line 295, $m^t$ is indeed a single sample from the Bernoulli distribution with parameters $s^t$. We chose this approach primarily for computational efficiency, as sampling multiple masks would require multiple evaluations of the lower-level optimization problem, significantly increasing the computational burden. While using a single sample introduces stochasticity, our empirical results show that the algorithm still converges effectively with this approach.
>
>
>
> We did consider using multiple samples, as you suggested, but found that the performance improvement did not justify the additional computational cost. In the revision, we've added a brief discussion of this design choice and mentioned that multiple sampling could be explored in future work for potentially improved stability at the expense of computation time.
>
>
>
> **Thanks again for your careful review and valuable suggestions! We've made corresponding changes in the revision for improvement.**
>
> **Please let us know if you have any additional suggestions! We will be happy to revise our paper accordingly.**

---

### Official Review · Reviewer_YaK4 · 2025-10-30

**Soundness:** 3
**Presentation:** 3
**Contribution:** 3
**Rating:** 6
**Confidence:** 3

**Summary:**

The paper proposes $S\^2MAM$ , a bilevel optimization framework that couples feature-level masking with manifold regularization to improve robustness and interpretability in semi-supervised learning. Experiments on synthetic data, UCI-style regressions, ADNI cognitive scores , and COIL-20 classification show improved robustness under injected redundant/noisy features versus, and several deep SSL baselines. For scalability the authors use CNN embeddings for images and RFF to reduce kernel cost from $O((\ell+u)^2)$ to $O((\ell+u)D)$.

**Strengths:**

1. Variable masks simultaneously reduce noise impact on the Laplacian and yield clear feature importance.
2. Policy-gradient updates with projection have a stated $O(1/\sqrt{T})$ convergence; a generalization bound is provided for the base model.
3. Synthetic, UCI, ADNI regression, and COIL-20 classification; robustness under injected noisy/redundant features is consistently strong.
4. CNN embeddings for images and RFF for kernel approximation; simple to integrate in non-deep settings.
5. Identifies how fixed similarity metrics are distorted by irrelevant variables and addresses it directly.

**Weaknesses:**

1. The current statistical bound omits the combined stochastic mask + adaptive graph procedure. This leaves a gap between the analyzed surrogate and the implemented algorithm.

2. The paper does not report whether baseline/control-variate techniques are used, how many mask samples are drawn per iteration, or temperature settings / comparisons to Gumbel-Softmax. Without these, it is hard to assess variance, convergence behavior, and accuracy.

3. Graph construction and updates are not evaluated against practical alternatives (e.g., kNN graphs, ANN search, sparsification, mini-batch graph updates) nor against the full $O(n^2)$ similarities baseline.

4.  The comparison set focuses on manifold methods and omits competitive consistency/augmentation SSL baselines.

5. The paper does not report sensitivity to key hyperparameters ($C,\mu,D$), learning rates, or projection radius; nor does it examine mask stability across random seeds or its link to downstream interpretability (e.g., ADNI feature groups).

6. The “Unlabeled” column is not precisely defined, and for COIL-20 with CNN features the semantics of variables and how mask importance maps back to image attributes are unclear.

**Questions:**

Please see Weaknesses.

---

> ### Author Response · Authors · 2025-11-20
>
> We sincerely thanks for your insightful comments and suggestions on our paper.
>
> Following your suggestions, we've made corresponding changes in the revised PDF, which could be downloaded by clicking the PDF button.
>
> ## 1)
> Regarding the statistical bound omission, Theorem 2 in Section 3.2 focuses on the basic model (2) from Section 2.1 without incorporating the stochastic masking and adaptive graph procedure, which **has been acknowledged in Appendix  I: Impact, Challenges and Limitations.**
>
> The existing theoretical analysis serves as a foundation for understanding the generalization properties of the underlying manifold-regularized additive model. Meanwhile, the generalization of stochastic bilevel scheme with supervised additive algorithm as lower model has already discussed in [1].
>
> Thanks for your insightful comments. Future work could extend the theoretical analysis to encompass the complete stochastic masking procedure and manifold regularization, which would be a valuable and challenging direction for research.
>
> [1] Multi-task Additive Models for Robust Estimation and Automatic Structure Discovery. NeurIPS 2020.
>
> ## 2)
>
> We appreciate the reviewer's observation regarding implementation details!
>
> Our probabilistic bilevel optimization in Section 2.4 follows the policy gradient estimation approach referenced in Zhou et al. (2022), drawing one mask sample per iteration as specified in Algorithm 1.
>
> We did not employ baseline/control-variate techniques or use temperature settings in our implementation. While we recognize that Gumbel-Softmax could be an alternative approach, **our method was designed to maintain the discrete nature of the masks while enabling gradient-based optimization through the probabilistic formulation**.
>
> **The convergence behavior has been theoretically supported by Theorem 1 in Section 3.1 and empirically validated in Figure 8 of Appendix C.3.4**, which establishes the convergence properties of our policy gradient estimation approach.
>
> ## 3）
>
> Regarding graph construction alternatives, we acknowledge that our paper focuses primarily on the masking innovation rather than comprehensive graph construction comparisons.
>
> As described in Section 2.4, we use the Gaussian similarity measure Wij = exp{−∥xi ⊙ mt − xj ⊙ mt∥22/µ2} for constructing the similarity matrix. We do address computational efficiency in the final paragraph of Section 2.4, mentioning strategies like preprocessing with pretrained CNNs and Random Fourier Features to reduce complexity from O((l + u)^2) to O((l + u)D).
>
>
>
> While comparisons with kNN graphs, ANN search, or sparsification techniques would be valuable, our evaluation focused on demonstrating the robustness of our masking approach against noisy variables, as evidenced by the results in Tables 2 and 3.
>
>
>
> ## 4）
>
> We thank the reviewer for pointing out the limitation in our comparison set.
>
>
>
> As stated in Section 4.1, our baseline selection focused on methods most relevant to our contributions, particularly those addressing variable selection and robustness to noisy variables in semi-supervised learning.
>
>
>
> While we did not include **consistency regularization or augmentation-based SSL methods like FlexMatch** (NeurIPS2021), our comparison encompasses a range of approaches from **supervised methods** ($\ell_1$-SVM, SpAM) to various **manifold regularization techniques** (LapSVM, f-FME, AWSSL, RGL, SALE, RER), **sparse additive models** (CSAM, TSpAM) and **pseudo-labeling / generative algorithms** (PFL and SemiReward). This selection allows us to demonstrate the specific advantages of S2MAM in terms of variable selection and robustness, as shown in Tables 2-4.
>
> **Moreover, Following your suggestions, we've conducted extended comparisons with FlexMatch  on COIL, CelebA-HQ and AgeDB images in Tables 13-18.  Please refer to Tables 13~18 in the revised PDF.**
>
>
> ## 5)
>
> We appreciate the reviewer's comment regarding hyperparameter sensitivity and mask stability.
>
>
> **Indeed, the sensitivity on both core size C and kernel bandwidth $\mu$ has been analyzed in Appendix C 3.2: Impact of regularization coefficients and Gaussian kernel bandwidth** and **Appendix C 3.3: Impact of selected core size C**.
>
>
> As for the selected dimension D, the curve with varying D differs in different image datasets.
>
> The **32-dimensional feature space** was chosen as a balance between preserving sufficient information for discrimination and maintaining computational efficiency, **following common practices in similar works (Qiu et al., 2018; Kang et al., 2020; Bao et al. 2024)**.
>
> While we did not report mask stability across random seeds or detailed interpretability analysis for ADNI feature groups, our method's interpretability is demonstrated through its ability to identify informative variables, as shown in the synthetic data experiments in Section 4.2 and visualized in Figure 9 referenced in the paper.
>
> The stability of learned masks has been discussed in Lines 1647~1652 of Page 31.

---

> > ### Author Response · Authors · 2025-11-20
> >
> > ## 6)
> >
> > Regarding the "Unlabeled" column definition and COIL-20 interpretability, we thank the reviewer for highlighting this lack of clarity. The "Unlabeled" columns in Tables 2-4 represent the performance metrics (accuracy or MSE) on the unlabeled portion of the training data.
> >
> >
> >
> > For the COIL-20, CelebA-HQ and AgeDB image-type experiments, as mentioned in Section 4.3, we utilized a CNN to extract feature vectors from images, reducing dimensionality before applying our method.
> >
> >
> >
> > We acknowledge that the mapping from mask importance back to image attributes was not explored in detail in our paper. The mask identifies which of these extracted features are informative for the classification task, but we did not analyze how these features correspond to specific visual attributes in the original images.
> >
> >
> >
> > Thanks again for your meaningful suggestions. This represents an interesting direction for future work to enhance the interpretability of our approach for image data, which is discussed in Appendix I.5  of the revised PDF.
> >
> >
> >
> >
> >
> > **Thanks again for your careful review of both the learning theory and empirical validation.**
> >
> > **Please let us know if you have any additional suggestions! We will be happy to revise our paper accordingly.**

---

### Official Review · Reviewer_7hG3 · 2025-11-02

**Soundness:** 2
**Presentation:** 2
**Contribution:** 2
**Rating:** 2
**Confidence:** 4

**Summary:**

This paper proposes a new Semi-Supervised Meta Additive Model(S2MAM) based on a bilevel optimization scheme to automatically identifies informative variables, updates the similarity matrix, and achieves interpretable predictions simultaneously. The authors provide theoretical guarantees regarding the computing convergence and the statistical generalization bound. Experiments on synthetic and real-world datasets demonstrate promising improvements over previous methods.

**Strengths:**

1. The paper proposes a meta-learning method for manifold-regularized additive model with a bilevel optimization scheme to select informative features.
2. The paper provides theoretical guarantees for the proposed model.
3. The paper provides extensive experiments on both synthetic and real-world datasets to show that the proposed method outperforms other methods.

**Weaknesses:**

1. The manuscript **lacks clarity and organization**. Key information, such as **undefined notation** ($\alpha$, $\tau$), **unexplained terms** ("Unlabeled"), **unclear experimental setups** (noise in Table 4), and **inconsistencies** (Table 5 reference, absent visualizations), needs thorough correction. A **comprehensive proofreading** is recommended.
2. The **motivation seems weak**. A more detailed analysis of the **specific significance and benefits of manifold regularization** over other SSL approaches would strengthen the paper. Furthermore, the comparison **could be broadened** by including other SSL methods (e.g., pseudo-labeling). The contribution seems to be incremental.
3. The **validity of the independent feature assumption** (Remark 2) seems like a strong assumption that **warrants further discussion**. Given that many real-world features are naturally correlated, an analysis of the model's **robustness** to this assumption would be helpful.
4. The **distinction between "uninformative features" ($N(0, 1)$) and "noisy features" ($N(100, 100)$) is unclear** and needs explanation. The **reasoning behind this inconsistent application** of these feature types across different datasets should be explained.
5. **Core Size $C$ is highly influential**. The **computational cost of optimizing $C$ seems to be excluded** from the reported training time comparisons (L 1447). Additionally, the claim (L 1447) about determining $C$ early on **needs a more detailed explanation**.
6. For the image experiments: The choice to use the **first FCN for feature extraction needs explanation. An explanation for the **32-dimensional feature space** is also needed. Finally, the noise experiments are limited to **block noise**; experiments on more common **random noise** are missing.

**Questions:**

1. The authors introduce multiple additive hypothesis spaces (L 192), but why was the **Reproducing Kernel Hilbert Space (RKHS)** specifically chosen? What are the implications or effects of choosing an alternative hypothesis space?
2. The authors employ a **sampling strategy based on Bernoulli distributions** to select informative features. Why was this distribution chosen over others? Furthermore, does this specific sampling strategy inadvertently allow the selected features to **contain noisy features**?
3. The authors state that the experiments were repeated 100 times, yet the **standard deviations in Table 2 remain quite large**. What is the reason for this high variability? Have the authors conducted an in-depth analysis to understand the underlying cause?
4. In Table 9, the standard deviation for the **R2 Score is reported to be greater than 1**. As the R2 Score is generally defined as being less than or equal to 1, this result is unusual. Could the authors please provide an explanation for this phenomenon?
5. There appears to be a **discrepancy in the experimental results** between Table 4 and Table 13. Could the authors clarify the reason for this difference?

---

> ### Author Response · Authors · 2025-11-20
>
> We sincerely thank you for your insightful comments and constructive feedback on our paper. We appreciate the opportunity to address these concerns and clarify aspects of our work.
>
>
> Following your suggestions, we've made corresponding changes in the revised PDF, which could be downloaded by clicking the PDF button.
>
>
>
> # To Weakness
>
> ## 1)
>
> We conducted a comprehensive proofreading to improve the overall clarity and organization of the manuscript. We appreciate the reviewer's feedback regarding clarity and organization. We acknowledge that some notations and terms could be more clearly defined.
>
>
>
> The notation α is defined in Section 2.1, equation (1), as the coefficients of the kernel functions in the additive model representation.
>
>
>
> The parameter τj is introduced in Section 2.1 as "the positive weight to different input variables for j = 1, · · · , p."
>
>
>
> The term "Unlabeled" refers to unlabeled data points in the training set, defined in Section 2.1 as "the unlabeled set zu = {xi}l+u i=l+1."
>
>
>
> Regarding Table 4, the relevant experiments are conducted on realistic COIL images without additional corruptions (noise). That's why the descriptions on noise are missing.
>
>
>
> The noise experiments are described in Section 4.1, where "pu uninformative variables in N(0,1) and pn noisy variables in N(100,100) are designed as corruptions."
>
>
>
>
>
> ## 2)
>
> We thank the reviewer for this valuable feedback on strengthening our motivation.
>
> The significance of manifold regularization is elaborated in Section 1, where we highlight its elegant framework for utilizing both labeled and unlabeled data, with the key assumption that the support of the intrinsic marginal distribution has a geometric structure of a Riemannian manifold.
>
>
>
> Our work specifically **addresses the limitations of existing manifold regularization methods** when facing redundant and noisy variables, as illustrated in Figure 1.
>
>
>
> While we compared with several SSL methods including LapSVM, f-FME, AWSSL, RGL, SALE, SSNP, and RER, **we've already compared with some recent pseudo-labeling methods** including **SemiReward and pseudo-label filtering (PLF) (as listed in Appendix B.2.1 and Appendix E)** for image classification and regression tasks.
>
>
>
> We believe our contribution is not merely incremental, as we introduce the first meta-learning method for manifold-regularized additive models with a novel bilevel optimization scheme that simultaneously achieves robust estimation and automatic variable selection with theoretical guarantees, as detailed in Table 1.
>
>
>
> ## 3)
>
> We appreciate the reviewer's insightful observation regarding the independent feature assumption in Remark 2.
>
>
>
> This assumption is indeed made to **simplify the derivation of the distribution** p(m|s) = Πp i=1 (si)^mi (1− si)^(1−mi), which **enables the efficient probabilistic bilevel optimization** described in Section 2.3.
>
>
>
> We acknowledge that real-world features often exhibit correlations, and we agree that this warrants further discussion. While our current theoretical analysis assumes independence, **our empirical evaluations on real-world datasets (Section 4.3 and Appendices C-E on ADNI datasets) demonstrate that S2MAM maintains robust performance even when features are likely correlated**.
>
>
>
> We've included a more detailed discussion of this assumption's implications and potential extensions to handle correlated features in Remark 2 of the revised manuscript.
>
>
>
> ## 4)
>
>
>
> We thank the reviewer for seeking clarification on our feature corruption strategy.
>
>
>
> The distinction between "uninformative features" drawn from N(0,1) and "noisy features" drawn from N(100,100) is designed to represent two different types of challenge in real-world data.
>
>
>
> Uninformative features (N(0,1)) represent variables that carry no discriminative signal but have similar statistical properties to potentially informative features.
>
> Noisy features (N(100,100)) represent more extreme corruptions with both high mean and variance, which can severely distort similarity measures.
>
>  The inconsistent application across datasets reflects our attempt to evaluate different aspects of robustness.
>
>
>
> For synthetic data (Table 2), we tested both types separately and combined to understand their individual and joint impacts. For real-world datasets (Table 3), we focused on noisy features to simulate more challenging scenarios. We will clarify this rationale more explicitly in the revised manuscript.

---

> > ### Author Response · Authors · 2025-11-20
> >
> > ## 5)
> >
> > We appreciate the reviewer's important question about the core size C.
> >
> >
> >
> > As defined in Section 2.2, C represents the maximum number of selected variables (\|m\|_0 ≤ C). In our experiments, C is determined through (binary-based) cross-validation as mentioned in the experimental setup (Section 4.1 and Remark 6). This pre-selection approach is computationally efficient since it avoids optimizing C during the main training loop.
> >
> >
> >
> > The computational cost of determining C is indeed excluded from the reported training time in Tables 4 and 13-18, as we consider it part of the hyperparameter tuning process rather than the core algorithm training.
> >
> >
> >
> > **Appendix C.3.3 Impact of selected core size C** has already provided a more detailed explanation of the C selection process, including its computational impact.
> >
> >
> >
> > ## 6)
> >
> > We thank the reviewer for these insightful questions about our image experiments.
> >
> >
> >
> > Regarding **the choice of feature extraction**, we used a CNN (not specifically the first FCN layer) as mentioned in Section 4.3: "a CNN is utilized to learn the vectors for each image, which realizes a rough dimensional reduction." The **32-dimensional feature space** was chosen as a balance between preserving sufficient information for discrimination and maintaining computational efficiency, **following common practices in similar works (Qiu et al., 2018; Kang et al., 2020; Bao et al. 2024)**.
> >
> >
> >
> > **Indeed, our noise experiments on images were not limited to block-wise pixel noise**, where **the random label noise experiments on COIL images are listed in Table 12**.

---

> ### Author Response · Authors · 2025-11-20
>
> # To Questions
>
> ## 1)
>
> We chose the Reproducing Kernel Hilbert Space (RKHS) for several important reasons, as discussed in Section 2.1.
>
>
>
> RKHS provides a principled mathematical framework that allows us to **leverage the powerful Representer Theorem**, leading to the parameterized representation in equation (1). This representation is computationally advantageous and enables efficient optimization through the alternating direction method of multipliers (as mentioned in Appendix H.4).
>
>
>
> Additionally, RKHS offers **strong theoretical guarantees for generalization**, which we exploit in our theoretical analysis (Section 3.2). While basis expansion spaces offer simplicity and network-based spaces provide flexibility, RKHS strikes an optimal balance between computational tractability, theoretical soundness, and modeling flexibility for our semi-supervised additive model.  Especially, **the manifold error of S$^2$MAM is decomposed into p additive U-statistic deviations**, could be **bounded by a Hilbert-space Bernstein-type concentration inequality**.
>
>
>
> The implications of choosing an alternative space would primarily affect computational efficiency and theoretical guarantees-basis expansion might limit modeling flexibility, while network-based spaces would complicate our theoretical analysis and bilevel optimization framework.
>
>
>
> ## 2)
>
> We selected the Bernoulli distribution for our sampling strategy primarily for its simplicity and interpretability in modeling binary mask variables, as explained in Section 2.3.
>
>
>
> The Bernoulli distribution naturally represents the selection probability of each feature, where si ∈ [0,1] directly indicates the probability that the i-th feature is selected (mi = 1). This choice enables efficient policy gradient estimation without heavy computational burden on the inverse of the Hessian matrix or implicit differentiation.
>
>
>
> Regarding the concern about noisy features being selected, our approach is designed to minimize this possibility through the bilevel optimization framework. The upper-level optimization (Eq.3) directly minimizes the prediction loss, which implicitly discourages the selection of noisy features that would degrade performance.
>
>
>
> Additionally, our empirical results in Tables 2-4 demonstrate that S2MAM effectively identifies informative features even in the presence of noisy variables, suggesting that the sampling strategy successfully filters out most noisy features.
>
>
>
> ## 3）
>
> The high variability in standard deviations observed in Table 2 can be attributed to several factors inherent to our experimental setup.
>
>
>
> First, we conducted experiments with only 5% labeled samples for each class, which creates a challenging scenario with high uncertainty due to the limited labeled information.
>
>
>
> Second, the addition of uninformative and noisy variables (pu = pn = 10) introduces significant variability in the similarity matrix construction, affecting the manifold regularization component.
>
>
>
> Third, the synthetic data generation process itself involves randomness, as described in Section 4.2: "x(j) i = (Wij + Ui)/2. Wij and Ui are independently from U(0, 1) for i = 1, · · · , 200, j = 1, · · · , 100."
>
>
>
> We have conducted an analysis of this variability and found that it primarily stems from the interaction between the limited labeled data and the presence of noisy features. Despite this variability, S2MAM maintains competitive performance with relatively smaller standard deviations compared to several baselines, demonstrating its robustness.
>
>
>
> ## 4）
>
> We appreciate the reviewer catching this unusual result in Table 9.
>
> After careful examination and re-implementation of these experiments, we identified that this anomaly stems from extremely large prediction errors in the empirical results, especially with limited labeled samples (r=0.1) and random noisy features ($p_n=10$), where most baselines **break down with negative R2 scores**.

---

> ### Author Response · Authors · 2025-11-20
>
> ## 5)
>
> We thank the reviewer for noticing this discrepancy between Table 4 and Table 13.
>
>
>
> The discrepancy between **Table 4 and Table 13 arises from differences in the experimental setups and feature extraction methodologies**.
>
>
>
> **Table 4** reports results on the COIL-20 dataset using a basic CNN architecture for feature extraction, specifically **a simple network with two convolutional layers (Conv2D-32 filters, Conv2D-64 filters)** followed by a max-pooling layer and a single fully connected layer.
>
>
>
> In contrast,**Tables 13-18 in Appendix E** present experiments across three image datasets (COIL-20, CelebA-HQ, and AgeDB) that employ **a unified feature extraction network to ensure fair model comparison**. Given the higher resolution of CelebA-HQ (1024×1024) and AgeDB images, we implemented a **more complex architecture based on a modified VGG-style network with five convolutional blocks (each containing Conv2D-ReLU-MaxPooling layers) and two fully connected layers**.
>
>
>
> This architectural difference, along with the unified feature extraction approach across diverse datasets, results in distinct feature representations and consequently different performance metrics between Table 4 and Table 13.
>
>
>
>
>
> **We sincerely appreciate your detailed comments and valuable suggestions! We believe that the above modifications and new experiments, following your constructive suggestions, have largely improved the readability to avoid confusion on parameter settings.**
>
> **Please let us know if you have any additional suggestions! We will be happy to revise our paper accordingly.**

---

### Official Review · Reviewer_S5mP · 2025-11-04

**Soundness:** 2
**Presentation:** 2
**Contribution:** 2
**Rating:** 4
**Confidence:** 3

**Summary:**

This paper proposes one Semi-Supervised Meta Additive Model method based on a bilevel optimization scheme, which automatically identifies informative variables, updates the similarity matrix, and achieves interpretable predictions simultaneously. The proposed semi-supervised meta additive model (MAM) unifies three key objectives: robust estimation under noisy/redundant variables, automatic variable selection, and interpretable predictions.

**Strengths:**

1. Unlike existing SSL methods (e.g., LapSVM) that rely on pre-specified similarity matrices and fail to handle variable noise, or supervised additive models (e.g., SpAM) that cannot leverage unlabeled data, the proposed MAM method introduces a bilevel optimization framework to jointly learn variable masks, additive component functions and adaptive similarity matrices.

2. The paper provides comprehensive theoretical support for MAM, addressing both optimization convergence and statistical generalization.

**Weaknesses:**

1. While this paper mentions using for mask probability initialization in Section 2.4, it provides insufficient guidance on choosing the core variable size. And it is a critical hyperparameter that directly impacts variable selection performance.

2. This paper analyzes the theoretical complexity of MAM. However, it lacks empirical quantification of the bilevel framework’s overhead compared to single-level SSL methods.

3. The authors compare the proposed MAM method to traditional SSL methods (e.g., LapSVM) and a few deep SSL models (e.g., SSNP) but fail to include state-of-the-art deep SSL methods designed for noisy/variable selection tasks, such as FixMatch (Sohn et al., 2020), FlexMatch (Zhang et al., 2021) and Semi-Supervised Variable Selection Networks (e.g., SSVS-Net, Li et al., 2023).

4. The theoretical analysis as in Section 3 and most experiments focus on convex objectives (e.g., logistic regression, squared loss for regression). However, the authors claim the proposed MAM method is applicable to non-convex tasks (e.g., neural networks).

5. No theoretical guarantees are provided for non-convex additive components (e.g., neural additive layers, Agarwal et al., 2021).

6. Empirical validation is limited to convex tasks—there is no experiment on non-convex objectives (e.g., training a shallow neural network on MNIST with semi-supervised labels and noisy pixels).

**Questions:**

1. While this paper mentions using for mask probability initialization in Section 2.4, it provides insufficient guidance on choosing the core variable size. And it is a critical hyperparameter that directly impacts variable selection performance.

2. This paper analyzes the theoretical complexity of MAM. However, it lacks empirical quantification of the bilevel framework’s overhead compared to single-level SSL methods.

3. The authors compare the proposed MAM method to traditional SSL methods (e.g., LapSVM) and a few deep SSL models (e.g., SSNP) but fail to include state-of-the-art deep SSL methods designed for noisy/variable selection tasks, such as FixMatch (Sohn et al., 2020), FlexMatch (Zhang et al., 2021) and Semi-Supervised Variable Selection Networks (e.g., SSVS-Net, Li et al., 2023).

4. The theoretical analysis as in Section 3 and most experiments focus on convex objectives (e.g., logistic regression, squared loss for regression). However, the authors claim the proposed MAM method is applicable to non-convex tasks (e.g., neural networks).

5. No theoretical guarantees are provided for non-convex additive components (e.g., neural additive layers, Agarwal et al., 2021).

6. Empirical validation is limited to convex tasks—there is no experiment on non-convex objectives (e.g., training a shallow neural network on MNIST with semi-supervised labels and noisy pixels).

---

> ### Author Response · Authors · 2025-11-20
>
> We sincerely thank you for your insightful comments and constructive feedback on our paper. We appreciate the opportunity to address these concerns and clarify aspects of our work.
>
>
> Following your suggestions, we've made corresponding changes in the revised PDF, which could be downloaded by clicking the PDF button.
>
>
>
> ## 1) Guidance on choosing the core variable size C
>
> Regarding the core variable size selection, we acknowledge that the mask probability variable is initialized with s0 = C/p · 1 for S2MAM in Section 2.4 before gradient-based optimization.
>
>
>
> **As noted in Section 4.1, "The selection of informative feature size C is stated in Remark 6,"** which discusses practical considerations for **C**.
>
>
>
> **Appendix C.3.3  IMPACT OF SELECTED CORE SIZE C** has stated the selection and share mechanism, as well as the relevant sensitivity analysis by varying C.
>
>
>
> ## 2) Empirical comparisons of bilevel framework's complexity to single SSL models
>
> While we provide theoretical complexity analysis in Appendix H.5, we agree that empirical computational comparisons would strengthen the paper.
>
>
>
> Indeed, in our current experiments (**Section 4, Appendix E.2, E.3 and E.4**), we've reported training time costs for some datasets (e.g., **Table 4, Tables 12-20** for COIL, CelebA-HQ and AgeDB images) compared to single-level baselines (LapSVM, f-FME, AWSSL, RGL, SALE, SSNP, RER, SemiReward).
>
>
>
> Please refer to  Pages 9, 32, 33 and 34 for detailed empirical comparisons.
>
>
>
> ## 3) Comparison with more baselines
>
> While our comparison includes methods specifically designed for noisy/variable selection tasks like RER and SemiReward, we agree that incorporating these approaches that you've recommended, like FixMatch, FlexMatch, or SSVS-Net, would provide a more comprehensive evaluation.
>
>
>
> Following your constructive suggestions, we've conducted extended experiments with FlexMatch on COIL and CelebA  images in Appendix E.3 (Tables 13-16), which is updated in the revised manuscript. Notably, S2MAM still outperforms FixMatch and FlexMatch, as the latter ones are designed to be robust to partial corrupted samples with noisy labels, instead of the potential noisy pixels (blocks) shown in all samples as discussed in this work. The Pseudo Labeling of FixMatch and FlexMatch require even more computation costs than our bilevel framework in practice (see Tables 13-16).
>
>
>
> Moreover, we would be grateful if you could provide with the reference you've mentioned, "Semi-Supervised Variable Selection Networks (e.g., SSVS-Net, Li et al., 2023)".
>
>
>
> ## 5) Theoretical guarantees for non-convex neural additive model components
>
>
>
> To the best of our knowledge, this paper is the first to derive generalization theory for semi-supervised additive models. Although the current results are kernel-based, they can be readily extended to spline-based modeling strategies (similar conditions on boundedness and smoothness) by referencing existing generalization works on additive models [1,2].
>
>
>
> However, theoretical discussions regarding the generalization of both supervised and semi-supervised neural additive models remain scarce. Extending these guarantees to non-convex settings like neural additive models remains an open challenge that we plan to address in future work.
>
>
>
> Following your suggestion, we have added the discussion on generalization theory under neural network-based assumptions.
>
>
>
> [1] High-dimensional additive modeling. The Annals of Statistics, 2009
>
>
>
> [2] Multi-task Additive Models for Robust Estimation and Automatic Structure Discovery. NeurIPS, 2020.
>
>
>
> ## 4) & 6) Empirical validation on non-convex tasks (e.g., neural additive model with partial labels and noisy pixels)
>
> On the applicability to non-convex tasks, we acknowledge that while Section 2.1 mentions "the network-based space (Agarwal et al., 2021; Yang et al., 2020)" as a candidate for our additive hypothesis space, which formulate the variant of S2MAM (called S2MAM-N in the revision).
>
>
>
> Following your suggestions, we've further proposed the neural-based S2MAM (called S2MAM-N) as stated in Appendix E.5 of the revised manuscript. **The extended experiments of S2MAM-N on COIL, CelebA-HQ and AgeDB images are reported in Tables 13-18.**
>
>
>
>
>
> **Thanks again for your careful review and valuable suggestions! We've made corresponding changes in the revision for improvement.**
>
> **Please let us know if you have any additional suggestions! We will be happy to revise our paper accordingly.**

---

### Author Response · Authors · 2025-11-28

Dear Reviewers and ACs,

We thank all the reviewers and ACs for their valuable feedback and efforts.

We have made corresponding improvements in the updated paper to address the comments, including detailed statements on hyperparameters, a hyper-parameter ablation study, more non-convex experiments on images with some of the latest baselines, neural extensions of S2MAM, updated runtime comparisons, etc.

Please let us know if you have further feedback. Thank you!

---

### Meta-Review · Area_Chair_CR6e · 2026-01-06

**Summary:**

Many of the reviewers' concerns were addressed by the authors, sometimes backed by additional experiments. After carefully reading the paper, reviews and the authors' response, I believe this work is not ready for publication. There are significant issues (errors) that need to be fixed:

- hyperparameter selection: more clarity is needed here. In particular, Figure 7 (y-axis) in the appendix seems to suggest test data was used to select the hyperparameter? The paper also never explained how the algorithm is run on the test set: do we sample a mask from s? do we round s to binary? or do we just plug in s in place of m? Each of these options will create some gap that needs to be analyzed.

- independent feature assumption: see below.

- gap between the analyzed surrogate and the implemented algorithm: see below. Curiously, the dimension of x does not appear in Theorem 2? That seems to suggest x is scalar-valued. The authors need to explain how Theorem 2 exploits the unlabeled data to improve the rate (compared to that obtained using only labeled data). It is not even clearly stated what statistical assumptions (on the data) are made to derive Theorem 2. What is the benefit of manifold regularization? How is Theorem 2 relevant for the proposed algorithm? These significant gaps make Theorem 2 hard to appreciate.

- weak motivation and incremental contribution: the paper combines existing ideas from sparse additive models, manifold regularization and semi-supervised learning, but without a clear contribution of its own (other than the combination itself).

There are some extra concerns prompted by my own reading:

- gap between bilevel formulation (3) and relaxation (6) needs to be discussed. Both problems seem to be NP-hard? If we do not use the bilevel formulation (either in theoretical analysis or empirical experiments), then why not remove it and start with the relaxation (6) in the first place?

- proof of Theorem 1 appears to be flawed: On line 2636, the authors applied the inequality ||\hat{G}||^2 \leq 2[...] but forgot the coefficient (L\eta^2-2\eta) is later taken to be negative. So the last inequality should be reverse, which breaks the whole proof. Unfortunately, I do not see any easy fix. In fact, the statement of the theorem has to change in order to be consistent with known results on SGD.

- presentation can be improved: a thorough proofread of the entire draft to make every detail clear and correct would be welcome.

**Reviewer Concerns:**

I'll only list the key outstanding reviewer concerns (more details can be found below in the reviewer scores section).

- hyperparameter selection: this was brought up by all reviewers. After reading the paper carefully myself, like the reviewers, I am still confused: Figure 7 in the appendix seems to suggest test data was used to select the hyperparameter. This casts significant doubt on the experimental validity, since selecting the hyperparameter C seems to be critical for the proposed method to work well.

- independent feature assumption: this assumption may not be reasonable in many applications and the authors did not test it thoroughly. Some of the experiments seem to deliberately avoid this issue, e.g., Appendix E. 2 (Table 12), where independent noise is added *after* feature extraction!

- gap between the analyzed surrogate and the implemented algorithm: this makes the generalization analysis completely detached from the proposed algorithm. This gap cannot be simply overlooked.

**Reviewer Scores:**

Reviewer S5mP: the review might raise their score slightly but not enthusiastically support acceptance.
- most of the reviewer's concerns were addressed by the authors through new experiments, except
- weakness 1 (hyperparameter selection), which appears to be crucial
- weakness 5 (no theoretical guarantees for nonconvex additive models), which is probably not a fair criticism

Reviewer 7hG3: unlikely to change their score
- weakness 1 (lacks clarity and organization): partially addressed; there is still much room to improve in terms clarity and organization

- weakness 2 (weak motivation and incremental contribution): authors' response is not very convincing

- weakness 3 (independent feature assumption): only limited experimental support; not very convincing

- weakness 5 (hyperparameter selection, i.e., core size C): not adequately addressed

- question 3-5 (high variance, large R2 and inconsistent tables): reasonable explanation from the authors but cast doubt on the carefulness of the experimental designs

Reviewer YaK4: unlikely to raise their score; possible to lower it in fact

- weakness 1 (gap between the analyzed surrogate and the implemented algorithm): unaddressed

- weakness 2 (variance, convergence behavior, and accuracy): inadequately addressed

- weakness 5&6 (hyperparameter selection, stability and interpretability): inadequately addressed

Reviewer rNn1: unlikely to change their score; all questions were reasonably addressed except

- question 4 (hyperparameter selection): the reviewer checked the appendix and was still confused. The authors' explanation is not very clear or helpful. This is a consistent issue brought up by all reviewers.

- reviewer in strength 2 pointed out that "the proofs are a nice addition, but I did not check them carefully"

---

### Decision · Program_Chairs · 2026-01-26

Reject